

# Exponential networks for linear partitions

Sibasish Banerjee[1], Mauricio Romo[2,3], Raphael Senghaas[4] and Johannes Walcher[4]

**1** Institut des Hautes Études Scientifiques, 91440 Bures-sur-Yvette, France
**2** Center for Mathematics and Interdisciplinary Sciences Fudan University,
Shanghai 200433, China
**3** Shanghai Institute for Mathematics and Interdisciplinary Sciences, Shanghai 200433, China
**4** Institute for Mathematics and Institute for Theoretical Physics,
Ruprecht-Karls-Universität Heidelberg, 69120 Heidelberg, Germany

## Abstract

Previous work has given proof and evidence that BPS states in local Calabi-Yau 3-folds can be described and counted by exponential networks on the punctured plane, with the help of a suitable non-abelianization map to the mirror curve. This provides an appealing elementary depiction of moduli of special Lagrangian submanifolds, but so far only a handful of examples have been successfully worked out in detail. In this note, we exhibit an explicit correspondence between torus fixed points of the Hilbert scheme of points on $\mathbb{C}^2 \subset \mathbb{C}^3$ and anomaly free exponential networks attached to the quadratically framed pair of pants. This description realizes an interesting, and seemingly novel, "age decomposition" of linear partitions. We also provide further details about the networks' perspective on the full D-brane moduli space.

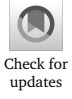

## Contents

> *When you have eliminated all which is impossible,*
> *whatever remains, however improbable, must be the truth.*
> (Sherlock Holmes)

# 1 Introduction and summary

Spectral networks, formally introduced in [1], are a combinatorial tool for studying BPS spectra of 4d $\mathcal{N} = 2$ supersymmetric field theories of "class $S[A_{K-1}]$", $K = 2, 3, \ldots$ The basic geometric setup includes a $K$-sheeted "spectral" covering $\Sigma \to C$ of punctured Riemann surfaces equipped with a meromorphic differential $\lambda$ that arises as the restriction of the tautological (Liouville) one-form by means of an embedding $\Sigma \hookrightarrow T^{\vee}C$. Physically, the "Seiberg-Witten curve" $\Sigma$ parameterizes the moduli of a canonical defect in the theory, and the space of coverings its Coulomb branch. With respect to a local trivialization of $T^{\vee}C$ with coordinates $(x, y)$, we have $\lambda = \lambda_i = y_i dx$ on the $i$-th sheet of the covering, corresponding to the various vacua of the defect theory. "BPS trajectories" tracing supersymmetric $(i, j)$-solitons on the defect are integral curves of the differential $\lambda_{(ij)} = \lambda_i - \lambda_j$ on $C$, with fixed phase $\vartheta \in \mathbb{R}/2\pi\mathbb{Z}$. A trajectory of type $(i, j)$ can begin and end at corresponding branch points of the covering, and at a preceding junction of an $(i, k)$ and $(k, j)$ trajectory. The topology of the resulting network depends in an intricate way on the geometric parameters: Jumps induced by variations with $\vartheta$ signal 4d BPS states carrying electric-magnetic charge in $H_1(\Sigma, \mathbb{Z})$, while the dependence of these degeneracies on the data of the covering satisfies the celebrated Kontsevich-Soibelman wall-crossing formula [2]. See [1,3,4] and related literature for full details of the construction.

In string/M-theory, the 4d $\mathcal{N} = 2$ theory describes the low-energy dynamics of the 6d $\mathcal{N} = (2, 0)$ theory living on $K$ M5-branes wrapped on $C$. By a much studied chain of dualities (see e.g. [5] for some fairly recent work) this is equivalent to type II string theory compactified on an open, singular, Calabi-Yau threefold $X$, in a limit in which its mirror, $Y$, reduces to the spectral geometry discussed above. In this frame, BPS states arise from D-branes wrapped on special Lagrangian submanifolds of $Y$ that are fibered by spheres and tori over those BPS trajectories, eventually lifted to geodesics on the Seiberg-Witten curve. This is in fact how the above description was first obtained, for $K = 2$, in [6].

Exponential networks, introduced in [7] and further developed in [8–11] are the variant of this construction, also initiated in [6] (see also [12]), that captures the BPS spectrum of M-theory on $S^1 \times X$, before taking any limit. In the geometric model, this amounts to removing the zero section from $T^{\vee}C$, at the price of working with the non-exact symplectic manifold $\left((T^{\vee}C)^{\times}, dx \wedge \frac{dy}{y}\right)$, a "multi-valued" Liouville one-form with logarithmic branch cuts on $\Sigma \to C$, or else its universal covering $\widetilde{\Sigma}$, as advocated in [8]. This multi-valuedness is intimately linked to Kaluza-Klein momentum around the $S^1$, i.e., the presence of the D0-brane. Exponential networks also feature a novel type of logarithmic singularity to account for non-compact D4-branes. As an important technical consequence of these junction rules (see Fig. 4), the generalization of the non-abelianization map of [1,13] depends locally on an infinite amount of soliton data (in distinction to spectral networks, which are "locally finite"

in this sense). This makes the Gaiotto-Moore-Neitzke/Kontsevich-Soibelman formalism significantly more complicated [11], but gives a fairly direct access to the intriguing enumerative geometry of special Lagrangian submanifolds [9, 10]. This remains the principal motivation also for the present paper.

In the meantime, exponential networks have also been studied for the purpose of exact WKB solution of $q$-difference equations satisfied by the open/closed topological string partition function [14, 15]. Relations to hyperkähler metrics have been anticipated in [16], and to integrable systems in [17, 18]. Most recently, a construction reminiscent of exponential networks has appeared in the calculation of periods of certain compact Calabi-Yau manifolds defined as Hadamard products of families of elliptic curves [19].

At the level of examples, exponential networks have been verified to capture the finite-mass BPS spectrum of some of the simplest toric Calabi-Yau threefolds, including the conifold [7], local $\mathbb{P}^2$ [7,9], and the local Hirzebruch surface $\mathbb{F}_0$ [10]. In [9] it was shown by a careful study of the non-abelianization map that exponential networks on the pair of pants mirror to $\mathbb{C}^3$

$$\Sigma = \{x - y - y^2 = 0\} \xrightarrow{\ x\ } C = \mathbb{C}_x^\times\,, \tag{1}$$

account rigorously for the expected degeneracy of a single D0-brane in flat complex three-dimensional space.

The spectrum of *framed* BPS states, however, including non-compact D4-, or, most ambitiously, the D6-brane, has been less forthcoming. In [7], it was shown that with a suitable cut-off, the BPS trajectory running into the $(\log x)^2$ singularity of (1) interacts with the D0-brane exactly as expected from a D4-brane wrapped on the toric divisor that is dual to $C$, interpreted as the classical moduli space of the D-brane probe realizing SYZ mirror symmetry in the sense of Hori-Vafa [20, 21]. Namely, the spectrum of open strings encodes the fields of the quiver,

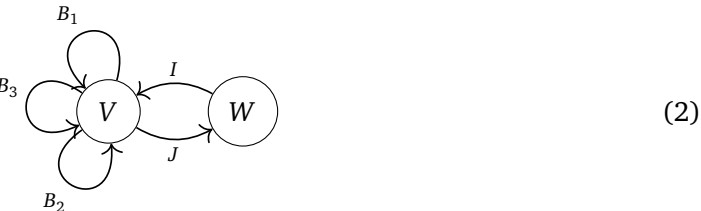 $$\tag{2}$$

and the count of holomorphic disks, the superpotential

$$\mathcal{W} = \mathrm{Tr}_V\big((B_1 B_2 - B_2 B_1 - IJ)B_3\big)\,. \tag{3}$$

Remarkably [7], the representations of (2) with $B_3 = 0$ are equivalent to those of the ADHM quiver

$$\tag{4}$$

with relation

$$\partial_{B_3}\mathcal{W} = B_1 B_2 - B_2 B_1 - IJ = 0\,. \tag{5}$$

Moreover, by a comparison of central charges, the parameter of the cut-off was interpreted as the total B-field piercing the toric divisor, in agreement with its role as a $\theta$-stability condition on the quiver. On general grounds, this implies that the (suitably restricted) spectrum of exponential networks in the homology classes of $N$ D4- and $k$ D0-branes must agree with the

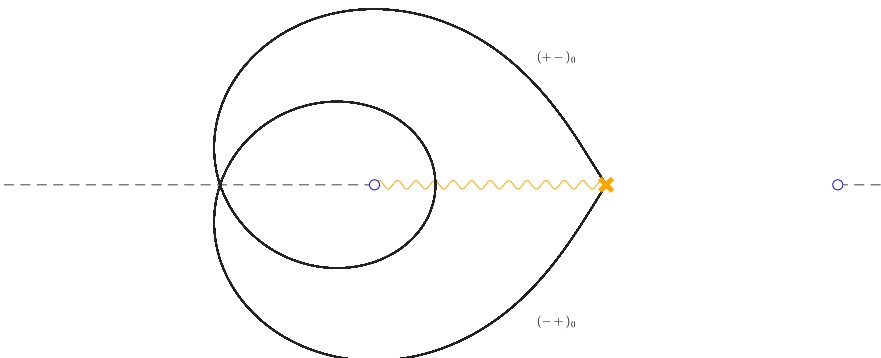

Figure 1: Critical finite web corresponding to a single D0-brane at the fixed point of the torus action, $\vartheta = 0$. The wavy line is a squareroot, the dashed line a logarithmic branch cut. The labels are for counter-clockwise orientation.

moduli of stable representations of the ADHM quiver (4) with $\dim V = k$ and $\dim W = N$, parameterizing self-dual $SU(N)$ connections with instanton number $k$. Specializing to $N = 1$, $k = n$, the moduli space of non-commutative instantons can be identified [22, 23] with the Hilbert scheme of points on $\mathbb{C}^2$ [24, 25], such that the Donaldson-Thomas-like enumerative invariant defined in [11] must equal $N_n$, the number of linear partitions of $n$. This is the well-known result for the (equivariant) Euler characteristic of the Hilbert scheme, or else the number of fixed points under the natural $(\mathbb{C}^\times)^2$ torus action lifted from the underlying $\mathbb{C}^2$. This leads to a simple strategy for reproducing the counting of the invariant in the B-model.

In [7, 11], it was pointed out that while the full $(\mathbb{C}^\times)^2$-action is not expected to be realized geometrically at the level of exponential networks, the first Betti number of the cycle over the generic D0-brane network is equal to 2, and the basis of the cohomology can naturally be interpreted as real generators of the corresponding flow. The distinguished finite web that was identified with the fixed point of the torus action begins and ends at the branch point, and does not include any trajectories emanating from the self-intersection, see fig, 1. Following this reasoning, and guided by the comparison with the representations of (4), similar finite webs were constructed in correspondence with torus fixed points for $n = 1, 2$ and 3 D0-branes. Starting at $n = 4$, however, there seemed to be more of these "deterministic" finite webs than partitions, and of rapidly growing complexity.

In this paper, we will resolve all these issues, and develop an explicit correspondence between $(\mathbb{C}^\times)^2$-fixed quiver representations, labeled by Young diagrams, and finite webs in exponential networks. There are two new key ingredients, which as we will discuss represent theoretical progress for general networks. The first is a proper account of the role of holomorphic disks appearing as in equation (5). In the example, we shall see concretely that the uncanceled disks correspond to a violation of the F-term relations in the quiver, and will thereby eliminate all extraneous networks observed in [7]. The second new element is a type of network that was ignored in [7] because it seemed to require an unnatural "non-deterministic" tuning of D0-branes "away from the origin". By a careful analysis of the cycles on the universal cover, we will show that, quite to the contrary, these networks are in fact proper torus fixed points, and that they moreover appear in just the right place and number to balance the total count.

Section 2 is a self-contained summary of ADHM representation theory. Section 3 will review the proper counting procedure via non-abelianization for exponential networks according to [11], and explain the resolution of the puzzles of [7], guided by examples of low weight.

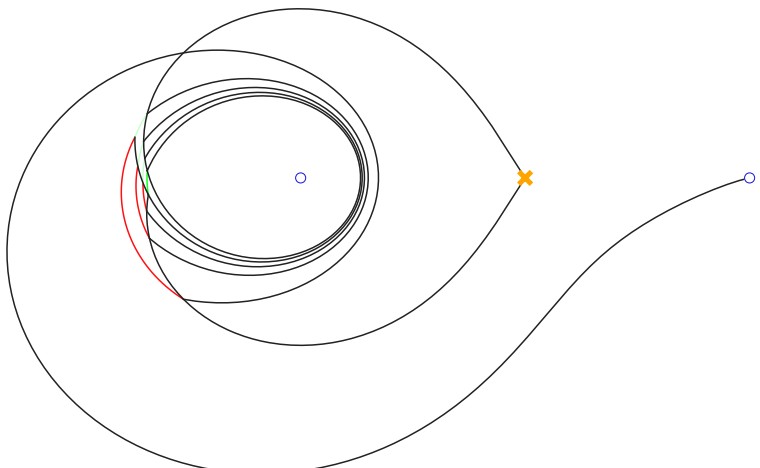

Figure 2: The finite web representing the bound state of a D4-brane with 11 D0-branes corresponding to the partition $(5, 3, 2, 1)$.

Section 4 describes, to the best of our current abilities, the algorithm for matching exponential networks to linear partitions. To whet the appetite, we display in fig. 2 the exponential network corresponding to $11 = 5 + 3 + 2 + 1$. Section 5 contains a brief description of possible research directions to explore in the future, in particular devising a strategy to describe the A-brane moduli space in detail, as was done in a closely related fashion, but a different context, in [26].

## 2 Recapitulation of quiver moduli

Much as in [7], one of our guides for unravelling the intricate combinatorics of exponential networks is the expected correspondence between finite webs on a fixed geometry and stable representations of a suitable BPS quiver with relations. The basic physical idea, going back to [27, 28], is to build up the entire BPS spectrum from a finite set of "basic objects", by bound state formation via tachyon condensation in the effective world-volume theory, which in the case of BPS states on Calabi-Yau threefolds is a supersymmetric quantum mechanics with 4 supercharges. In the quiver description, the elementary objects correspond to the nodes, the dimension vector determines the gauge group, and the arrows are matter fields in bifundamental representations. The relations give the F-terms, and the stability condition the D-terms. Bound states corresponding to supersymmetric vacua are solutions to the D- and F-term equations, modulo the product of unitary gauge groups sitting at each node. Equivalently, one may consider the algebraic quotient of the set of "stable" representations modulo the complexified gauge group, where stability (in the sense of Mumford) is defined by King's $\theta$-stability [29] with respect to a suitable "central charge" function on the set of nodes, which corresponds physically to the Fayet-Iliopoulos parameters entering the D-term constraints.

In the simplest examples, stable quiver representations modulo gauge equivalence give rise to isolated vacua, and can hence directly be identified with single-particle BPS states. In the presence of continuous deformations, one should strictly speaking first work out the full moduli space, and then quantize its effective dynamics. In the case at hand, (4) viewed as a subquiver of (2), we can localize with respect to the $(\mathbb{C}^\times)^2$ flavor symmetry to effectively identify a basis of the space of BPS states of given charge directly with the torus fixed points,

which are in a well-known correspondence with linear partitions/Young diagrams. In order to identify the corresponding finite webs in exponential networks, it will be necessary to have a detailed description of the actual quiver representations, to which we now turn.

## 2.1   $\text{Hilb}^n(\mathbb{C}^2)$ from GIT

Following concretely [25] and [22], a representation of the ADHM quiver 4 is a tuple $M = (V, W, B_1, B_2, I, J)$ where $V, W$ are finite-dimensional complex vector spaces, and $B_1, B_2 \in \text{End}(V)$, $I \in \text{Hom}(W, V)$, and $J \in \text{Hom}(V, W)$ are such that $[B_1, B_2] + IJ = 0$ (F-term relations). We are interested in such representations with fixed dimension vector $(\dim(V), \dim(W)) = (n, 1)$ for $n = 0, 1, \ldots$ Since the diagonally embedded $\mathbb{C}^\times = GL(1, \mathbb{C}) \hookrightarrow GL(V) \times GL(W)$ acts trivially on arrows, while $GL(W) = GL(1, \mathbb{C})$, it is sufficient to discuss the quotient by $GL(V) \cong GL(n, \mathbb{C})$. There are different points of view on this quotient, each with its own merits.

The naive set-theoretic quotient of course is ill-behaved, i.e., it is not an algebraic variety. A much better notion is that of a categorical quotient. Generally speaking, a categorical quotient of an object $\mathcal{X}$ equipped with the action of an algebraic group $\mathcal{G}$ is a morphism $\pi : \mathcal{X} \to \mathcal{Y}$ to an object $\mathcal{Y}$ with trivial $\mathcal{G}$-action that is $\mathcal{G}$-invariant and universal, i.e., any $\mathcal{G}$-invariant morphism $\mathcal{X} \to \mathcal{Z}$ factors uniquely through $\pi$. In the case of interest, in which

$$\mathcal{X} = \text{Spec}\big(A := \mathbb{C}[\mathcal{V}_n^\vee]/([B_1, B_2] + IJ)\big), \tag{6}$$

is an affine variety, obtained by "solving" the F-term relations inside of the vector space

$$\mathcal{V}_n = \text{End}(V) \oplus \text{End}(V) \oplus \text{Hom}(W, V) \oplus \text{Hom}(V, W), \tag{7}$$

with coordinates (the entries of) $B_1$, $B_2$, $I$, and $J$, and $\mathcal{G} = GL(V)$, one can define the GIT quotient as [25, 30]

$$\mathcal{X} \twoheadrightarrow \mathcal{X} /\!/ \mathcal{G} := \text{Spec}(A^{\mathcal{G}}), \tag{8}$$

where $A^{\mathcal{G}} \subset A$ denotes the $\mathcal{G}$ invariants in the coordinate ring, $A$. The GIT quotient is always a categorical quotient and always exists for affine schemes [31]

As it stands, the GIT quotient (8), which is simply isomorphic to the symmetric product $\text{Sym}^n(\mathbb{C}^2)$, and to Uhlenbeck's partial compactification in this case, is still singular as an algebraic variety. One way to resolve the singularities follows from its description as a hyperkähler quotient. To this end, we first recall that for physics purposes, the vector spaces $V$ and $W$ should also carry a hermitian structure, and the quotient should only be taken with respect to the real Lie group $G = U(n) \subset GL(n, \mathbb{C}) = \mathcal{G}$, but after solving also the D-term constraints,

$$[B_1, B_1^\dagger] + [B_2, B_2^\dagger] + II^\dagger - J^\dagger J = 0, \tag{9}$$

where the adjoint is taken with respect to the chosen inner product. In this instance, the bijection between this "symplectic quotient" and the algebraic GIT quotient is a consequence of the Kempf-Ness theorem. Remarkably, however, the action of $G = U(n)$ on the ambient vector space $\mathcal{V}_n$, given explicitly by

$$g \cdot (B_1, B_2, I, J) = (gB_1 g^{-1}, gB_2 g^{-1}, gI, Jg^{-1}), \quad g \in U(n), \tag{10}$$

preserves the full hyperkähler structure defined by [22]

$$\hat{i} \cdot (B_1, B_2, I, J) = (iB_1, iB_2, iI, iJ),$$
$$\hat{j} \cdot (B_1, B_2, I, J) = (-B_2^\dagger, B_1^\dagger, -J^\dagger, I^\dagger),$$
$$\hat{k} = \hat{i}\hat{j},$$

and as such is associated with the hyerkähler moment map $\tilde{\mu} : \mathcal{V}_n \to \mathbb{R}^3 \otimes \mathfrak{u}^\vee(n)$, whose components are related to F- and D-terms according to

$$2i\mu_{\hat{i}} = 2\mu_\mathbb{R}(B_1, B_2, I, J) = [B_1, B_1^\dagger] + [B_2, B_2^\dagger] + II^\dagger - J^\dagger J \,, \tag{11}$$

$$\mu_{\hat{j}} + i\mu_{\hat{k}} = \mu_\mathbb{C}(B_1, B_2, I, J) = [B_1, B_2] + IJ \,. \tag{12}$$

This results in the triple identification

$$\mathcal{V}_n /\!/\!/\, U(n) := (\mu_\mathbb{R}^{-1}(0) \cap \mu_\mathbb{C}^{-1}(0))/U(n) \cong \mu_\mathbb{C}^{-1}(0) /\!/\, GL(n, \mathbb{C}) \cong \mathrm{Sym}^n(\mathbb{C}^2) \,, \tag{13}$$

of singular moduli, and suggests the natural resolution [24] by deformation of the D-term equation (9) to

$$[B_1, B_1^\dagger] + [B_2, B_2^\dagger] + II^\dagger - J^\dagger J = \zeta \, \mathrm{Id} \,, \tag{14}$$

with $\zeta \in \mathbb{R}$ a resolution parameter.[1] The resulting, generically smooth moduli space is

$$\mathcal{V}_n /\!/\!/_\zeta U(n) := \left( \mu_\mathbb{R}^{-1}(\zeta \, \mathrm{Id}) \cap \mu_\mathbb{C}^{-1}(0) \right)/U(n) \,, \tag{15}$$

which as an algebraic variety turns out to be isomorphic to the Hilbert scheme of points $\mathrm{Hilb}^n(\mathbb{C}^2)$.

The resolution of the moduli space can also be interpreted purely algebraically in the framework of GIT, which also leads to a concrete derivation of the identification with the Hilbert scheme. Algebraically, this amounts to replacing the affine GIT quotient (8) with a projective version, built with the help of an ample (in this case, trivial) line bundle $\mathcal{L} = \mathcal{X} \times \mathbb{C} \to \mathcal{X}$, that is equipped with the lift of the $\mathcal{G}$-action twisted by a character $\chi : \mathcal{G} \to \mathbb{C}^\times$. Namely, one defines

$$\mathcal{X} /\!/_\chi \mathcal{G} = \mathrm{Proj}(R^\mathcal{G}) \,, \tag{16}$$

where

$$R = \bigoplus_{k \geq 0} \Gamma(\mathcal{X}, \mathcal{L}^{\otimes k}) \,, \tag{17}$$

is graded by $\chi$, and shows that this parameterizes precisely those "stable" orbits on whose closure $d\chi$ satisfies a certain positivity condition (such that they are intersected precisely once by the deformed D-terms). On the other hand, the degree 0 part of $R$ being equal to $A$ in 6 defines a canonical resolution map $\mathrm{Proj}(R^\mathcal{G}) \to \mathrm{Spec}(A^\mathcal{G})$.

For quiver representations, stability can be phrased in terms of so-called $\theta$-stability, introduced by King in [29]. In the case of interest (4), a $\theta$-stability parameter is just a pair of integers $(\theta_V, \theta_W)$, viewed as an infinitesimal character on $GL(V) \times GL(W)$. The character $\chi_\theta$ in the sense of (16) is given for $g = (g_V, g_W) \in GL(V) \times GL(W)$ by

$$\chi_\theta(g) = (\det g_V)^{\theta_V} (\det g_W)^{\theta_W} \,. \tag{18}$$

A representation $M$ is *$\theta$-semistable* if $\theta$ is trivial on its stabilizer, i.e.,

$$\theta(M) = \theta_V \dim(V) + \theta_W \dim(W) = 0 \,,$$

and non-negative on all orbit closures, i.e., $\theta(M') \geq 0$ for all subrepresentations $M' \subseteq M$. A representation $M$ is *stable* if it is semistable and $\theta(M') = 0$ only for $M' = 0$ or $M$.

To work out the stable representations, we note that for fixed dimension vector $(n, 1)$, the condition $\theta(M) = 0$ allows essentially three inequivalent stability parameters: $(\theta_V, \theta_W) = (-1, n)$, $(0, 0)$ and $(1, -n)$. In the trivial case, $(\theta_V, \theta_W) = (0, 0)$, any representation is semi-stable, but only simple modules with either $V = 0$ or $W = 0$ are stable, and

---

[1] $\zeta$ can be given the physical interpretation as non-zero $B$-field, making the world-volume of a D4-brane into a non-commutative space [22].

$\mathcal{X} /\!/_0 \, \mathcal{G} = \mathcal{X}^{\text{s.s.}}/\mathcal{G}$ simply recovers the affine GIT quotient. In the other cases, we use the fact ( [25], Lemma 2.8, valid only for $\dim(W) = 1$), that the subspace $S \subset V$ generated by the image of $I$ under the action of $B_1$ and $B_2$, is annhilated by $J$. This implies that $M' = S \oplus 0$ and $M'' = S \oplus W$ are subrepresentations of $M$. Then, for $M$ to be stable, we require that either $S = 0$ or $\theta(M') = \theta_V \dim(S) > 0$, i.e., $\theta_V > 0$. When $S = 0$, both $W$ and $\ker J$ are subrepresentations of M. The former implies that $\theta_W > 0$ and $\theta_V < 0$. Thus $\ker J = 0$, because $\theta(\ker J) = \theta_V \cdot \dim \ker J$ which implies $\dim(V) \leq 1$ to preserve stability. In this case, the only possible dimension vectors are $(0, 1)$ and $(1, 1)$. Otherwise, $\theta_V > 0$, meaning we can fix $\theta = (1, -n)$, implies that $M$ is automatically stable against any subrepresentation that is fully contained in $V$. For the representation $M'' = S \oplus W$ we compute,

$$\theta(M'') = \dim(S) - n \, . \tag{19}$$

This is non-negative if and only if $S = V$. This is also sufficient for stability, since it implies that any subrepresentation containing a non-zero vector of $W$, must already be all of $M$, and hence $J \equiv 0$.

To summarize, the stable representations for $(\theta_V, \theta_W) = (1, -n)$ are exactly those of dimension $(n, 1)$ such that image of $I$ generates all of $V$ under the action of $B_1$ and $B_2$. Fixing a basis 1 of $W$, we will refer to $I(1)$ as a "stability vector". In the end, the moduli space of $\theta$-stable representations, for $\theta = (1, -n)$, modulo the action of $\mathcal{G}$ can be written as

$$\mathcal{X} /\!/_{\chi_\theta} \, \mathcal{G} = \left\{ (B_1, B_2, I) \, \middle| \, \begin{array}{ll} \text{(i)} & [B_1, B_2] = 0 \\ \text{(ii)} & \text{(stability) there exists} \\ & \text{no subspace } S \subsetneq \mathbb{C}^n \text{ such that} \\ & B_{1,2}(S) \subset S \text{ and } I(1) \subset S \end{array} \right\} \bigg/ GL(n, \mathbb{C}) \, . \tag{20}$$

This algebraic description of the moduli space can be identified with $\text{Hilb}^n(\mathbb{C}^2)$, the Hilbert scheme of $n$ points on $\mathbb{C}^2$. Formally, $\text{Hilb}^n(\mathbb{C}^2)$ is defined as the moduli space of 0-dimensional closed subschemes of $\mathbb{C}^2$ whose coordinate ring as a vector space is isomorphic to $\mathbb{C}^n$. This is equivalent to an ideal $I \subset \mathbb{C}[z_1, z_2]$ such that $\mathbb{C}[z_1, z_2]/I \cong \mathbb{C}^n$. To identify the Hilbert scheme with (20), set $V = \mathbb{C}[z_1, z_2]/I$, let $B_{1,2}$ correspond to multiplication by $z_{1,2}$ which obviously commute as endomorphisms of $V$, and the stability vector be given by $1 \in \mathbb{C}[z_1, z_2]/I$. The map of GIT quotients

$$\mathcal{X} /\!/_{\chi_\theta} \, \mathcal{G} \to \mathcal{X} /\!/ \, \mathcal{G} \, , \tag{21}$$

is the Hilbert-Chow morphism $\text{Hilb}^n(\mathbb{C}^2) \to \text{Sym}^n(\mathbb{C}^2)$. The identification

$$\mathcal{X} /\!/_{\chi_\theta} \, \mathcal{G} \cong \left( \mu_{\mathbb{R}}^{-1}(i \cdot d\chi_\theta) \cap \mu_{\mathbb{C}}^{-1}(0) \right) / U(n) \, , \tag{22}$$

which shows that the deformed of the D-term 14 is the same as choosing a $\theta$-stability parameter for the quiver representations, can be viewed as consequence the Kempf-Ness theorem.

## 2.2 Torus fixed points and Young diagrams

By definition, a stable representation of (4) with $(\dim V, \dim W) = (n, 1)$, identified with a point in $\text{Hilb}^n(\mathbb{C}^2)$ via (20), is a fixed point for the natural $(\mathbb{C}^\times)^2$-action induced from the rescaling of the linear coordinates on $\mathbb{C}^2$ iff there exists a homomorphism $\lambda : (\mathbb{C}^\times)^2 \to GL(n, \mathbb{C})$ satisfying the following conditions:

$$\begin{aligned} t_1 B_1 &= \lambda(t)^{-1} B_1 \lambda(t) \, , \\ t_2 B_2 &= \lambda(t)^{-1} B_2 \lambda(t) \, , \\ I &= \lambda(t)^{-1} I \, , \end{aligned} \tag{23}$$

for all $t = (t_1, t_2) \in (\mathbb{C}^\times)^2$. By (23) $\lambda$ is unique for any such fixed point, and induces a weight decomposition

$$V = \bigoplus_{k,l} V(k,l), \tag{24}$$

where

$$V(k,l) = \left\{ v \in V \mid \lambda(t) \cdot v = t_1^k t_2^l v \right\}.$$

It follows that $B_1 V(k,l) \subset V(k-1,l)$ and $B_2 V(k,l) \subset V(k,l-1)$. Since $I(1) \in V(0,0)$ is one-dimensional, while $V$ is spanned by elements of the form $B_1^k B_2^l \cdot I(1)$ (with $k,l \in \mathbb{Z}_{\geq 0}$), this implies that $V(k,l) = 0$ if either $k > 0$ or $l > 0$, while each weight space is either $0$ or 1-dimensional. As a consequence, the commutativity of the diagram

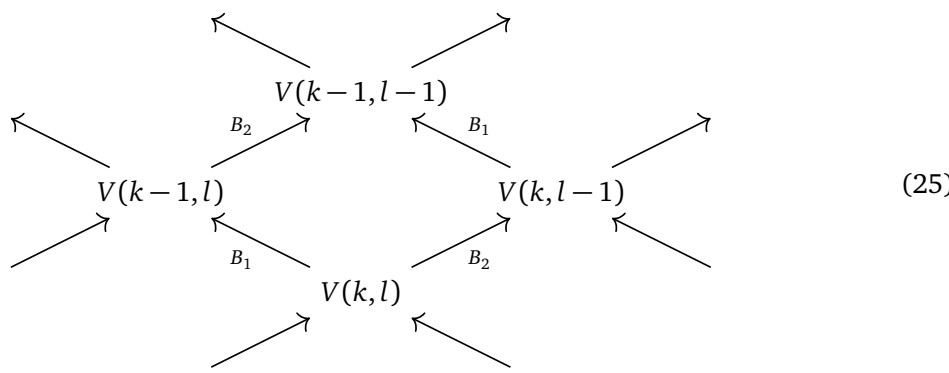

$$\tag{25}$$

which follows from the F-term relation $[B_1, B_2] = 0$, implies that $V(k,l)$ can be non-zero only if either $k = 0$ and $V(k,l+1) \cong \mathbb{C}$, $l = 0$ and $V(k+1,l) \cong \mathbb{C}$, or both $V(k+1,l) \cong \mathbb{C}$ and $V(k,l+1) \cong \mathbb{C}$. (In particular, situations like

$$\tag{26}$$

are excluded by these conditions). There being no other conditions, the identification of each pair $(-k,-l)$ with $V(k,l) \cong \mathbb{C}$ with a box at the corresponding location in the non-negative quadrant gives a bijective correspondence between fixed points of the torus action on $\mathrm{Hilb}^n(\mathbb{C}^2)$ and Young diagrams of weight $n$.

The main technical aim of this paper is to reproduce this "box stacking" description of torus fixed points from the mirror dual perspective of exponential networks. As alluded to in the introduction, this involves some rather non-trivial book-keeping, along with the resolution of certain puzzles witnessed in [7]. To get going, after a clarification of the relevant aspects of BPS state counting via exponential networks, we will end the next section with an illustration of the claimed identification for $n = 1, 2, 3, 4$, with quiver representations that follow. Section 4 will give the general proof, based on a tailor-made embellishment of the box-stacking story.

- For $n = 1$, there is just a single Young diagram with one box,

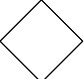

  corresponding to the representation with $V \cong \mathbb{C}$ and $B_1 = B_2 = 0$.

- For $n = 2$ we have Young diagrams for partitions

$$(2): \qquad\qquad (1,1): \qquad\qquad\qquad\qquad (27)$$

  corresponding to the following choices of $B_1, B_2$

$$(2): \quad B_1 = \begin{pmatrix} 0 & 0 \\ 1 & 0 \end{pmatrix}, \quad B_2 = \begin{pmatrix} 0 & 0 \\ 0 & 0 \end{pmatrix}, \quad I = \begin{pmatrix} 1 \\ 0 \end{pmatrix},$$

$$(1,1): \quad B_1 = \begin{pmatrix} 0 & 0 \\ 0 & 0 \end{pmatrix}, \quad B_2 = \begin{pmatrix} 0 & 0 \\ 1 & 0 \end{pmatrix}, \quad I = \begin{pmatrix} 1 \\ 0 \end{pmatrix}. \qquad (28)$$

- For $n = 3$, there are 3 distinct partitions: The Young diagram

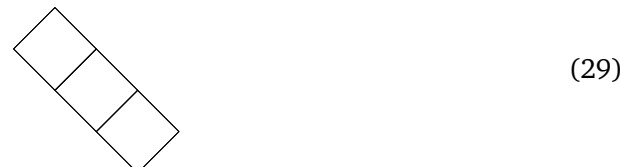

$$(29)$$

  for the partition $(3)$ corresponds to the matrices

$$B_1 = \begin{pmatrix} 0 & 0 & 0 \\ 1 & 0 & 0 \\ 0 & 1 & 0 \end{pmatrix}, \quad B_2 = \begin{pmatrix} 0 & 0 & 0 \\ 0 & 0 & 0 \\ 0 & 0 & 0 \end{pmatrix}, \quad I = \begin{pmatrix} 1 \\ 0 \\ 0 \end{pmatrix}. \qquad (30)$$

  The flipped diagram for the transposed partition $(1,1,1)$ can be obtained by exchanging $B_1$ and $B_2$. Lastly, the fixed point for the third possible Young diagram $(2,1)$,

$$(31)$$

  has both $B_1$ and $B_2$ non-zero:

$$B_1 = \begin{pmatrix} 0 & 0 & 0 \\ 1 & 0 & 0 \\ 0 & 0 & 0 \end{pmatrix}, \quad B_2 = \begin{pmatrix} 0 & 0 & 0 \\ 0 & 0 & 0 \\ 1 & 0 & 0 \end{pmatrix}, \quad I = \begin{pmatrix} 1 \\ 0 \\ 0 \end{pmatrix}. \qquad (32)$$

It is interesting at this point to record that the "non-representation" (26), to which (for $k = l = 0$) one would attach the box configuration

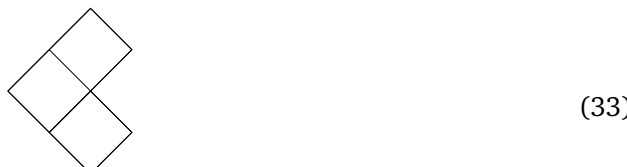

$$(33)$$

(which of course is not a Young diagram), with matrices

$$B_1 = \begin{pmatrix} 0 & 0 & 0 \\ 1 & 0 & 0 \\ 0 & 0 & 0 \end{pmatrix}, \quad B_2 = \begin{pmatrix} 0 & 0 & 0 \\ 0 & 0 & 0 \\ 0 & 1 & 0 \end{pmatrix}, \quad I = \begin{pmatrix} 1 \\ 0 \\ 0 \end{pmatrix}, \tag{34}$$

is still "$\theta$-stable" in the sense that the vector $I$ generates the full "non-module" under the action of $B_1$ and $B_2$. In other words, $(B_1, B_2, I)$ still satisfy the D-term constraint. However, while $B_1 B_2 = 0$, we have

$$B_2 B_1 = \begin{pmatrix} 0 & 0 & 0 \\ 0 & 0 & 0 \\ 1 & 0 & 0 \end{pmatrix},$$

and hence crucially $[B_1, B_2] \neq 0$, i.e., this violates the F-term constraints. This point of view will play a central role in our story: Using exponential networks, we will naturally first construct objects (finite webs) which satisfy the D-term constraint, and then check if they also satisfy the F-terms. This is a sensible thing to do also from the representation theoretic point of view: The quotient in (22) is taken on the set-theoretic intersection of solutions of the D- and F-terms inside $\mathcal{V}_n$ and there is a priori no preferred ordering.

Note that up to now, for all torus fixed points of $\text{Hilb}^n(\mathbb{C}^2)$ the F-term relation was satisfied trivially in the sense that $B_1 B_2 = B_2 B_1 = 0$. This is not generally true and the first counter example appears for $n = 4$ with the Young diagram

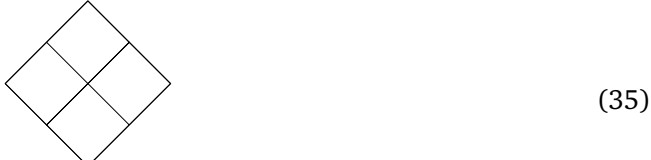

$$\tag{35}$$

and matrices

$$B_1 = \begin{pmatrix} 0 & 0 & 0 & 0 \\ 1 & 0 & 0 & 0 \\ 0 & 0 & 0 & 0 \\ 0 & 0 & 1 & 0 \end{pmatrix}, \quad B_2 = \begin{pmatrix} 0 & 0 & 0 & 0 \\ 0 & 0 & 0 & 0 \\ 1 & 0 & 0 & 0 \\ 0 & 1 & 0 & 0 \end{pmatrix}, \quad I = \begin{pmatrix} 1 \\ 0 \\ 0 \\ 0 \end{pmatrix}. \tag{36}$$

Indeed,

$$B_1 B_2 = \begin{pmatrix} 0 & 0 & 0 & 0 \\ 0 & 0 & 0 & 0 \\ 0 & 0 & 0 & 0 \\ 1 & 0 & 0 & 0 \end{pmatrix}, \quad B_2 B_1 = \begin{pmatrix} 0 & 0 & 0 & 0 \\ 0 & 0 & 0 & 0 \\ 0 & 0 & 0 & 0 \\ 1 & 0 & 0 & 0 \end{pmatrix}. \tag{37}$$

## 3 BPS counting with exponential networks

The quiver (4), which by the ADHM construction parameterizes self-dual $SU(N)$ connections on $\mathbb{R}^4$ with instanton number $k$, in string theory captures the interactions for bound state formation between $k$ D$p$- and $N$ D$(p+4)$-branes in flat space [32]. This becomes an instance of BPS state counting in type IIA string theory on a background of the form $\mathbb{R}^{1,3} \times X$, when $X = \mathbb{C}^3$ is the (simplest non-trivial!) toric Calabi-Yau 3-fold, $p = 0$, and the D4-branes are extended only in time drection in $\mathbb{R}^{1,3}$. Mathematically, this can be framed in terms of $\Pi$-stable objects in the derived category of coherent sheaves on $X$, see for instance [33].

Our interest in this paper is to reproduce the representations of the ADHM quiver with $N = 1$, $k = n$, and in particular the counting of BPS bound states via Young diagrams, using

exponential networks on the pair of pants (1). The initial idea [6] pursued in [7], was to study type IIB string theory on $\mathbb{R}^{1,3} \times Y$, where $Y$ is the mirror of $X$. In this picture, BPS states arise from D3-branes wrapping $\mathbb{R} \times L$, where $(L, \eta)$ is an A-brane consisting of a special Lagrangian $L$ equipped with a unitary local system $\eta$. By exploiting the conic fibration structure of local mirror Calabi-Yau, this reduces to a one-dimensional problem of calibrated geometry on the "mirror curve", which in the example is given by (1). The concrete proposal of [7] however relied on seemingly ad hoc junction rules, and was missing out on certain subtleties. The precise physical picture, explained in [9] in continuation of [1,13] is based on the identification of the mirror curve as the quantum moduli space of a codimension-2 defect in a 5-dimensional M-theory background [20, 21, 34], tracking 3d/5d BPS indices providing combinatorial data on the full exponential network, finally extracting the state count via Kontsevich-Soibelman wall-crossing [1, 2].

The purpose of this section is to reconcile these different points of view, in preparation of the matching between linear partitions and exponential networks that we present in section 4. In particular, we will give a rigorous justification of the junction rules proposed in [7] based on non-abelianization [8] and an account of the finite webs of [7] via critical saddle connections or foliations [11]. The upshot will be a direct identification of quiver representation data with non-abelianization data, and a prescription for (special) Lagrangian surgery directly on the mirror curve that takes holomorphic disks into account. All these phenomena will be illustrated for $n = 1, 2, 3, 4$ in more or less the order in which we discovered them.

## 3.1 Geometry of exponential networks

The definition of exponential networks relies on two kinds of data. The geometric data, in general, depends on a covering $\Sigma \to C$ of Riemann surfaces which is induced from an embedding $\Sigma \hookrightarrow (T^\vee C)^\times$ and which for each choice of phase $\vartheta \in \mathbb{R}/2\pi\mathbb{Z}$ determines a family of calibrations for one-dimensional submanifolds of $C$. The main examples, with $C = \mathbb{C}^\times$, arise from mirror curves, which we will use as starting point here. The combinatorial data, which we review in the next subsection 3.2, captures the soliton content of a certain effective 3-dimensional quantum field theory engineered from the geometric data.

To begin with, recall that given a toric Calabi-Yau threefold $X$, its mirror can be identified with a Calabi-Yau $Y$ of the form

$$Y = \{H(x, y) = uv\} \subset \mathbb{C}_x^\times \times \mathbb{C}_y^\times \times \mathbb{C}_u \times \mathbb{C}_v, \tag{38}$$

where $H(x, y)$ is a certain Laurent polynomial determined by the toric data and subject to the usual (framing) ambiguities, equipped with the holomorphic three-form

$$\Omega = \frac{dx}{x} \wedge \frac{dy}{y} \wedge \frac{du}{u}, \tag{39}$$

As shown in [6], and reviewed in [7], the study of a class of supersymmetic A-branes, special Lagrangian submanifolds in $Y$ fibered by 2-spheres or possibly pinched 2-tori, can be reduced to the "mirror curve"

$$\Sigma = \{H(x, y) = 0\} \subset \left(\mathbb{C}^\times\right)^2, \tag{40}$$

the locus where the conic bundle (38) degenerates, endowed with the multivalued one-form

$$\lambda = \log y \frac{dx}{x} = \log y \, d\log x, \tag{41}$$

which is the restriction of the Liouville one-form to $\Sigma$. More precisely, fixing a local trivialization of the intermediary covering

$$\pi : \Sigma \xrightarrow{x} \mathbb{C}_x^\times, \tag{42}$$

with branches labeled as $y_i$ for $i = 1, \ldots, K$, where $K$ is the degree of the covering, one is interested in real three-dimensional submanifolds of $Y$ that are fibered over trajectories in $\mathbb{C}_x^\times$ by $S^1$-fibrations over intervals with beginning and endpoint fixed to the $i$-th and $j$-th sheet, possibly winding (especially if $i = j$) some number $n$ times around the puncture in the fibers of $(T^\vee C)^\times$. These submanifolds are calibrated by $\mathrm{Re}\left(e^{-i\vartheta}\Omega\right)$ iff their image under $\pi$ is calibrated by

$$\lambda_{(ij)_n} = (\log y_j - \log y_i + 2\pi i n)\frac{dx}{x}, \tag{43}$$

i.e., it is a local integral curve of the ordinary differential equation

$$\left(\log y_j - \log y_i + 2\pi i n\right)\frac{d\log x(t)}{dt} \in e^{i\vartheta}\mathbb{R}_+, \tag{44}$$

parameterized by $t \in \mathbb{R}$. A trajectory that locally solves (44) will be labeled by $(ij)_n$, and the two points $y_j$, $y_i$ in the fiber will be given an orientation according to the sign with which they appear in (44). Orientation reversal replaces $(ij)_n$ with $(ji)_{-n}$. Note that the differential equation is non-trivial (though very simple) also when $i = j$ (as long as $n \neq 0$).

In computations, it is convenient to transition to the cover $\widetilde{\Sigma} \to \Sigma$ defined by $(x, \tilde{y}) \mapsto (x, \exp \tilde{y})$, on which the differential $\lambda$ in (41) is single valued. The composition $\tilde{\pi} : \widetilde{\Sigma} \to \Sigma \xrightarrow{\pi} C$ is then an infinite "logarithmic" cover with branches that can be locally labeled by pairs $(i, N) \in \{1, \ldots, K\} \times \mathbb{Z}$. Out of necessity, we will reserve the symbols $p \simeq (x, i, N)$ to denote a point in $\widetilde{\Sigma}$ above $x = \tilde{\pi}(p)$. It is important to emphasize that $\widetilde{\Sigma}$ is merely an auxiliary object to make $\lambda$ well-defined and all physical information is contained in $\Sigma$ and $\lambda$ only. In particular, any cycle in $\widetilde{\Sigma}$ representing a closed special Lagrangian in $Y$ has to be invariant under shift of the $\mathbb{Z}$-labels [8]. This means that it must be contained in the equivariant homology[2] $H_1^{\mathbb{Z}}(\widetilde{\Sigma}, \mathbb{Z}) \cong H_1(\widetilde{\Sigma}/\mathbb{Z}, \mathbb{Z}) = H_1(\Sigma, \mathbb{Z})$, see [35, 36]. For the purposes of the differential equation (44), we observe that the branch points of $\pi$ also lift to $\mathbb{Z}$-invariant families and support the only zeroes of $\lambda_{(ij)_n}$ for $n = 0$.

Given the logarithmic cover we can find local solutions to the ODE (44) starting from any pair of points which live in the fiber over some point $x$. On the other hand, the form of global solutions, which in particular should give rise to the precious compact special Lagrangians, depends on the identification of suitable boundary and junction conditions at points where branches with different labels meet. The idea of exponential networks, as introduced in [7, 8] following [1], is precisely to reduce the study of such boundary and stability conditions to a purely combinatorial algebraic problem.

Formally, an exponential network $\mathcal{W}(\vartheta)$ on $\mathbb{C}_x^\times$ is simply a web of local trajectories, called $\mathcal{E}$-walls, that are integral curves of the ordinary differential equation (44) labeled by all possible $(ij)_n$ (including $i = j$), and generated by the following local rules. First of all, trajectories called *primary walls* are allowed to begin (or end, with respect to their natural orientation coming from the parameterization of $x(t)$) at a branch point of the covering (42). While directly at a branch point of type $(ij)$, $\dot{x} = 0$ for the $(ij)_0$ trajectory, the non-Lipschitzness of the ODE means that there are (three) non-stationary solutions that start nearby and reach back to the branch point in finite time. Note that away from the branch points, the labels on primary walls can change across branch cuts for $\pi$ or the logarithm. Second, at an intersection of an $\mathcal{E}$-wall of type $(i_1 j_1)_{n_1}$ with an $\mathcal{E}$-wall of type $(i_2 j_2)_{n_2}$, we allow new, so-called *descendant walls* to start as follows:

- When $j_1 = i_2$, but $i_1 \neq j_1$, $j_2 \neq i_2$, and $j_2 \neq i_1$, a single $\mathcal{E}$-wall of type $(i_1 j_2)_{n_1+n_2}$. This is the elementary generalization of the corresponding rule for spectral networks [1].

---

[2]Since the equivariant homology $H_1^{\mathbb{Z}}(\widetilde{\Sigma}, \mathbb{Z})$ agrees with $H_1(\Sigma, \mathbb{Z})$ we will always use to the latter to refer to cycles in $\Sigma$ as well as to equivariant cycles in $\widetilde{\Sigma}$.

- When $j_2 = i_1$, but $i_1 \neq j_1$, $j_2 \neq i_2$ and $j_1 \neq i_2$ a single $\mathcal{E}$-wall of type $(i_2 j_1)_{n_1 + n_2}$. This follows from the first rule by exchanging 1 and 2.

- When $j_1 = i_2$ and $j_2 = i_1$, but $i_1 \neq j_1$, four infinite families of $\mathcal{E}$-walls, indexed by a positive integer $w$, as follows:

    - one $\mathcal{E}$-wall of type $(i_1 j_1)_{(w+1)n_1 + wn_2}$ for each $w = 1, 2, \dots$
    - one $\mathcal{E}$-wall of type $(i_1 i_1)_{wn_1 + wn_2}$ for each $w = 1, 2, \dots$
    - one $\mathcal{E}$-wall of type $(j_1 j_1)_{wn_1 + wn_2}$ for each $w = 1, 2, \dots$
    - one $\mathcal{E}$-wall of type $(j_1 i_1)_{wn_1 + (w+1)n_2}$ for each $w = 1, 2, \dots$

These rules were introduced in [7] as a consequence of "charge conservation"[3] based on assigning "multiplicities" $(w, w+1)$, $(w, w)$ or $(w, w+1)$ to the incoming trajectories, such that, for example, the junction

$$(w+1)(ij)_{n_1} + w(ji)_{n_2} \rightarrow (ij)_{(w+1)n_1 + wn_2}, \tag{46}$$

is associated to the identity of differentials

$$(w+1)\lambda_{(ij)_{n_1}} + w\lambda_{(ji)_{n_2}} = \lambda_{(ij)_{(w+1)n_1 + wn_2}}. \tag{47}$$

- When $i_1 = j_1 = i_2 \neq j_2$, an infinite family of $\mathcal{E}$-walls, indexed by a positive integer $w$, of type $(i_1 j_2)_{wn_1 + n_2}$. This rule has a similar explanation as the previous one.

- When $i_1 \neq j_1 = i_2 = j_2$, an infinite family of $\mathcal{E}$-walls, indexed by a positive integer $w$, of type $(i_1 j_2)_{n_1 + wn_2}$. Again, this follows from the previous one be exchanging indices 1 and 2.

From the point of view of the logarithmic cover, the rules can be interpreted geometrically as lifting the multiple open paths to successive logarithmic sheets over the junction and matching up their ends, respecting orientation to form a closed cycle. In other words, we can think of the junction rule (46) as a limiting case of the resolution depicted in 3. More algorithmically, the necessity of the infinite number of descendants follows from homotopy invariance of parallel transport, discussed in the next subsection [8]. The full fan of possible descendants at a junction of $(ij)_{n_1}$ and $(ji)_{n_2}$ is shown in Figure 4.

Note that for generic $\vartheta$, exponential networks (just like spectral networks) do not have any subwebs that would lift to finite length closed and calibrated cycles on the mirror curve (and hence to compact Lagrangians in $Y$). Moreover, even when such finite webs appear, it is *a priori* not clear to what extent they should count as true "BPS bound states", and if so, how it contributes to the BPS index. Indeed, in general such "critical subwebs" are merely special members of a whole family of calibrated cycles whose global structure is difficult to access (and even more so, to quantize). Before describing the 3d/5d combinatorial soliton data that allows addressing this in some generality, we anticipate some geometric features of the local toric mirror symmetry situation that will play an important supporting role in the example.

The point is that, say in the type IIA realization, the node labeled "$W$" in the (extended) ADHM quiver (4) (or (2)) corresponds to a non-compact D4-brane, and should therefore never lift to a finite-length trajectory on $\Sigma$ in the mirror picture. In [7], by studying the fate of the

---

[3]in other words, a version of Kirchhoff's rule stating that

$$\sum_a \lambda_a = 0, \tag{45}$$

where the sum extends over all $\mathcal{E}$-walls running into the junction, and which is hence manifestly crossing-invariant.

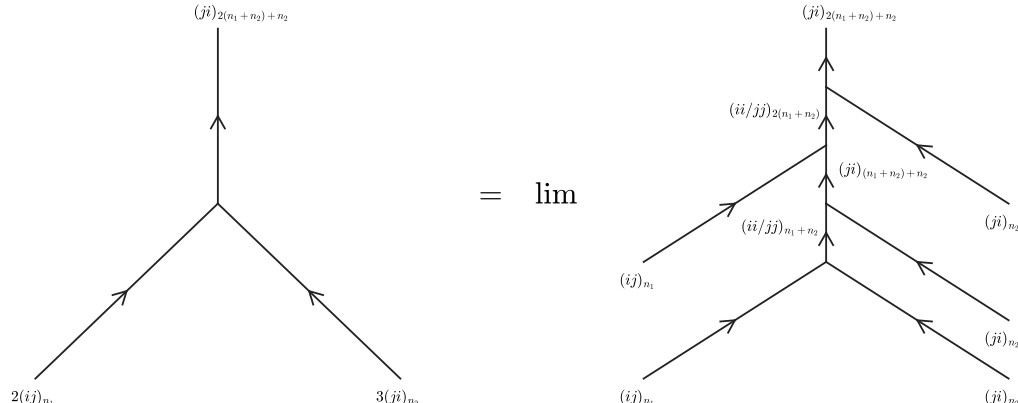

Figure 3: Resolution of a junction with multiplicities.

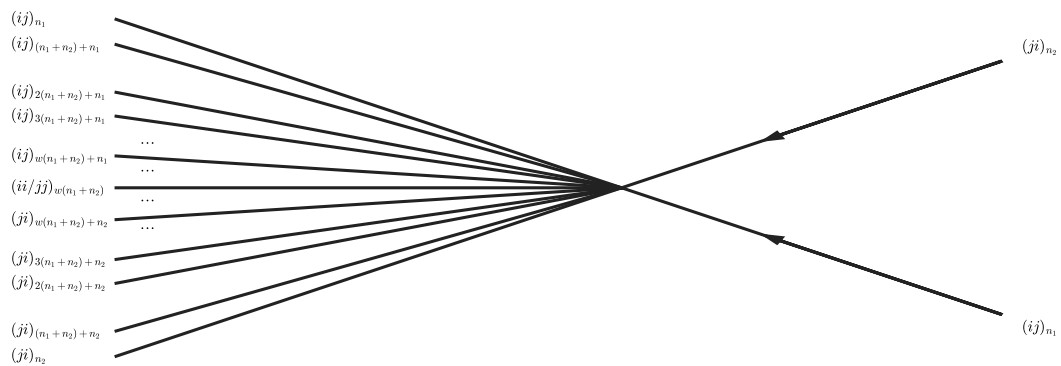

Figure 4: Descendant $\mathcal{E}$-walls emerging from a $(ij)_{n_1}$- $(ji)_{n_2}$ - type junction.

compact toric divisor of the local $\mathbb{P}^2$ geometry in the large volume limit, it was argued that the non-compact D4-brane can be regularized by introducing a physical cutoff near one of the punctures that in particular supplies a well-defined phase to its central charge. In the $\mathbb{C}_x^{\times}$-plane, this introduces a marked point that acts as an additional source for the exponential network to complement the local rules specified above.[4]

A completely physical way to effectively compactify the type IIA background with toric $X$ is to introduce a so-called $\Omega$-background [37–40] parameterized by $\epsilon_1, \epsilon_2, \epsilon_3 \in \left(\mathfrak{u}(1)^3 \subset \mathrm{Lie}((\mathbb{C}^{\times})^3)\right)^{\vee}$ with $\epsilon_1 + \epsilon_2 + \epsilon_3 = 0$. This deformation has the effect that the support of BPS branes inside $X$ must be preserved by some subtorus of $(\mathbb{C}^{\times})^3$, which in particular contributes to regulating the theory on the non-compact D4-brane, and which will localize the BPS states according to the discussion in subsection 2.2. The torus action on $X$, and in particular its degeneration along the toric diagram, also allows us to think of the Hori-Vafa [20] mirror $Y$ in terms of the moduli space of certain special toric (Aganagic-Vafa/Harvey-Lawson) Lagrangian A-branes $L \subset Y$ [21, 34], with a particular choice of framing. This gives a physical explanation of the seemingly ad hoc identification of toric curves and divisors with particular finite or regulated webs on $\mathbb{C}_x^{\times}$. The main drawback is that the orientation of the probe with respect to the $\Omega$-background in fact breaks the toric symmetries, and we indeed have to study the soliton combinatorics to extract the correct $(\mathbb{C}^{\times})^2$-invariant BPS count.

---

[4]In the hierarchy of branch points, such a point acts as a zeroth-order branch point.

## 3.2 3d/5d theory, non-abelianization, and BPS index

The microscopic picture underlying exponential networks and the physical interpretation of DT invariants in this framework was worked out in [8]. The procedure, which can be interpreted mathematically as an adaptation of non-abelianization advocated in [1], frames the problem of counting A-branes in terms of the combinatorics of certain kinky-vortices that we briefly recall now.

Consider M-theory on a background of the form $\mathbb{R}^{1,3} \times S^1 \times X$, where $X$ is a Calabi-Yau 3-fold, with an M5-brane that wraps a special Lagrangian submanifold $L \subset X$ and extends along $\mathbb{R}^{1,1} \times S^1 =: \mathbb{R}_\tau \times \mathbb{R}_\sigma \times S^1$, where $\tau$ is the timelike and $\sigma$ the spacelike coordinate of $\mathbb{R}^{1,1}$, and focus on the 3d/5d BPS states arising from M2-branes that wrap a curve $\mathcal{C} \subset X$ with boundary $\partial\mathcal{C}$ ending on a non-trivial cycle in $L$ and extend in the $\tau$-direction, possibly with momentum around the circle $S^1$ [41].

From the perspective of the 3d $\mathcal{N} = 2$ theory on $\mathbb{R}^{1,1} \times S^1$ engineered by the M5-brane, these M2-branes are co-dimension-2 BPS vortices, i.e. they are pointlike in the spatial directions [42]. From the string theory perspective, when the circle $S^1$ shrinks, the M5-brane descends to a D4-brane on $\mathbb{R}^{1,1} \times L$ with the BPS states being kinks on $\mathbb{R}^{1,1}$ [8]. These are field configurations that do not depend on the time direction $\mathbb{R}_\tau$, but evolve along the spatial direction $\mathbb{R}_\sigma$. Restoring the circle, one has the hybrid "kinky-vortices" on $\mathbb{R}^{1,1} \times S^1$ that interpolate between two possibly distinct vacua at $\sigma \to \pm\infty$. Each of these vacua corresponds to a BPS configuration of the M5-brane on $\mathbb{R}^{1,1} \times S^1 \times L$, so each of them is a point on the eventually quantum corrected moduli space of A-branes characterized by the geometric moduli of $L$ and complexified by the moduli of a $U(1)$ line bundle $\eta$. It is of interest to understand the degeneracy or supersymmetric indices of these 3d/5d BPS kinky-vortices.

Restricting $X$ to a toric Calabi-Yau and $L$ to a "Harvey-Lawson" Lagrangian with topology $\mathbb{R}^2 \times S^1$ leads to a one-dimensional moduli space that can be identified with the mirror curve $\Sigma \subset \mathbb{C}_x^\times \times \mathbb{C}_y^\times$ given in (40). From the point of view of the 3-dimensional theory, in which the $x$-coordinate parameterizes the classical moduli space of $L$ along a leg of the toric diagram, $\log x$ acts as Fayet-Iliopoulos parameter for a $U(1)$ vector multiplet whose scalar $\log y$ corresponds to the transverse position. Holomorphic disks ending on $L$ contribute to the superpotential, $W$, for $\log y$ such that its critical locus coincides with the mirror curve $H(x,y) = 0$. In particular, the branch of the logarithm corresponds to the holonomy of the gauge field at infinity in $\mathbb{R}^2 \subset L$. Accordingly, the kinky-vortices modeled by M2-branes ending in $S^1 \subset L$ can interpolate between two different solutions of $H(x,y) = 0$ for fixed $x$ and/or change the gauge field at infinity by one unit of flux.

In this picture, the $\mathcal{E}$-walls of an exponential network, solutions of (44), have the interpretation as trajectories of 3d/5d kinky-vortices mapping under variation of parameters to straight lines in the $W$-plane with constant phase $\vartheta$. Primary walls near branch points are Lefschetz thimbles over the vanishing cycles in the local singularity, and carry essentially a unique kinky-vortex each. Descendant walls emanate from junctions by wall-crossing according to a suitable Picard-Lefschetz formula and carrying a somewhat intricate kinky-vortex degeneracy as a consequence of "wall-crossing for 3d/5d BPS states" that was derived quantitatively in ref. [8], following [1].

For this purpose, one studies, in addition to the kinky-vortices, supersymmetric interfaces between 3-dimensional theories at different values of $x_1, x_2 \in C$, together with their lifts to $\Sigma$, and $\widetilde{\Sigma}$, under the constraints that (i) these interfaces be topological, in the sense that they depend only on the homotopy class of a path $\wp$ taken between $x_1$ and $x_2$ on $C$, and (ii) their vacuum degeneracy satisfy its own "wall-crossing for **framed** 3d/5d states" by bound state formation with kinky-vortices when the path intersects an $\mathcal{E}$-wall.

Concretely, the fact that when the $S^1$ shrinks the 3d/5d BPS states reduce to kinks implies

that the map

$$X_a : \mathcal{L}_{p_1} \to \mathcal{L}_{p_2} , \tag{48}$$

between the vacua of the massive Landau-Ginzburg model with superpotential $W(\ln x, \ln y)$ labeled by $p_1, p_2 \in \widetilde{\Sigma}$ defined by a supersymmetric interface corresponding to a path $a$ in $\widetilde{\Sigma}$ coincides with parallel transport by means of the flat connection $\nabla^{ab}$ in the line bundle $\mathcal{L} \to \widetilde{\Sigma}$ of vacua provided by the $tt^*$-connection. Note that even though massive models are anomalous, they still have sensible $tt^*$-connections [43]. The connection $\nabla^{ab}$ being flat implies that parallel transport only depends on the homotopy class of the path relative to its endpoints. In fact, because $\mathcal{L}$ has rank one (namely, $\nabla^{ab}$ is an "abelian flat connection"[5]), it is enough (by slight abuse of notation) to remember only the relative homology class of $a \in H_1(\widetilde{\Sigma}, p_1, p_2)$. Quite obviously, these parallel transport variables satisfy the algebra

$$X_a X_b = \begin{cases} X_{ab}, & \mathrm{end}(a) = \mathrm{beg}(b), \\ 0, & \mathrm{otherwise}, \end{cases} \tag{49}$$

where endpoints of paths/homology classes $a$ and $b$ must match on $\widetilde{\Sigma}$. Also note that $\nabla^{ab}$ is shift-symmetric with respect to the covering $\widetilde{\Sigma} \to \Sigma$.

Given $(\mathcal{L}, \nabla^{ab})$ and its physical interpretation, the supersymmetric interfaces attached to open paths $\wp \subset C$ from $x_1$ to $x_2$ can be viewed as defining, in dependence on the kinky-vortex data on the exponential network $\mathcal{W}(\vartheta)$, and for generic $\vartheta$, parallel transport

$$F(\wp, \vartheta) : E_{x_1} \to E_{x_2} , \tag{50}$$

in the infinite rank vacuum bundle $E \to C$ that is obtained by pushing down $\mathcal{L}$ along $\tilde{\pi}$. More precisely, the push-down is at first well-defined only outside the ramification locus, and is then modified across the exponential network by re-gluing according to (50). Demanding homotopy invariance is equivalent to the flatness of an associated "non-abelian connection", $\nabla^{na}$, which allows continuation of $E$ back to all of $C$. In this way, exponential networks just define a generalization of GMN non-abelianization.[6]

Concretely, the fiber of $E$ over any $x \in C \setminus \mathcal{W}(\vartheta)$ is defined by

$$E_x = \bigoplus_{p = (x, i, N) \in \tilde{\pi}^{-1}(x) \subset \widetilde{\Sigma}} \mathcal{L}_p . \tag{51}$$

For a path $\wp$ in $C$ that does not intersect the exponential network, parallel transport in $E$ is defined as the direct sum of parallel transport along relative homology classes $a$ above $\wp$ in all sheets of the covering, connecting the preimages of the endpoints of $\wp$ respectively. Namely, letting $U \subset C$ be a connected open set containing $\wp$ disjoint from the ramification locus, and fixing a local trivialization

$$\tilde{\pi}^{-1}(U) = \bigcup_{(i,N)} U^{(i,N)} , \tag{52}$$

we decompose

$$\tilde{\pi}^{-1}(\wp) = \bigcup_{(i,N)} \wp^{(i,N)} , \tag{53}$$

with $\wp^{(i,N)} \subset U^{(i,N)}$, and extend (51) as a trivialization of $E$ to all of $U$. Then, with respect to this decomposition we can write the diagonal connection

$$F(\wp, \vartheta) = D(\wp) := \bigoplus_{(i,N)} X_{\wp^{(i,N)}} , \tag{54}$$

---

[5]More precisely, we should work with twisted flat connections as in [1], but we will sheepishly ignore this pesky detail.

[6]In [1], for the spectral networks associated to Hitchin systems, the construction of higher-rank flat bundles from the data of a covering flat line bundle was introduced. Our case differs crucially because $\Sigma$ now is not a spectral curve of a Hitchin system, rather it is a moduli space of the Lagrangian defects engineered by M5-brane.

which is independent of $\vartheta$ as indicated as long as the exponential network remains disjoint from the fixed $\wp$.

The parallel transport along a path $\wp$ in $C$ that does cross an $\mathcal{E}$-wall in the exponential network gets corrected by the 3d/5d kinky-vortices running along the lift of the $\mathcal{E}$-wall in $\widetilde{\Sigma}$. Suppose $\wp$ crosses an $\mathcal{E}$-wall of type $(ij)_n$ at the point $x \in C$. Then locally around $x$ the $\mathcal{E}$-wall lifts to sheets $(i, N)$ and, with opposite orientation, to sheets $(j, N + n)$, for all possible $N$, and by construction of the exponential network we know that these points are connected by some calibrated trajectories or in other words, there exist some 3d/5d kinky-vortices interpolating the corresponding vacua. The associated relative homology classes provide "detours" for the direct connection recorded in (54).

Here, the crucial quantity is the degeneracy $\mu(a)$ of kinky-vortices interpolating vacua $(x, i, N)$ and $(x, j, N + n)$ along the relative homology class $a \in H_1(\widetilde{\Sigma}, (x, i, N); (x, j, N + n))$, where $x \in C$ is that point fixed on an $\mathcal{E}$-wall of type $(ij)_n$.[7] The combinatorics of exponential networks forces the $\mu(a)$ to be rational numbers in general, which is different from the spectral networks where the $\mu(a)$ are always integers. In terms of the junction rules on page 14, the underlying reason is that concurrent $\mathcal{E}$-walls of type $(ii)_n$ and $(ii)_m$ are physically indistinguishable from an $\mathcal{E}$-wall of type $(ii)_{n+m}$. Denoting by

$$\Gamma_{(ij)_n}(x) = \left( \prod_N H_1\big(\widetilde{\Sigma}; (x, i, N), (x, j, N + n)\big) \right)^{\mathbb{Z}}, \tag{55}$$

the shift-invariant version of these relative homology groups that could possibly support kinky-vortices, the precise form of the off-diagonal correction term advocated in [8] is

$$\Xi_{(ij)_n}(x) = \sum_{a \in \Gamma_{(ij)_n}(x)} \mu(a) X_a. \tag{56}$$

Namely, noting that the point $x$ splits the path $\wp$ into two pieces $\wp_0$ and $\wp_1$, the full parallel transport formula is

$$F(\wp, \vartheta) = D(\wp_0) e^{\Xi_{(ij)_n}(x)} D(\wp_1), \tag{57}$$

where the RHS depends on $\vartheta$ through the topology of the exponential network $\mathcal{W}(\vartheta)$ and its intersections with $\wp$. The crucial point, as shown in [8], is that the kinky-vortex degeneracies $\mu(a)$ can be determined uniquely by demanding that the parallel transport isomorphisms defined by (54) and (57), together with, obviously, $F(\wp \wp', \vartheta) = F(\wp, \vartheta) F(\wp', \vartheta)$ when $\wp$ and $\wp'$ match up, are homotopy invariant for any path $\wp \subset C$. See [8] for details.

The information about the "vanilla" 5d BPS states, which are our main interest in this paper, is contained in the dependence of the non-abelianization map on $\vartheta$. Specifically, at a "critical" angle $\vartheta_c = \arg Z_\gamma$ equal to the argument of the period[8] or "central charge" $Z_\gamma$ of the calibrating differential (41) around a homology class $\gamma \in H_1(\Sigma, \mathbb{Z})$, the network might degenerate by forming so-called "double walls" (or "two-way streets") in the sense that $\mathcal{W}(\vartheta_c)$ contains $\mathcal{E}$-walls sitting on top of each other with opposite orientation and thereby signal the existence of a calibrated cycle built from the various lifts of the $\mathcal{E}$-walls that meet up above the double walls.

Let us denote by $\mathcal{W}_c$ the collection of double walls, thought of as a "subweb" inside $\mathcal{W}(\vartheta_c)$, and assume for simplicity that the subset of $H_1(\Sigma, \mathbb{Z})$ containing classes $\gamma$ with $e^{-i\vartheta_c} Z_\gamma \in \mathbb{R}_+$

---

[7]The fact that $\mu$ depends only on the relative homology class (and not on the calibrated trajectory as such), follows from standard considerations of Picard-Lefschetz theory. It is important to note, however, that the kinky-vortex degeneracy is only defined for relative homology classes $a$ with $\tilde{\pi}(\partial a) = 0$, i.e., when the endpoints lie over the same point in $C$. The renunciation of a dedicated symbol at this point allows more uniform calculations within the fixed algebra (49).

[8]While the differential $\lambda$ itself is not well-defined on $\Sigma$, the pairing with shift-invariant homology classes in $\widetilde{\Sigma}$ is, see page 20 for further discussion.

is generated by a single element $\gamma_c$.[9] Then, to define a BPS count in the class $k\gamma_c$ for each $k = 1, 2, \ldots$, we wish to use the kinky-vortex degeneracy to assign integer coefficients to the various local preimages of $\mathcal{W}_c$ under $\tilde{\pi}$ such that these can be assembled to closed cycles $\mathbf{L}_k$ that are calibrated at $\vartheta_c$, and whose homology classes are multiples of $k\gamma_c$ that can be identified with the sought-after invariant.

Algebraically, the idea is that while the non-abelianization map itself is not well-defined at $\vartheta_c$, and in general the non-abelian flat connection $\nabla^{na}$ will change discontinuously when $\vartheta$ is varied from a slightly smaller value $\vartheta_-$ to a slightly larger value $\vartheta_+$, the change can be captured by a wall-crossing formula of the form [1]

$$F(\wp, \vartheta_+) = \mathcal{K}\big(F(\wp, \vartheta_-)\big), \tag{58}$$

valid for every $\wp$, where the Kontsevich-Soibelman wall-crossing operator $\mathcal{K}$ is a universal substitution of the parallel transport algebra (49) that compensates for the "change in detour" induced by the $\mathbf{L}_k$, much like the detour formula (57) in framed 3d/5d wall-crossing.

To describe the local data, say $x \in \mathcal{W}_c$ with an $\mathcal{E}$-wall of type $(ij)_n$ coming in from the left and supporting kinky-vortices on various $a \in \Gamma_{(ij)_n}(x)$, and one of type $(ji)_{-n}$ coming in from the right with kinky-vortices on various $b \in \Gamma_{(ji)_{-n}}(x)$. While the multiplicities $\mu(a)$ and $\mu(b)$ are constant between $\vartheta_-$ and $\vartheta_+$, parallel transport along a short path $\wp$ crossing the network at $x$ will generically jump because of the change in detour ordering (first $b$, then $a$ for $\vartheta_-$; first $a$, then $b$ for $\vartheta_+$), as reflected in the mismatch between the two expressions

$$\begin{aligned}
F(\wp, \vartheta_-) &= D(\wp_0) e^{\Xi_{(ji)_{-n}}(x)} e^{\Xi_{(ij)_n}(x)} D(\wp_1), \\
F(\wp, \vartheta_+) &= D(\wp_0) e^{\Xi_{(ij)_n(x)}} e^{\Xi_{(ji)_{-n}}(x)} D(\wp_1),
\end{aligned} \tag{59}$$

where $\wp_0$, $\wp_1$ are suitably interpreted. A moment's thought shows that this can be brought into the form of (58) with the help of the new generating function [1,8]

$$Q(x) = 1 + \sum_{a,b} \mu(a)\mu(b) X_{ab}, \tag{60}$$

that depends only on parallel transport variables $X_{ab}$ along closed cycles in $\widetilde{\Sigma}$, i.e., beginning and ending at the same point above $x$. Namely, the non-trivial action of the wall-crossing operator is

$$\begin{aligned}
\mathcal{K}(X_{\wp^{(i,N)}}) &= X_{\wp^{(i,N)}} Q(x), \\
\mathcal{K}(X_{\wp^{(j,N)}}) &= X_{\wp^{(j,N)}} Q(x)^{-1},
\end{aligned} \tag{61}$$

for fixed $i$ and $j$, and varying $N$, and where the split of $\wp^{(i,N)}$ at $(x, i, N)$ and $\wp^{(j,N)}$ at $(x, j, N)$ is left implicit, while the sign in the exponent accounts for the relative orientation.

Under the above assumption, the concatenated cycles $ab$ appearing in (60) all represent multiples of the critical homology class $\gamma_c$. This implies that, forgetting the marked point above $x$ if desired, $Q(x)$ is in fact a Laurent series in the single variable $X_{\gamma_c}$, and can be factorized

$$Q(x) = \prod_{k=1}^{\infty} (1 + X_{k\gamma_c})^{\alpha_{k\gamma_c}(x)}, \quad \text{with } \alpha_{k\gamma_c}(x) \in \mathbb{Z}. \tag{62}$$

---

[9]In [1], this assumption is not very restrictive, since it automatically holds as long as one stays away from walls of marginal stability on the Coulomb branch, which is a codimension 1 subspace. In our case, the geometry of $\Sigma$ is dictated by the toric Calabi-Yau $X$, and we might not have the luxury to freely perturb our system away from a wall of marginal stability if we happen to find ourselves sitting on one. However, in the example studied in this paper, we are lucky and this assumption holds.

Observing in passing that the local coefficients $\alpha_{k\gamma_c}(x)$ do not vary along the double walls, the closed cycle[10]

$$\mathbf{L}_k = \bigcup_{x \in \mathcal{W}_c} \alpha_{k\gamma_c}(x)\tilde{\pi}^{-1}(x), \tag{63}$$

has the property that for any short path $a$ in $\widetilde{\Sigma}$ intersecting $\mathbf{L}_k$ in a point over $x \in C$, the intersection number $\langle \mathbf{L}_k, a \rangle$ agrees with $\pm\alpha_{k\gamma_c}(x)$ when $a$ is on the $i$-th or $j$-th sheet, respectively, and zero otherwise. As a consequence, we can elegantly rewrite the wall-crossing formula (61) as

$$\mathcal{K}(X_a) = X_a \prod_{k=1}^{\infty} (1 + X_{k\gamma_c})^{\langle \mathbf{L}_k, a \rangle}, \tag{64}$$

harmoniously for all $a$. Finally, the 5d BPS invariant of charge $k\gamma_c \in H_1(\Sigma, \mathbb{Z})$ is defined by

$$\Omega(k\gamma_c) = \frac{[\mathbf{L}_k]}{k\gamma_c} \in \mathbb{Z}. \tag{65}$$

**A remark on multiplicities** As explained around eq. (46), the junction rules for exponential networks can be understood in terms of charge conservation after attaching multiplicities $w$ to $\mathcal{E}$-walls that were used in [7] to construct compact cycles from finite webs directly at the critical angles $\vartheta_c$. These multiplicities are similar to the coefficients $\alpha_{k\gamma_c}(x)$ appearing in (63), and in fact in our main example they will be equal (up to a sign). Whether or not this can be generalized, it is important to keep in mind that the multiplicities only have meaning for finite webs. They do not exist for networks at generic phase $\vartheta \neq \vartheta_c$ and thus cannot appear when computing parallel transport of flat connections.

**A remark on lifts and central charges** There is a subtlety in the above discussion related to the calculation of the "central charge"

$$Z_\gamma = \int_\gamma \lambda. \tag{66}$$

While working with $\widetilde{\Sigma}$ has the indispensable advantage that that we can solve eq. (44) for the BPS trajectories honestly because the differential $\lambda$ is actually well defined, physical charges being attached to cycles in $\Sigma$ requires us to involve equivariant cycles in $\widetilde{\Sigma}$ as we have seen above. Taken at face value, this equivariance implies that the expression (66) will generally just be infinite and not a good definition of the central charge. This problem has a simply remedy in those cases in which the cycle $\gamma$ splits into a disjoint union

$$\gamma = \bigcup_N \gamma_N, \tag{67}$$

of isomorphic closed cycles $\gamma_N$ such that we can define

$$Z_\gamma = \int_{\gamma_N} \lambda, \tag{68}$$

independent of the choice of $N$. While in general, there might not be any section $\gamma_N \hookrightarrow \gamma$ realizing a splitting (67) into closed cycles, if we remove the logarithmic cut, the covering

---

[10]Note that $(ii)$-type double walls are not naively captured by the operator $\mathcal{K}$ exactly as described. This is because $(ii)$-type $\mathcal{E}$-walls lift to trivial shift-invariant chains in $\widetilde{\Sigma}$, while the kinky-vortex degeneracies are in general not integral. However, consistency of the lift along with central charge of the BPS state determine the lift of this $ii$-type wall. We will see examples of this below.

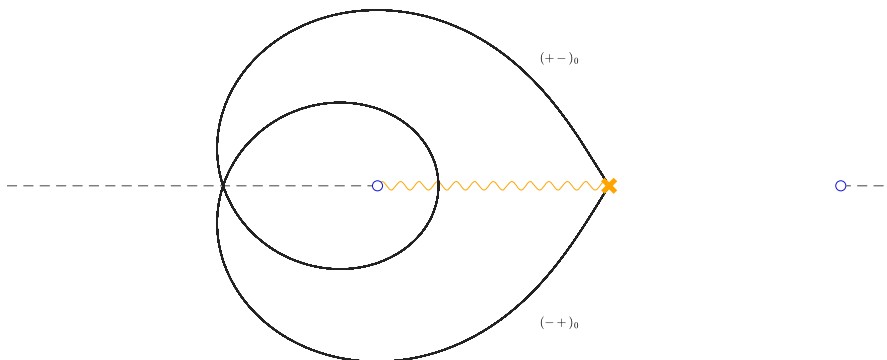

Figure 5: Piece of $\mathcal{W}_c$ at $\vartheta_c = 0$ representing the D0-brane at the origin in $\mathbb{C}^3$.

map decomposes into a disjoint family of copies of $\Sigma \setminus \{$ logcut $\}$, which ensures that we can find a section $\gamma_N \hookrightarrow \gamma$ respecting the calibration(!), and continuous away from the logcut.[11] If $\gamma$ had no section, the closure of $\gamma_N$ in $\widetilde{\Sigma}$ will not be a closed cycle. However, we can still integrate $\lambda$ over it and define the central charge $Z_\gamma$ using (68).

## 3.3 Finite webs on the three-punctured sphere and torus fixed points

Girt with the extraction of BPS invariants from wall-crossing of the non-abelianization map, we now return to the exponential networks on the mirror curve for $\mathbb{C}^3$, which is the three-punctured sphere defined in the presentation $\mathbb{C}_x^\times \times \mathbb{C}_y^\times \supset \Sigma \xrightarrow{x} \mathbb{C}_x^\times$ by the equation

$$x - y - y^2 = 0, \tag{69}$$

see (1). This being a two-fold cover, we let the sheet labels $i$ and $j$ take values "+" and "−". They collide at the single branch point above $x = -1/4$ that sources a single finite web at the critical angle $\vartheta = 0$, as depicted in Fig. 5.[12] As shown in [7], this finite web lifts on $\Sigma$ to the "Seidel Lagrangian" that circles around the three punctures. Moreover, based on arguments similar to those expounded here, involving the three self-intersections, the intersection with the D4-brane, which we review momentarily, and a heuristic investigation of its moduli space, it was argued in [7] that this finite web should be identified with a D0-brane at the origin of $\mathbb{C}^3$. This identification was justified rigorously in [8] by a careful study of the infinite tower of descendent walls that originate at the self-intersection, and the wall-crossing of the non-abelianization map. We will accordingly label the homology class of the Seidel Lagrangian by $\gamma_{D0}$. In the normalization (41), the associated central charge computed by eq. (66) is $Z_{D0} := Z_{\gamma_{D0}} = 4\pi^2$, and the index calculated in [8] is $\Omega(\gamma_{D0}) := \Omega(D0) = -1$.

As anticipated on page 14, in order to get a more interesting spectrum of exponential networks, we need to take into account also the BPS trajectory emanating from the branch point to the right in the $w$-plane and headed toward to puncture at $x = 0$, see Fig. 6. As observed in [7], the length of this trajectory diverges as $t \sim \frac{1}{2}|(\log x)^2|$ on approach of the singularity, and should hence be identified with the non-compact D4-brane wrapping the toric divisor that is dual to the base curve $C = \mathbb{C}_x^\times$, viewed as classical moduli space of the chosen

---

[11]Removing the logcut typically disconnects the cycle and each piece has a $\mathbb{Z}$-family of possible lifts. Crucially the integral defining the central charge is piecewise independent of the lift and hence the central charge is well-defined.

[12]Throughout the paper, we adopt the convention of [7] and plot in a variable $w = \frac{x}{1/4-x}$ such that the puncture at $x = \infty$ lies at finite distance at $w = -1$, which is formally the second branch point of the two-fold covering. The dashed line is the logarithmic cut.

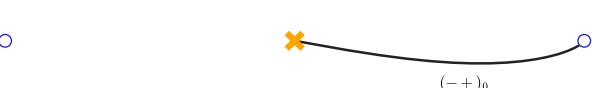

Figure 6: Piece of $\mathcal{W}_c$ at $\vartheta_c = \vartheta_{\mathrm{D4}} < 0$ representing a regularized D4-brane on a toric divisor.

Harvey-Lawson Lagrangian (see beginning of sect. 3.2). More precisely, such a non-compact trajectory exists for any value of $\vartheta$, and after regularizing the divergence at some large value $t_{\max}$, we can rigidify the situation by requiring the trajectory to end at a specific point $x_0$ with $|x_0| \sim e^{\sqrt{2t_{\max}}}$ near the puncture. This selects a unique trajectory as "regularized D4-brane" out of the family, with a fixed phase $\vartheta_{\mathrm{D4}}$ that can identified with the total B-field $(B_{12}+B_{34})$ piercing the toric divisor. While the full D4 can be recovered by letting $x_0 \to 0$ along $e^{-e^{i\vartheta_{\mathrm{D4}}/2}\sqrt{2t_{\max}}}$ with $t_{\max} \to \infty$, the regularized picture fits better into the framework of 3d/5d BPS state counting in terms of exponential networks,[13] if we admit $x_0$ as an additional source of an $\mathcal{E}$-wall of type $(-+)_n$, for some preferred $n$ that determines how often the regularized trajectory winds around the origin before reaching $x_0$. Namely, with $n = 0$ as most natural choice, we will recover the regularized trajectory as unique "finite web" at the critical phase $\vartheta_c = \vartheta_{\mathrm{D4}}$. In terms of the discussion around (65), the charge of the D4-brane is a relative homology class $\gamma_{\mathrm{D4}}$ on $\Sigma$ ending above $x_0$, while its central charge calculated by the formula (66) is $Z_{\mathrm{D4}} = e^{i\vartheta_{\mathrm{D4}}} t_{\max}$. Since across $\vartheta_{\mathrm{D4}}$ the exponential network witnesses a single ordinary double wall, the BPS index defined in (65) will be simply $\Omega(\mathrm{D4}) = 1$. The geometric state of affairs at $\vartheta_{\mathrm{D4}}$ is depicted in Fig. 6.

As is evident from the pictures, the cycles corresponding to the D0- and D4-branes intersect above the ramification point once with each orientation. Together with the three self-intersections of the D0-brane this can be encoded in the "extended ADHM quiver" (2), with the superpotential (3) arising from two holomorphic disks bounding their lift to $\Sigma$ as shown in figure 7, and we can set out to identify the moduli of quiver representations, and especially $\mathrm{Hilb}^n(\mathbb{C}^2)$ for $n = 1, 2, 3, \ldots$, in our exponential networks.

To reiterate, describing the full moduli space purely in terms of A-branes is a rather challenging problem, on which we offer our best attempt in section 5, and we shall here be content to recover just the torus fixed points described in section 2.2. Specifically, we will show that the BPS index defined by (65) for finite webs of charge corresponding to one D4 and $n$ D0-branes is equal to the Euler characteristic of $\mathrm{Hilb}^n(\mathbb{C}^2)$, by exhibiting a specific finite and "indecomposable" web of central charge

$$Z_{(1,n)} = |Z_{\mathrm{D4}}| e^{i\vartheta_{\mathrm{D4}}} + n Z_{\mathrm{D0}}, \tag{70}$$

for each linear partition of weight $n$. Note that for finite $Z_{\mathrm{D4}}$, and $\vartheta_{\mathrm{D4}} \neq 0$, the central charges for different $n$ are different and the finite webs of interest appear at distinct angles

$$\vartheta_c = \vartheta_{(1,n)} := \arg Z_{(1,n)}. \tag{71}$$

---

[13]From the point of view of the 5d theory, the D4 brane provides framing for the particles engineered by D0 branes, and similarly, the source at the cutoff point provides a framing for the exponential networks.

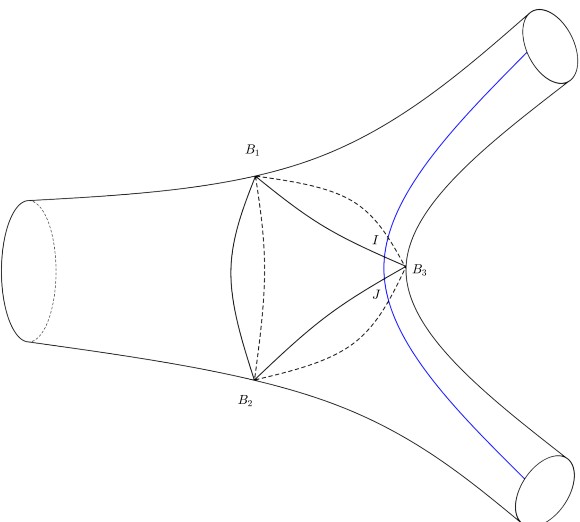

Figure 7: Cycles representing a single D0 with a D4-brane, with intersections named corresponding to the arrows in (2).

Along the way of discovery, this discrimination provided us with a strong numerical check that our identification was on the correct track, if the central charge obtained by integration over some candidate finite web was indeed $Z_{(1,n)}$. By moving the cutoff $x_0 \to 0$ as explained above, we could in principle send $|Z_{D4}| \to \infty$ so that $\vartheta_{(1,n)} \to \vartheta_{D4}$ for all $n$. This would, however, take away the numerical check (and make the pictures less pretty), so we choose to present the results with a finite cutoff (and insist on always speaking of "finite webs" as representing BPS states).

The decomposition of the finite web at critical $\vartheta_c = \vartheta_{(1,n)}$ into torus fixed points can be understood in two complementary ways. From the point of view of subsection 3.2, to compute the 5d index, we need to determine the cycle $\mathbf{L}_k$ in (63) representing a multiple of $k\gamma_c$ for each $k = 1, 2, \ldots$. In our situation involving D0-D4 bound states, based on the fact that the primitive homology class $\gamma_c = \gamma_{D4} + n\gamma_{D0}$, and after regularization, the coefficients $\alpha_{k\gamma_c}(x)$ in (63) are non-zero only when $k = 1$, and can be identified with the multiplicities $w$ appearing in the junction rules (46). The compatibility of these rules (with multiplicities included) with the parallel transport algebra (49) then ensures that if at each step we pick a particular descendant from the infinite fan in Fig. 4 in a manner consistent with the resolution of Fig. 3[14], the resulting concatenation above the $\mathcal{E}$-walls will indeed define a closed cycle on $\Sigma$ by formula (63). This allows us to build individual components of $\mathbf{L}_1$ directly, bypassing the rather tedious formalism explained above, and ultimately justifying the approach of [7]. The kinky-vortex multiplicities of course are crucial in this justification by ensuring the homotopy independence of the parallel transport function $F(\wp, \vartheta)$.

Alternatively, and mirroring the build-up of Young diagrams via "box stacking", as sketched for small $n$ in section (2.2), we can construct indecomposable finite webs of given $n$ as bound states of smaller constituents, ultimately reducing everything to one D4 and $n$ D0-branes, viewed as basis of the corresponding ADHM module. Namely, locally around each "four-way intersection" of two such constituents, we can "flip the right of way" on the upstairs strands to which the $\mathcal{E}$-walls lift on the covering curve $\Sigma$. Depending on the type of bound state we

---

[14]Note that even with fixed multiplicities, the outgoing wall can be formed in more than one way. As we shall see, this fact plays an important role in the counting.

intend to form, this will manifest in the insertion of additional detours in the finite web on $C$, which after continuous deformations will eventually be calibrated at the correct angle $\vartheta_{(1,n)}$ of the full bound state. Physically, this procedure can be viewed as "condensation of open string tachyons" that are localized at the intersection points. Mathematically, reconnecting the strands is a one-dimensional projection of what is known as "oriented Lagrangian surgery" of the special Lagrangians to which the finite webs lift inside the Calabi-Yau threefold $Y$.[15] At the end, the connected sum of all these finite webs is identified with the critical part of the exponential network at the phase $\vartheta_{(1,n)}$.[16]

To claim the correspondence between ADHM representations and finite webs for each $n$, we will explain how to translate the representation data at a torus fixed point, i.e., commuting matrices $B_1$ and $B_2$ satisfying (23), alternatively encoded into a Young diagram, into the combinatorial structure of a finite web of $\mathcal{E}$-walls, and, vice-versa, how to read off the matrices from the finite web. In this translation, we will think of non-zero matrix entries as "surgery parameters" that describe how the finite web was built from the elementary constituents (D4- and D0-branes, see Figs. 5 and 6). This identification is the basic bridge between fixed points derived from the quiver description and the exponential networks. Similar observations were made in a somewhat different context in [26], based on general grounds explained in [11].

In the remainder of this section, we will explain how these different points of view combine to produce all required finite webs for small values of $n$, paying particular attention to the crucial facts that

(i) each set of surgery parameters determines a unique finite web, and

(ii) some finite webs do not correspond to torus fixed points, because the lifted cycles bound non-canceling holomorphic disks. This happens precisely when the matrices capturing the surgeries do not satisfy the F-terms derived from the superpotential (3). Among these is the ADHM relation (5), $[B_1, B_2] = 0$ when $IJ = 0$, and the condition that there be no surgery between different D0-branes at the branch point (i.e., $B_3 = 0$, such that (2) indeed reduces to the ADHM quiver (4)).

In section 4, we will describe how this correspondence can be extended to work for all $n$.

## 3.4 Bound states from surgery, and holomorphic disks

The following progression is meant to parallel the discussion on page 10. The basic idea is to identify each box in a Young diagram with a specific D0-brane that partakes in a bound state, in order to understand the build-up of the finite webs.

$n = 1$  At the angle $\vartheta_{(1,1)}$ we find a single finite web, shown in Fig. 8. It is easy to confirm numerically that this web forms as a double wall, with one wall sourced at the branch point, and the other at the cutoff near the puncture. In other words, this represents a 5d BPS state with $\Omega(D4 + D0) = 1$, as can be confirmed from the wallcrossing shown in Fig. 9. This state of course is in correspondence with the unique torus fixed point of $\text{Hilb}^1(\mathbb{C}^2)$, the ADHM module

---

[15]Note that one can define the oriented Lagrangian surgery of Joyce [44] through Lagrangian connected sum. Given two special Lagrangian submanifolds $L_1$ and $L_2$, $L_1 \# L_2$ is constructed by gluing in a fixed local model in Darboux charts around the intersection points. The class of $L_1 \# L_2$ is uniquely fixed. Although topologically it is the usual connected sum, symplectically, the sum depends on the orientations and hence one has the graded connected sum. Given a choice of the complex structure, for fixed Kähler form, the orientations determine whether Lagrangian surgery produces stable bound states.

[16]Similar considerations (with multiplicities in fact) appear in a different situation in appendix B.2 in [45].

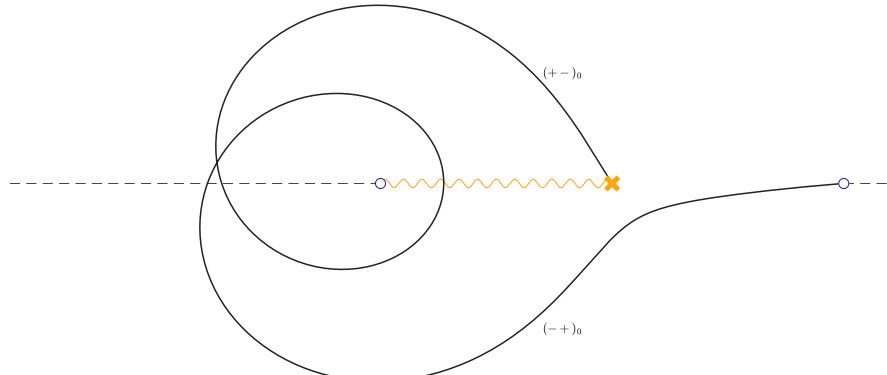

Figure 8: Finite web representing the bound state of a single D0 with a D4-brane at $\vartheta_{(1,1)}$.

for the (1) partition

$$
\begin{array}{c}
\mathbb{C} \cdot 1 \\
I \uparrow \\
\mathbb{C}
\end{array}
\tag{72}
$$

Geometrically, the web lifts to a cycle in $\Sigma$ in the same homology class as $\gamma_{\text{D4}} + \gamma_{\text{D0}}$, shown in Fig. 10. Note that the cycles for the D0 and D4 shown in Figs. 5 and 6, respectively, are calibrated for different values of $\vartheta$. We can connect these two cycles by a surgery at the branch point, according to the interpretation of the entry "1" in $I$ above as surgery parameter. The connected cycle can then be deformed continuously to the cycle shown in Fig. 10 and which hence is calibrated for $\vartheta = \vartheta_{(1,1)}$.

$n = 2$  Adding a second D0-brane to the D4-D0 bound state we just described gives us an occasion to demonstrate explicitly how the surgery procedure reproduces the bound states expected for the two partitions of $n = 2$, $(1,1)$ and $(2)$. Fig. 11 shows the cycle for a D0 in green and the unique D0-D4 bound state in blue. We emphasize again that the cycles are calibrated for different $\vartheta$. The solid lines are the strands moving on the $+$-sheet, while the dotted lines are moving on the $-$-sheet. The lines change from solid to dotted precisely at the square root branch cut. The logarithmic cut is on the $+$-sheet, so the sheet index $N$ on the solid lines increases by 1 when crossing from above, while being invariant on the dotted lines. To bind the free D0 to the cycle, we need to connect the cycles via surgery at an intersection point. We will focus on surgeries at the intersections inside the red circle. Surgery on the other intersection is related by a continuous deformation (see section 5).

We have two possibilities to perform the surgery. We can either do it on the $+$-sheet or on the $-$-sheet, and since the surgery has to be performed in a way that preserves the orientation of the cycle, there is only one possible way to do it on each sheet. One of these should lead to the (2) partition and the other to the $(1,1)$ partition. To set up the correspondence with the

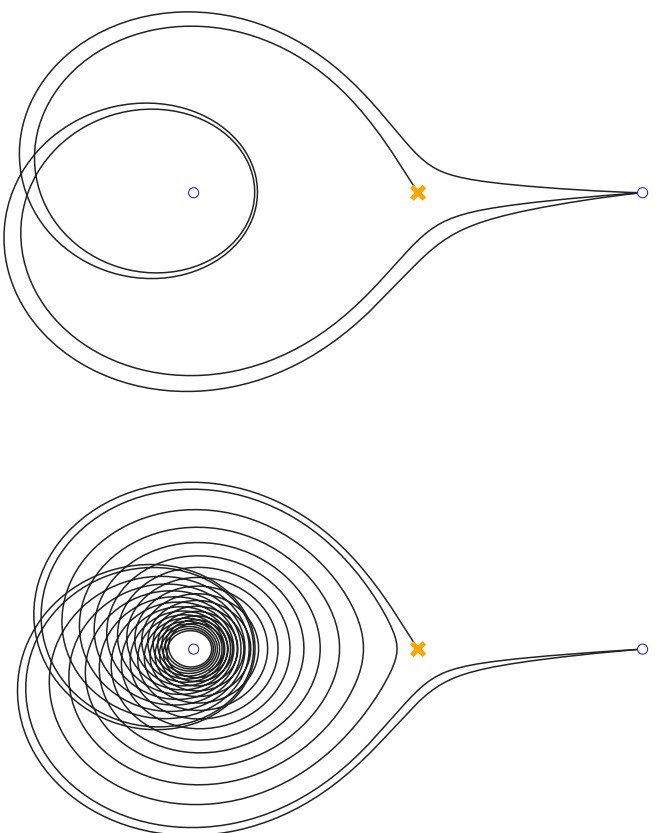

Figure 9: Wallcrossing for the (1) partition. The upper panel is for $\vartheta_+ \gtrsim \vartheta_{(1,1)}$, the lower $\vartheta_- \lesssim \vartheta_{(1,1)}$.

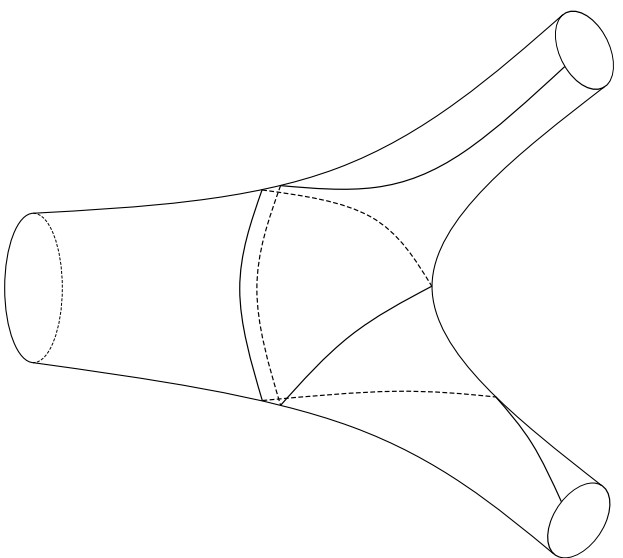

Figure 10: Cycle representing a bound state of a single D0 with a D4-brane.

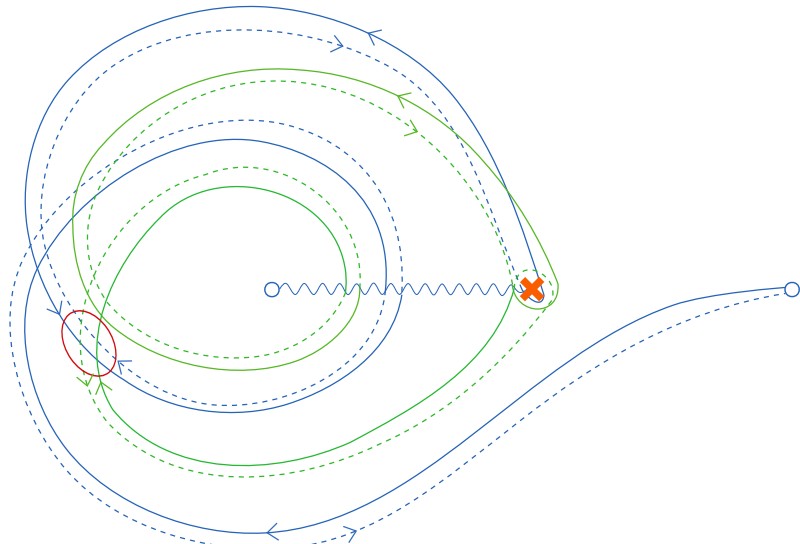

Figure 11: Cycles for D0, and D0-D4 on the $x$-plane.

two ADHM modules given by eq. (28),

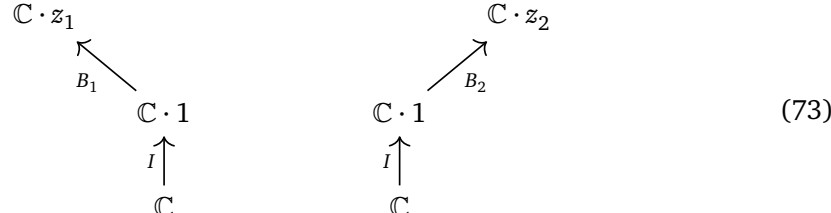

$$\tag{73}$$

we identify the box at the bottom of the Young diagram (i.e., the image of $I$) with the D0 that was bound to the D4 in Fig. 8, the non-zero entry in $B_1$ that attaches a box to the left with a surgery on the "+"-sheet, and the non-zero entry in $B_2$ that attaches a box the right with a surgery on the "−"-sheet. Note that this process is compatible with the $\mathbb{Z}/2$-symmetry transposing the Young diagram and exchanging the two sheets of the covering.

The result of the surgery on the +-sheet is shown in Fig. 12. Since the original blue and green lines were calibrated for different $\vartheta$, we need an additional finite deformation in such a way that always two strands in $\Sigma$ come together to make a calibrated trajectory in $C = \mathbb{C}_x^\times$ at the critical angle $\vartheta_{(1,2)}$. At the end, we obtain exactly a lift of the finite web at $\vartheta_{(1,2)}$ shown in Fig. 13, in which the green trajectory is an $\mathcal{E}$-wall of type $(--)_{-1}$.

This illustrates the use of the junction rules for exponential networks given on page 14: The line coming around the puncture is an $\mathcal{E}$-wall of type $(-+)_{-1}$, while the trajectory coming from the branch point is of type $(+-)_0$. At the junction, we can generate either a $(++)_{-1}$ or a $(--)_{-1}$ trajectory. These being calibrated by the same differential, the finite webs have the same shape. It's clear that the cycle in Fig. 12 is the lift of the finite web with $(--)_{-1}$ and that if we performed the surgery on the −-sheet, we would get a $(++)_{-1}$ trajectory instead.

To appreciate the relevance of the logarithmic differential in the story, it is instructive to study the lift to $\widetilde{\Sigma}$ systematically by tracking the pieces of the cycle, starting from the branch point or cutoff, along each line through the full web. Borrowing notation from [7], this involves lifting the labels $(ij)_n$ of $\mathcal{E}$-walls to named pairs of strands $x|y(i_{N+n}j_N)$, where the logarithmic branch $N$ is chosen for each $\mathcal{E}$-wall such that the upper and lower strands (taken with multiplicities) match up according to the Kirchhoff rule at each junction, and and letters

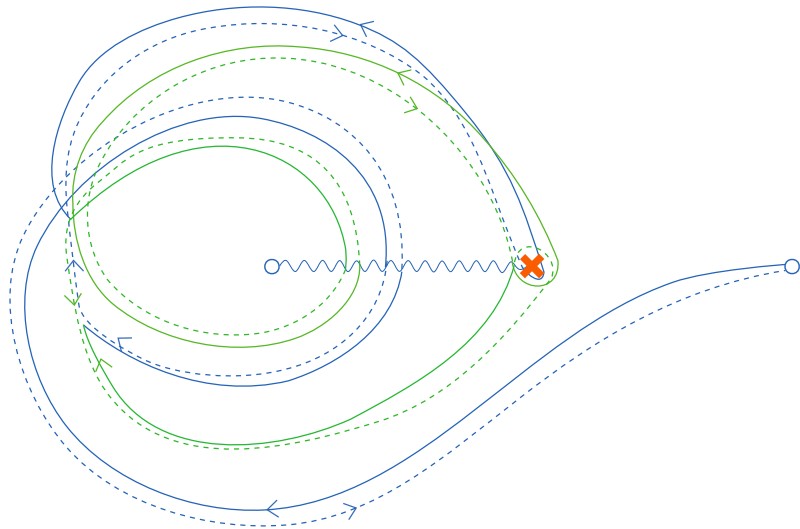

Figure 12: Cycle for bound state of 2D0-D4 on the $x$-plane.

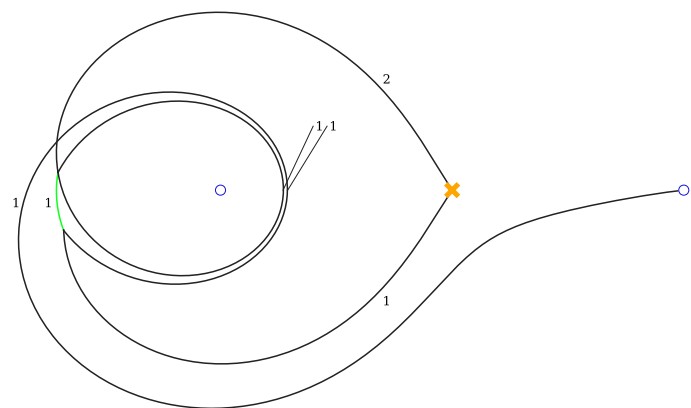

Figure 13: Finite web representing the bound state of two D0 with a D4 at $\vartheta_{(1,2)}$.

$x, y$ keep track of the individual strands upstairs. (When $x = y$, we write it only once.)[17]

To illustrate this procedure for the finite web in Figure 13 with label $(--)_{-1}$ on the green lines, we lift the $\mathcal{E}$-wall of type $(+-)_0$ representing the base D4-D0 to the "pair of strands" label $a_1(+_0-_0)$. After crossing the logarithmic[18] and square-root cut, this comes back around the puncture as $a_1(-_{-1}+_0)$, where it pairs with the double strand $b_1(+_0-_0)$ coming out of the puncture to become the green double strand $a_1|b_1(-_{-1}-_0)$, releasing the $a_1(-_{-1}+_{-1})$ into the puncture to morph into $b_1(+_{-1}-0)$, before crossing the square-root and logarithmic cut to itself return to the puncture as $b_1(-_{-1}+_{-1})$. This is captured by the flow chart (which of

---

[17]The conventions for these labels will be discussed in more detail in section 4.4.

[18]Recall that in our conventions, the logarithmic cut is on the +-sheet, and its subscript decreases by 1 when crossing from below.

course, in principle could be drawn for any value of $N$)

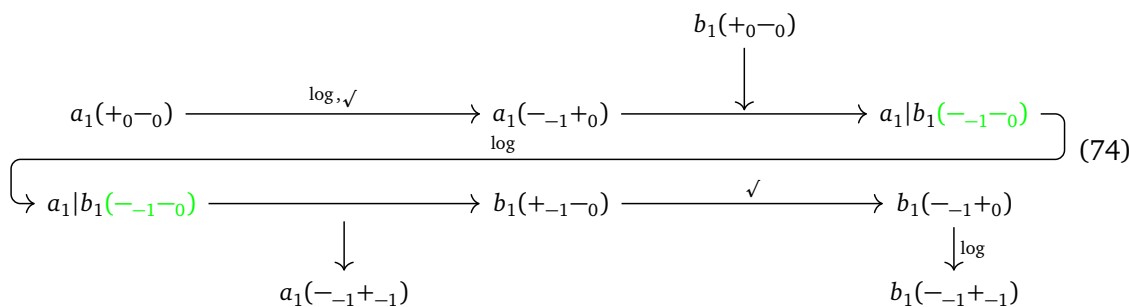

$$\tag{74}$$

and is easily checked to agree with the cycle in Fig. 12. Similarly, we find the following chart for the network in figure 13 with label $(++)_{-1}$ along the green line:

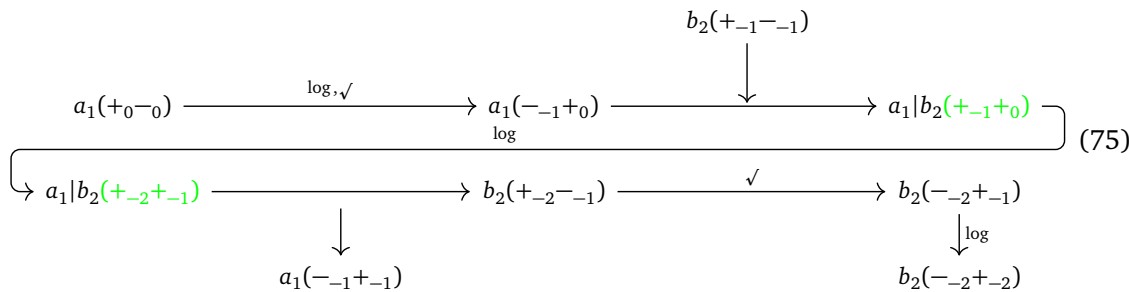

$$\tag{75}$$

A noteworthy feature of this presentation of the cycles is the shift of logarithmic sheet from beginning to end, which is $-1$ for the web with $(--)_{-1}$ green line and $-2$ for $(++)_{-1}$. This is related to the observation that the strands coming out of the chart (which flow back to the puncture from above) live on the same logarithmic sheet in the $(--)_{-1}$ case but on different ones for $(++)_{-1}$. These sheets indicate the shift of the indicated D0-brane that needs to be performed before the surgery can take place on $\widetilde{\Sigma}$. It turns out that for the general finite web of weight $n$ that we discuss below, the vector $(k_1, k_2, k_3, \dots)$ specifying the number $k_N$ of D0-branes that must be supplied with sheet shift by $N$ is precisely the partition one wishes to construct, while the logarithmic difference is the number of non-zero entries of $(k_1, k_2, k_3, \dots)$, i.e., the length of the partition. Here, the breaking of the symmetry under transposition of Young diagrams can be traced to the placement of the logarithmic cut on one of the sheets (the $+$-sheet in our convention).

To illustrate the wall-crossing in this case, we plot the relevant parts of the full exponential network at angles slightly above and below $\vartheta_{(1,2)}$ in Fig. 14. It is apparent from the topology of these networks that the critical double walls make up precisely the finite web in Fig. 13 which we can split into two components as mentioned above.

The observation that the two finite webs sit on top of each other, distinguished only by the type of the $(\pm\pm)$ lines, is a general fact: Partitions that are related by transposition have the same finite webs with reversed labels for the $(\pm\pm)$ lines.

**$n = 3$** The construction of bound states of one D4 and 3 D0-branes involves finite webs of different shape for the first time, and is also the first occurrence of holomorphic disks in critical

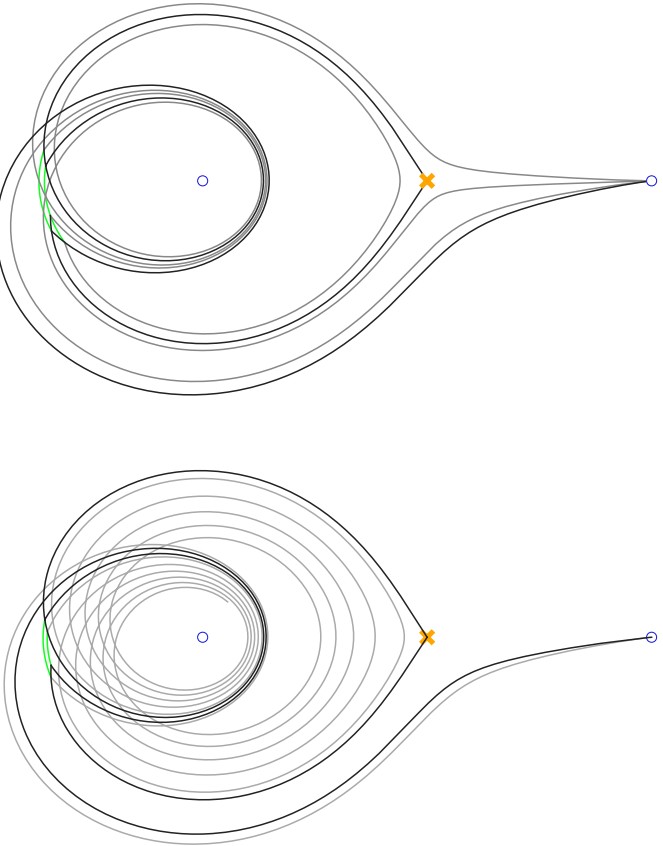

Figure 14: Wallcrossing across $\vartheta_{(1,2)}$.

finite webs. We begin with the (3) partition, corresponding to the ADHM tree module

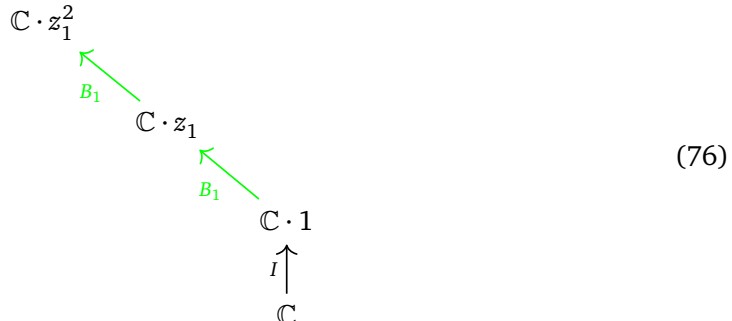

(76)

Following the procedure explained above with surgery parameters extracted from the matrices (30) we deduce that this bound state should have two double walls of type $(--)_{-1}$. Indeed, we find such a finite web at the critical angle $\vartheta_{(1,3)}$, shown in Figure 15. Of course, the finite web for the transposed partition $(1,1,1)$ looks the same with the two green lines being of type $(++)_{-1}$ instead.

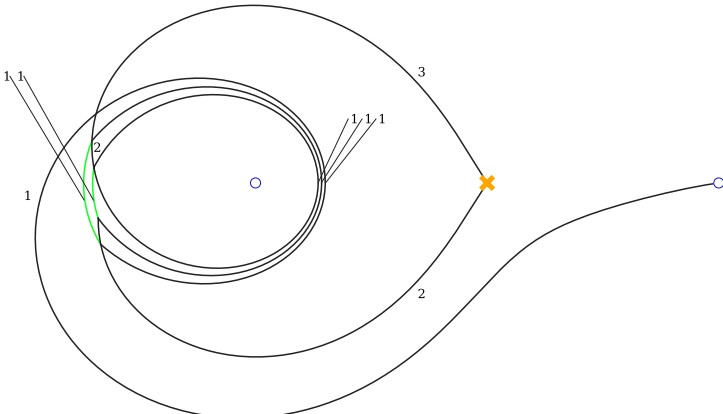

Figure 15: Finite web representing the bound state of three D0's with a D4 at $\vartheta_{(1,3)}$.

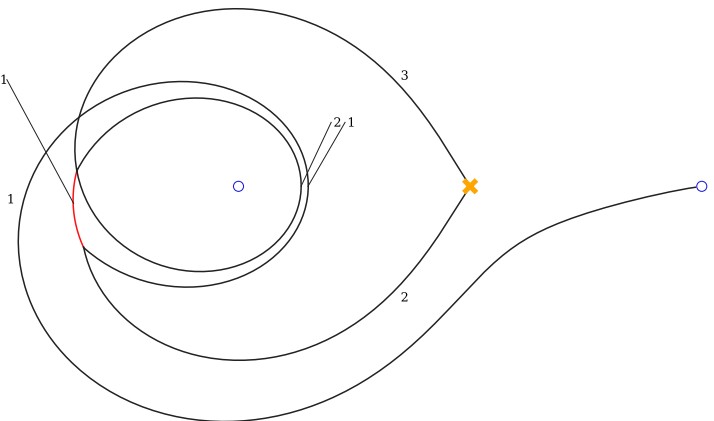

Figure 16: Finite web for the $(2,1)$ partition.

Finally, the tree module for the $(2,1)$ partition is

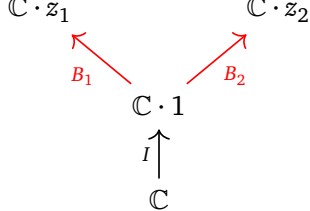

with matrices given in (32). According to the surgery procedure, there should be a corresponding finite web that contains one line of type $(+-)_{-2}$, which indeed exists at $\vartheta_{(1,3)}$ as shown in Fig. 16. Note that the absence of green lines reflects the symmetry of the $(2,1)$ partition under transposition.

**Holomorphic disks**    Naively [7], one might ask about two more finite webs, of the same shape as Fig. 15, but with one of the green lines of type $(++)_{-1}$ and the other of type $(--)_{-1}$. Working backward, it is not hard to see that one could attempt to engineer such a configuration by putting two D0-branes on top of the D4-D0 bound state of Fig. 8, see Fig. 11, and performing one surgery on the $-$-sheet, and one on the $+$-sheet. This would correspond to the surgery

parameters of eq. (34), in other words, to the "non-module"

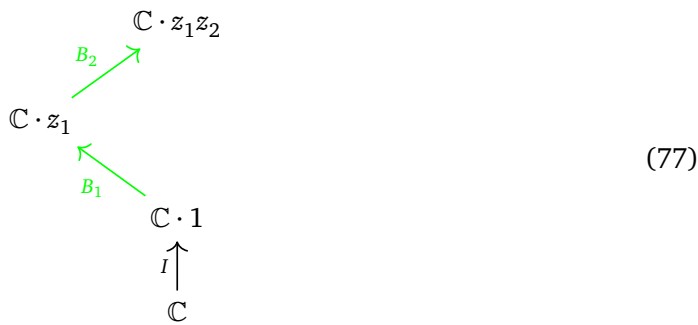

$$(77)$$

On general grounds, the failure of $B_1$ and $B_2$ to commute, i.e., the violation of the F-term constraints, should manifest in the existence of a holomorphic disk bounding the finite web. To see this, let us assume for definiteness that the outer line carries $(--)_{-1}$ and the inner line $(++)_{-1}$. Then using the same conventions as above, we would associate the following chart to the finite web

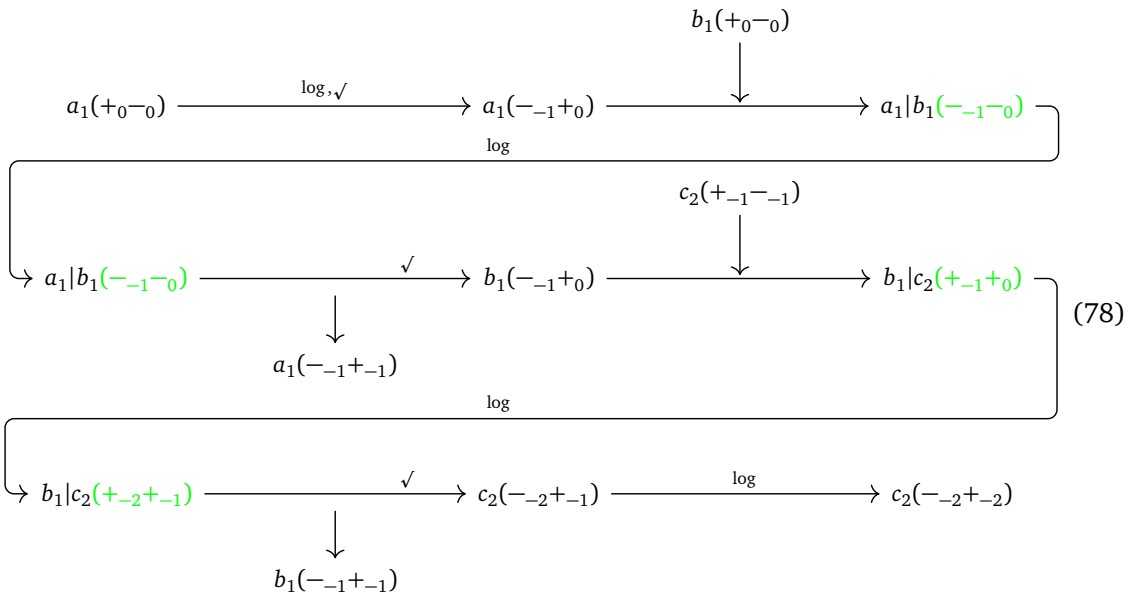

$$(78)$$

This exhibits a holomorphic disk on the $-_{-1}$ sheet with boundary starting at the branch point to the South-West along the $c_2-_{-1}$ line, turning sharp right into $b_1-_{-1}$ and around the puncture at $\infty$, before making another right into $a_1-_{-1}$ and going back to the branch point from the North to close. By shift invariance, there is such a disk for every other value of $N$ as well. Also note that the labels $a_1, c_2$, of the participating strands at the branch point, which is the F-term that is being violated, indicate precisely the position of the non-zero entry of $[B_1, B_2]$.

As a result, the purported web with green lines of different types, while lifting to a fine calibrated cycle in the same homology class as a D4 + 3D0, i.e., satisfying the D-term constraint, supports a holomorphic disk and hence in fact violates the F-term constraints.[19] It therefore does not correspond to any point in the ADHM moduli space, much less a fixed point of the torus action.

---

[19]From the point of view of the Fukaya category on $\Sigma$, this would be interpreted as an "anomaly" or "obstruction" in the square of the Floer differential.

Again, this is a general fact. "Gravitationally unstable" box configurations, such as

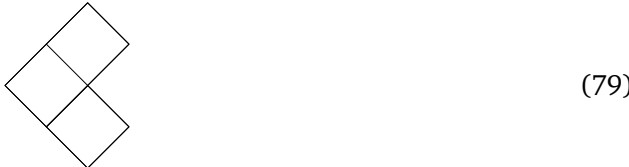

$$(79)$$

corresponding to (77) or its transpose corresponding to the other combination of $(++)_{-1}$ and $(--)_{-1}$ on the two green lines, and signaled by non-commuting matrices $B_1, B_2$, always lift to cycles on $\Sigma$ (or $\widetilde{\Sigma}$) bounding holomorphic disks of the type described above.

**$n = 4$** We have two more elements of the relation between networks and partitions to uncover before we can describe it in general.

For the (4) partition, the situation is not too different from before, see in particular fig. 15. The finite web will have 3 green lines which all have the same label $(--)_{-1}$. Transposing the partition leaves the shape of the web unchanged, however the green lines now carry labels $(++)_{-1}$, which corresponds to the partition $(1,1,1,1)$. Having mixed labels $(++)$ for some strands and $(--)$ for others, gives a fine calibrated finite web. However, as discussed in the $n = 3$ case, one finds holomorphic disks, and the resulting cycle will be obstructed. Matching with the fixed points of the moduli space of quiver representations, this means that the resulting "non-module" doesn't respect the F-term condition $[B_1, B_2] = 0$ (since $J \equiv 0$). From the partitions point of view, it means that the resulting box configurations are not gravitationally stable. For general $n$, even though the finite web corresponding to the $(1,1,\ldots,1)$ partition naively has a combinatorial factor of $2^{n-1}$, only 2 of them are unobstructed.

Moving on, the ADHM module for the $(2,2)$ partition,

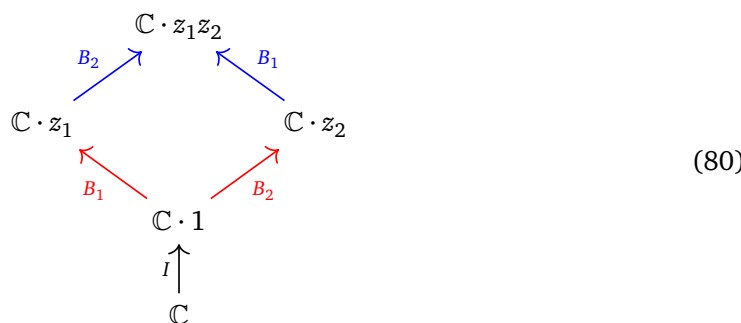

$$(80)$$

constructed with surgery parameters in (36), corresponds to the finite web in Fig. 17 with a line of type $(+-)_{-1}$, which we depict in red, on the outside and another line of type $(-+)_{-2}$, shown in blue, on the inside. Again, this is symmetric under transposition, or exchange of the $+$- and $-$-sheets.

A noteworthy property of this finite web is that the lifted cycle bounds two holomorphic disks, one on each sheet, that cancel each other. This amounts to the statement that $B_1 B_2 = B_2 B_1 \neq 0$ for the tree module, see eq. (37). Again, this is a general fact and similar cancellations occur as soon as the partition contains a box configuration of shape

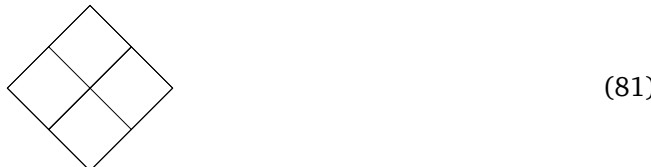

$$(81)$$

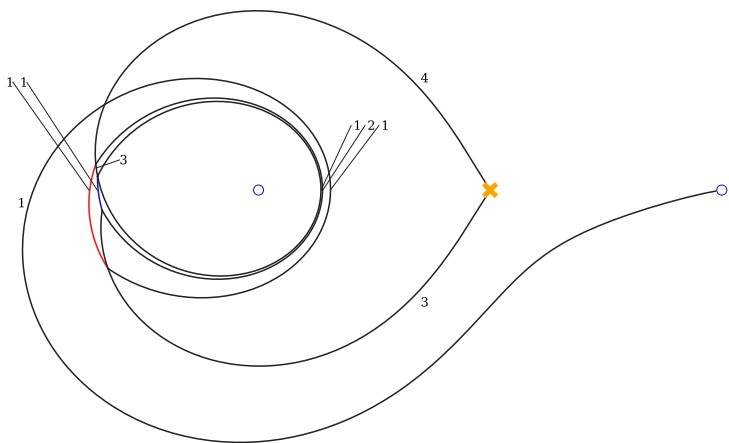

Figure 17: Finite web for the $(2, 2)$ partition.

To finish off, we consider the $(3, 1)$ partition, which by symmetry will take care of the $(2, 1, 1)$ partition as well. A novel feature of the associated ADHM module

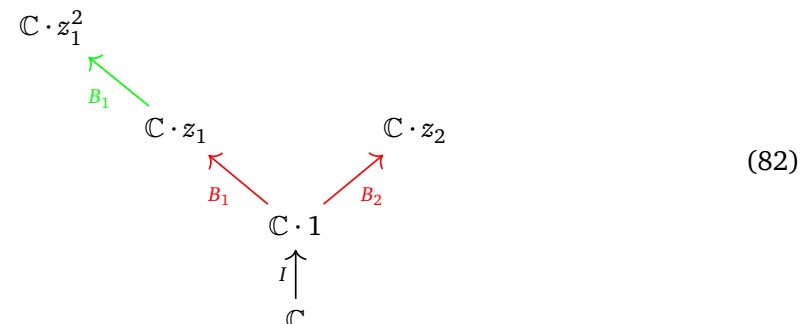

$$(82)$$

is the presence of two "ends", basis vectors $v$ such that $B_1 v = B_2 v = 0$ at different "heights" in the Young diagram, in the example generated by $z_1^2$ and $z_2$. While it is at first [7] rather non-obvious how this should look at the level of the finite webs, a little meditative experimentation with surgery parameters at the Black Sheep Coffee Lab reveals that the correct finite web is as shown in Fig. 18.

Namely, in addition to the red line of type $(+-)_{-1}$ and the green line of type $(--)_{-1}$, left-overs from the surgeries, we require a peculiar auxiliary line we choose to depict in gray, and which in the example is of type $(--)_{-2}$, that emerges from the junction of the red line with the original black line of type $(-+)_{-1}$ in order to close it off at the lower height.[20] From the point of view of bound state formation, this configuration results from deforming the constituent D0-brane as shown in Fig. 19 (see ref. [7] and section 5 for further information about such deformations) along with performing the surgery, such that the gray line in the network in Fig. 18 is a remnant of the gray line in Fig. 19 introduced by the deformation. Note that the structure of a red line on the outside and a green line on the inside, which is expected from the structure of the tree module (82) is preserved under this deformation.

As we are now ready to show in the next section, this works in general: The homology class of any cycle representing a bound state of a D4 with $n$ D0-branes is $\gamma_{D4} + n\gamma_{D0}$. Any calibrated cycle is a continuous deformation of the cycle obtained by performing suitable surgeries be-

---

[20]Only the assignment $(--)_{-2}$ to the gray line does not produce a holomorphic disk.

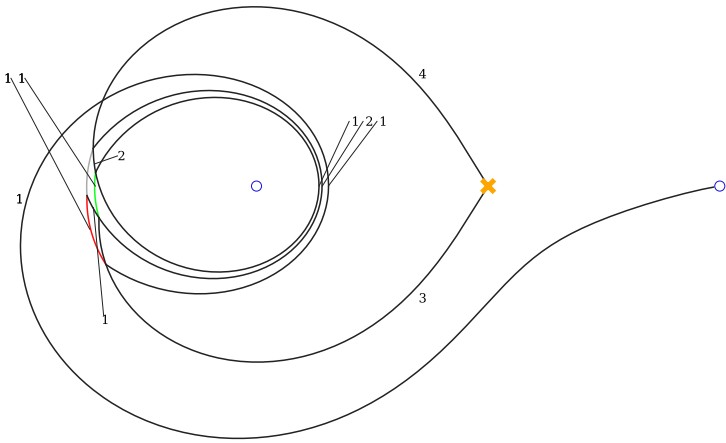

Figure 18: Finite web for the $(3, 1)$ partition.

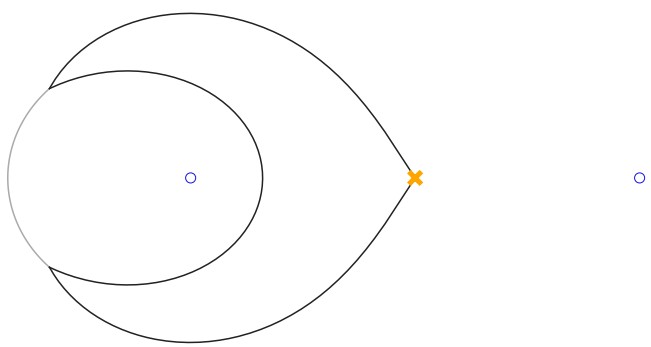

Figure 19: Deformation of the D0-brane.

tween the constituents, allowing for prior deformations of the D0s along their moduli space. Ignoring any cycle that bounds a holomorphic disk, and relegating any further remedial comments to section 5, it remains to describe the exact correspondence with torus fixed points of the ADHM moduli space, partitions, and tree modules.

# 4 Networks for partitions

In this section, generalizing the considerations of section 3, we will explain how to construct an anomaly-free finite web on the pair of pants (1) for each $(\mathbb{C}^\times)^2$-fixed representations of the ADHM quiver (4) with $\dim W = 1$, $\dim V = n$, labeled by partitions/Young diagrams with $n$ boxes. We emphasize that although we are convinced that the correspondence exists for all $n = 1, 2, 3, \dots$, we do not consider our present description a full proof just yet, and intend to give a complete account elsewhere. For the convenience of those readers who have skipped sections 2 and 3, we will recollect all building blocks and construction rules, but must begin with a bit of terminology.

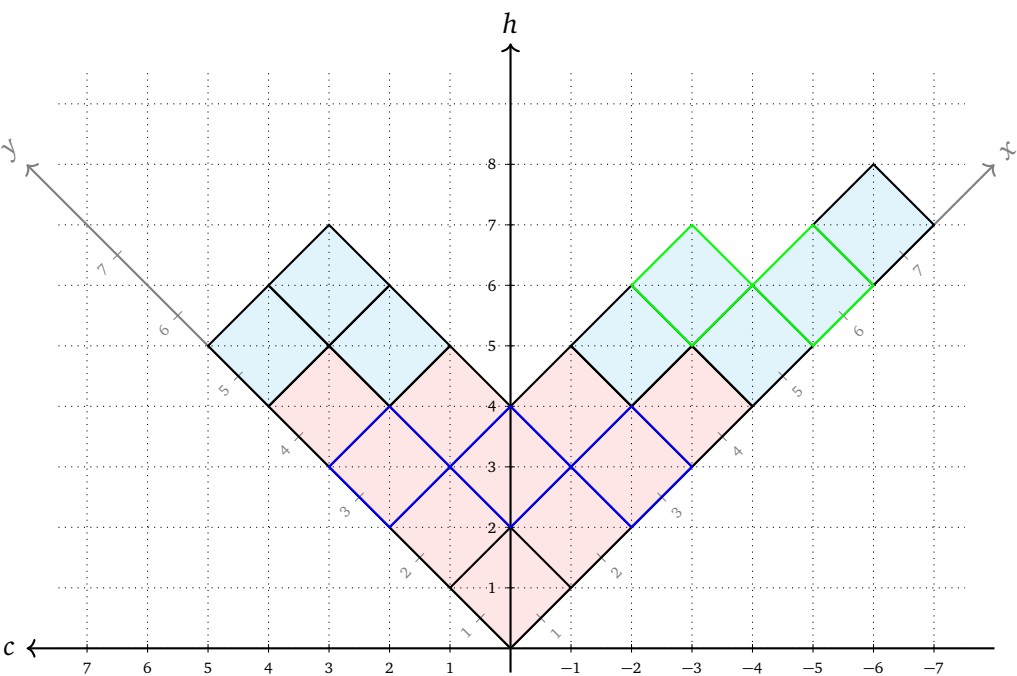

Figure 20: The life of the partition $(5, 5, 2, 2, 2, 1, 1)$, of girth 4 and weight 18. Shaded in red is the youth, bordered in blue, the (penultimate intact) generation at age 3, and bordered in green: a team of mavericks. There are two endlings at age 6, and one at age 7.

## 4.1 The life of partitions

According to standard (English) conventions, the row number in Young diagrams is the $x$-coordinate, and the column number, the $y$-coordinate, both valued in the positive integers. Thinking in Russian style, the horizontal coordinate $c = y - x$ (increasing to the left), is the standard *content* of a box. Motivated by the network construction (and speaking somewhat French) we will refer to the vertical "height", $h = x + y - 1$, as the *age* of a box, and to the set of boxes in any given Young diagram, $Y$, at the same age, as a *generation*, $\text{gen}(h) = \{\square \in Y \mid h(\square) = h\}$. As usual, older generations come first. The number of boxes in a generation, $\text{wt}(h) = \# \text{gen}(h)$, is called its *weight*. Clearly, both content of a box and weight of a generation are bounded by the age, with $|c(\square)| \le h(\square) - 1$ and $wt(h) \le h$. A generation with $\text{wt}(h) = h$ may be called *intact*.

It is a deep fact about partitions that starting at the beginning with $\text{wt}(1) = 1$, the weight of successive generations at first increases linearly with age, $\text{wt}(h) = h$, but that as soon as the weight stops growing once, it can only remain constant or decrease, but never start growing again. We will refer to the maximal weight of a generation as the *girth* of the partition, which is hence also the age of the last intact generation, $g(Y) := \max\{h \mid \text{wt}(h) = h\}$. The set of generations with $h \le g(Y)$ will be called the *youth* of the partition.

As soon as $h > g(Y)$, generations can not only not grow any longer, but also split up. We will refer to the connected components of a generation as *teams* (yoked together at the corners), and the number of boxes they contain as weight, as before. After splitting, teams evolve separately, and can neither join nor grow again. Teams that do not touch the sides, i.e., contain no box of maximal content, must decrease monotonically in weight and die out at least linearly with increasing age. Teams that touch the sides (we call them *mavericks*) can retain their weight for a while, but of course must also eventually split and/or disappear. Boxes without descendant in either $x$- or $y$-direction are called *endlings*. See fig. 20 for an example.

## 4.2 Building blocks

The basic ingredient for matching exponential networks with partitions is the identification between a box in a Young diagram and the finite web corresponding to a D0-brane, depicted in fig. 5. The corresponding network is critical at $\vartheta = \vartheta_{0,1} = 0$, and, in a convenient normalization, the length/mass of the finite web is $l_{D0} = |Z_{D0}| = Z_{D0} = 4\pi^2$. To engineer bound states between the regularized D4-brane, shown in fig. 6, and multiple copies of the D0-brane, initially placed on top of each other, that web has to be deformed, both away from $\vartheta = 0$, and out on its moduli space, in the direction shown in fig. 19. This corresponds to the resolution of the self-intersection on either the $+$ or $-$-sheet of the double cover eq. (1). The same resolution is also used to attach the various D0-branes to each other by surgery, as illustrated in figs. 11 and 12.

In view of this identification, we will refer to the segments of the web connecting the branch point with the self-intersection as the *flaps* of the D0-brane (thought of as a box), and to the part encircling the puncture, as its *core*. In fig. 5, the flaps account for 1/5 each of the total length of the finite web, and the core for the remaining 3/5. Namely, $l_{\text{flap}} = 4\pi^2/5$, $l_{\text{core}} = 12\pi^2/5$. By continuity, this division remains true for small deformations, and justifies the terminology to some degree.

To keep track of the types of the various strands making up our finite webs, we find it convenient to orient the flaps away from the branch point, such they both carry labels $(+-)_0$ (see again fig. 5). Consequently, if we orient the core locally into the self-intersection, and remember the square-root cut running from the branch point to the puncture, as well as our convention that the logarithmic cut is on the $+$-sheet (see footnote 18 on page 28), the upper part of the core carries labels $(-+)_1$, and the lower, $(-+)_{-1}$. Henceforth, all strands stemming from the flaps or the core will be depicted as black lines without any branch labels.

For our preferred choice of $\vartheta_{D4} < 0$, the D4-brane will bind to the cluster of $n$ D0-branes by detaching one of the lower flaps from the branch point, as seen most clearly for $n = 1$ in fig. 8. All other flaps remain on top of each other at the branch point. This results in an initial multiplicity of $n$ for the upper, and $n-1$ for the lower flaps, which will subsequently decrease as a result of interactions.

The cores of the D0-branes, on the other hand, will in general not coincide any longer in the critical finite webs. Although they are all calibrated by the same differential $e^{-i\vartheta_{1,n}}\lambda_{(-+)_{\pm 1}}$, they start from different points along the flaps. A central element of our matching networks with partitions is that cores corresponding to boxes in the *same team* (connected components of a generation at a fixed age in the terminology of the previous subsection) remain on top of each other. Namely, the core multiplicity is identified with the weight of the team.

## 4.3 Stacking rules

The basic junction rules for exponential networks, reviewed in section 3, stipulate that incoming $\mathcal{E}$-walls labeled $(ij)_{k_1}$ and $(jk)_{k_2}$ will result in just one outgoing $\mathcal{E}$-wall $(ik)_{k_1+k_2}$, essentially as in ordinary spectral networks, as long as $i \neq k$. In the exceptional case, $\mathcal{E}$-walls $(ij)_{k_1}$ and $(ji)_{k_2}$ generate not only $(ii)_{k_1+k_2}$ and $(jj)_{k_1+k_2}$, but also two more infinite sets of walls labeled $(ij)_{(w+1)k_1+wk_2}$ and $(ji)_{wk_1+(w+1)k_2}$ for $w = 1, 2, \ldots$, see fig. 4. At the level of 3d kinky-vortex degeneracies, these walls can be thought of as the result of tracing the incoming walls alternately $w/w+1$ times, see fig. 3, remarks on page 20, as well as ref. [8]. For the construction of finite webs, and in particular for understanding their charge and mass, the preferred interpretation is as a literal "multiplicity" assigned to the incoming strands [7], in the same sense as at the end of the previous subsection.

In our building up networks for partitions, we arrive concretely at intersections of core strands labeled $(-+)_{-1}$ with the lower flaps labeled $(+-)_0$. In principle, if the multiplicities are

large enough, this can result in any number of outgoing lines of type $(+-)_{-k}$, $(--)_{-k}/(++)_{-k}$, and $(-+)_{-k}$, ordered as in 4. We depict them in red, green, and blue, respectively, and collectively refer to them as *straps*. All the while, black lines representing cores or flaps that did not participate in the interaction continue with reduced multiplicity.

Following the identification of multiple cores with teams of boxes in the Young diagram, we wish to interpret a red strap of type $(+-)_{-k}$ as stacking a *new generation of higher weight*, $k+1$, onto an extant team, corresponding to a piece in the ADHM-module diagram of the form

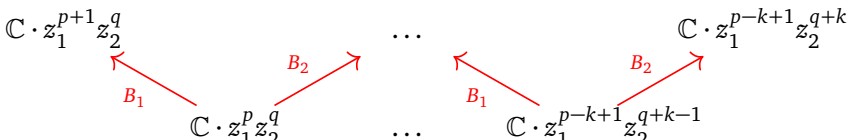

Similarly, a green strap of type $(--)_{-k}$ will correspond to a chain of the form

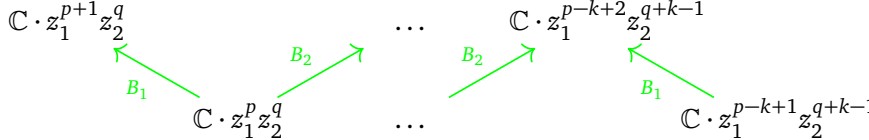

while type $(++)_{-k}$ to the transposed chain,

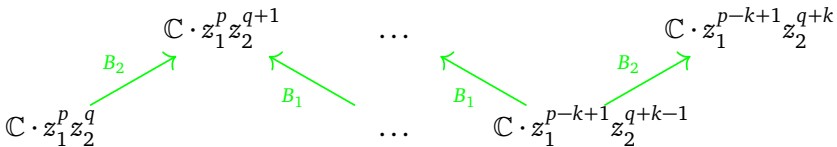

We emphasize that, of course, and according to the evolution of partitions described in subsection 4.1, the first type of stacking is only possible during the youth of the partition, when intact generations consist of a single team, and the other two only for mavericks touching the respective sides of the diagram. Chains of the form

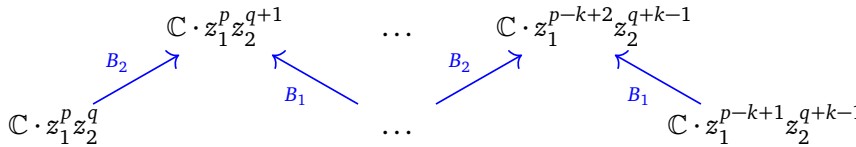

corresponding to an outgoing blue strap of type $(-+)_{-k}$, are possible (only) during old age for teams away from the sides. From the networks' point of view, these exclusions should be viewed as due to the presence of holomorphic disks, as discussed on page 32.

Note that the full specification of the straps (colored lines) requires the subscript $k$. In our pictures, however, which have at most a dozen boxes or so total, this label can mostly be inferred without ambiguity from the change of multiplicities along the black lines, so we will usually omit it. Green lines of type $(++)_{-k}$ cannot be distinguished at all from those of type $(--)_{-k}$ (nor from superpositions of lines with lower $k$). Exchanging the two corresponds to transposition of the Young diagram. When this plays a role, it will be indicated in the pictures or the text.

To finish the construction of any given partition under our local rules, it remains to explain how the straps that emerged from the interaction of cores and flaps at the lower intersection, as well as those cores that did not interact with the flaps, and which correspond to endlings in the terminology of section 4.1, join back together with the upper flaps, after crossing the logarithmic cut. As an invitation to the issue, we flash the references to the examples for weight up to 4 discussed in section 3.4.

- The finite web corresponding to the unique partition of $n = 1$ is shown in fig. 8.

- For $n = 2$, there is also a unique finite web, shown in fig. 13, with the two different partitions corresponding to the green strap being of type $(--)_{-1}$ or $(++)_{-1}$.

- For $n = 3$, there are two different pictures. The finite web corresponding to the partitions $(3)$ and $(1, 1, 1)$ is shown in fig. 15, with two green straps of the same type (lest there be a holomorphic disk). The finite web for the symmetric partition $(2, 1)$ is in fig. 16. The innermost core line has multiplicity 2, matching the weight of the last intact generation at age 2.

- For $n = 4$, there are 3 pictures, corresponding to 5 partitions up to transposition. The web for $(4)$ and $(1, 1, 1, 1)$ looks essentially like fig. 15 with one additional core and one more green line.[21] The web for $(2, 2)$, which is the first with a "Durfee square" properly speaking, and also the first featuring a blue line, can be seen in fig. 17. Finally, the web for $(3, 1)$ and $(2, 1, 1)$, which are the first partitions with endlings of different age, is shown in fig. 18.

It appears in particular from this last example that after stacking up boxes using straps of various types, the key to closing the finite webs are the interactions between these straps and the core lines corresponding to endlings of a later, younger age. We will accordingly refer to the lines coming out of these interactions (the prototypical of which is shown in gray in fig. 18) as *vestigial*.

## 4.4 On endlings and vestigial $\mathcal{E}$-walls

To solve their mystery, we have found it convenient, following [7], to fix a label for each box in the Young diagram, and to keep track where the corresponding core line contributes in the finite web, in analogy with the flow-chart presentation of the cycles on page 29. Letting $\mathrm{lett}(h)$ be the $h$-th letter in the alphabet, we label a box at age $h$ and $x$-coordinate $x$ as $\mathrm{lett}(h)_x$. For instance, the lowest box which represents the D0 initially bound to the D4 as "stability vector" of the ADHM module always has the label $a_1$. The second generation can include boxes labeled $b_1$ and/or $b_2$, etc.

The idea is now that in a general bound state constructed by attaching $n - 1$ D0-branes to the D0-D4 cycle via surgery, we have identified each core line with a team of boxes of weight equal to the multiplicity of that core line, and so we can assign it the box labels as well. The same logic can be applied to the various pieces of the flaps. In contrast, the straps (lines drawn in color) carry (possibly multiple) composite labels, since they come to life only after some surgery operations and hence can not be associated with a unique box. In section 4.3 we explained the rules for the colored lines. Now we label these lines by the boxes that contributed to their creation, with the bottom row on the left and the top row on the right, separated by a $|$. For instance, in Figure 13, if the green line is of type $(++)_{-1}$ it is labeled $a_1 | b_2$ while if it is of type $(--)_{-1}$ its label is $a_1 | b_1$.

To figure the labels that we assign to the vestigial lines, we return to the web for the $(3, 1)$ partition shown in fig. 18, with a green line of type $(--)_{-1}$. The middle of the 3 core lines having multiplicity 2 and coming from the boxes $b_1$ and $b_2$, the green line carries the label $b_1 | c_1$. The core line labeled $b_2$ however continues until it intersects the red line carrying label $a_1 | b_1 b_2$. Since $b_2$ is a line of type $(-+)_{-1}$, this behaves like a line in the bottom row, and we should label the resulting vestigial line as $a_1 b_2 | b_1 b_2$. In view of surgery, it is worth remarking

---

[21]This finite web is shown in fig. 38 for a somewhat different purpose.

that $b_2$ has indeed the correct log level, such that the lifts of $a_1|b_1b_2$ and $b_2$ intersect in $\widetilde{\Sigma}$ in the right way. This can be captured by a box scheme of the form

$$
\begin{array}{c}
b_1 \quad b_2 \\
a_1 \quad b_2
\end{array}
\tag{83}
$$

in agreement with the vestigial line being of type $(--)_{-2}$.

For the transposed partition $(2, 1, 1)$, we instead have $b_2$ and $c_3$ together forming a $(++)_{-1}$ green strap and $b_1$ remaining, which meets the red strap labeled $a_1|b_1b_2$, so the vestigial line should carry labels $b_1a_1|b_1b_2$ and be of type $(++)_{-2}$. Here we remark with respect to surgery that the complete transposition also requires moving the log cut to the opposite sheet. If we don't do that, the cycle we obtain in the logarithmic cover $\widetilde{\Sigma}$ is a single connected cycle extending through all sheets, as opposed to the situation where we have disconnected copies of cycles, isomorphic to the cycle in $\Sigma$. With reference to the discussion on page 20, this means that the projection $\tilde{\gamma} \to \gamma$ would not always have a section. For this reason, it is important to work with the infinite tower (i.e., the full preimage $\pi^{-1}(\gamma) \subset \widetilde{\Sigma}$) as advocated in [8].

The general rule is that core lines corresponding to a team of endlings interact (only) with a strap carrying precisely matching labels, generating a specific vestigial line, and possibly continuing with reduced multiplicity and labels replaced with their respective ancestors. As a result, the vestigial lines, like the straps, carry composite labels of two sets of adjacent boxes that differ in age and weight by at most one. This also fixes their type by extension of section 4.3, and, at the end of the day, the topology of the finite web corresponding to the given partition.

We illustrate the salient features and some consequences of these rules on a number of further examples. First of all, consider all partitions of weight 11, girth 4, and maximal age 5, which can all be obtained by putting exactly one box on top of the $(4, 3, 2, 1)$ partition

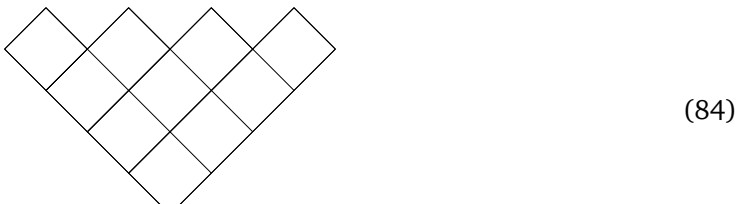

$$\tag{84}$$

and whose finite web is made with red straps carrying labels

$$
\begin{aligned}
c_1c_2c_3&|d_1d_2d_3d_4\,, \\
b_1b_2&|c_1c_2c_3\,, \\
a_1&|b_1b_2\,.
\end{aligned}
\tag{85}
$$

Putting the final box on the left-most position gives us the $(5, 3, 2, 1)$ partition by introducing a green strap of type $(--)_{-1}$ labeled $d_1|e_1$. As the team of endlings $d_2, d_3, d_4$ intersect the first (innermost) red strap, it produces a vestigial line of type $(--)_{-4}$ carrying labels $c_1d_2d_3d_4|d_1d_2d_3d_4$, while the endlings continue with multiplicity 2 and labels $c_2, c_3$. These then interact with the next red strap such that the second vestigial line, of type $(--)_{-3}$, carries labels $b_1c_2c_3|c_1c_2c_3$, before the story ends as in (83). We symbolize this as follows:

$$
\begin{aligned}
c_1d_2d_3d_4|d_1d_2d_3d_4 &\rightsquigarrow c_2c_3\,, \\
b_1c_2c_3|c_1c_2c_3 &\rightsquigarrow b_2\,, \\
a_1b_2|b_1b_2\,, &
\end{aligned}
\tag{86}
$$

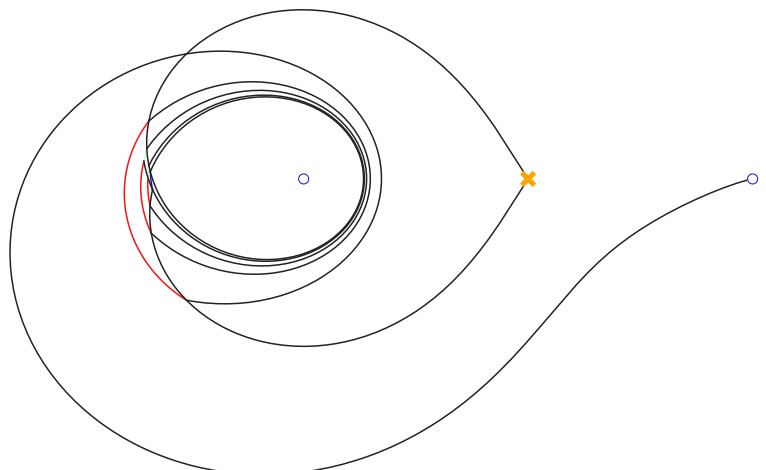

Figure 21: Finite web for partition $(4, 4, 2, 1)$.

trusting the reader will draw the picture for themselves. To instead put the 11-th box on the next-to-left-most position in order to make the partition $(4, 4, 2, 1)$, we use a blue strap labeled $d_1 d_2 | e_2$, and remain with only two endlings $d_3, d_4$. Then our rules produce vestigial lines with labels

$$
\begin{aligned}
c_1 c_2 d_3 d_4 | d_1 d_2 d_3 d_4 &\rightsquigarrow c_3 \,, \\
b_1 b_2 c_3 | c_1 c_2 c_3 \,, &
\end{aligned}
\tag{87}
$$

while the last red strap in (85) remains untouched, see fig. 21.

To, finally,[22] put the extra box at the center, giving the partition $(4, 3, 3, 1)$, we use a blue line labeled $d_2 d_3 | e_3$, remaining with two teams of endlings labeled $d_1, d_4$. When these meet the first red strap, they can attach one left and one right to immediately produce the vestigial line of type $(-+)_{-4}$ with labels

$$
d_1 c_1 c_2 c_3 d_4 | d_1 d_2 d_3 d_4 \,,
\tag{88}
$$

while the two outer red straps are unaffected. Although (88) looks non-symmetric, it is indeed symmetric under labeling with $x$- and $y$-coordinates, which is necessary since we are dealing with a symmetric partition.

The finite web for the $(4, 1, 1)$ partition, shown in fig. 22 requires three green and one red straps with labels

$$
\begin{aligned}
c_1 &| d_1 \,, \\
b_1 &| c_1 \,, \\
b_2 &| c_3 \,, \\
a_1 &| b_1 b_2 \,,
\end{aligned}
\tag{89}
$$

the middle two of which lie on top of each other and have opposite type. At the intersection with the endling $c_3$, our rules stipulate that the $b_2 | c_3$ strap of type $(++)_{-1}$ turn into a vestigial line with labels $c_3 | c_3$ of type $(--)_{-1}$, while the endling continues with label $b_2$. As a consequence, in the region above the black endling line, we have no coinciding $(++)$ and $(--)$ strands, which ensures that there be no non-canceling holomorphic disks. This example illustrates that endlings do not have to intersect visibly with straps on the first occasion (in

---

[22]Partitions $(4, 3, 2, 2)$ and $(4, 3, 2, 1, 1)$ follow by transposition.

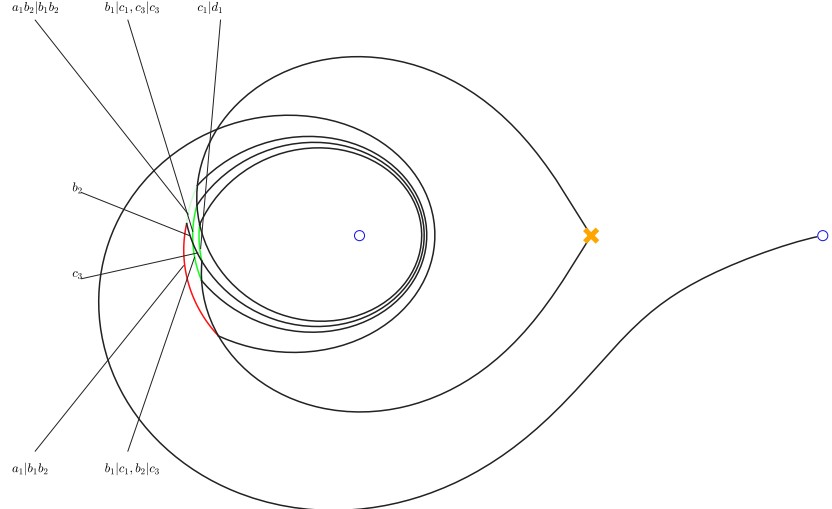

Figure 22: Finite web for partition $(4, 1, 1)$.

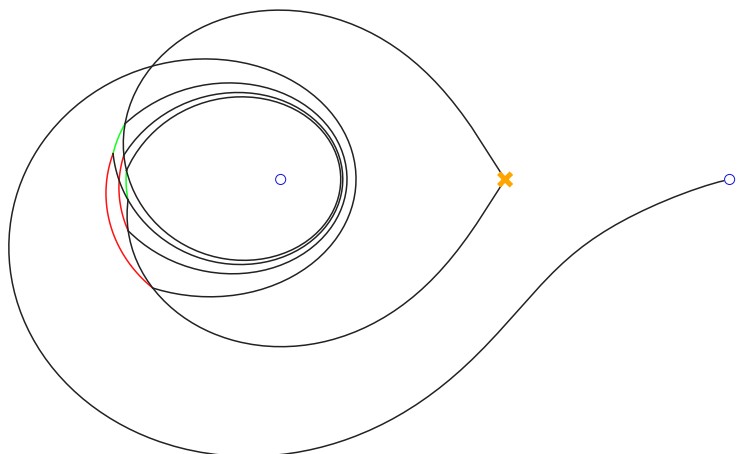

Figure 23: Finite web for partition $(4, 2, 1, 1)$.

the sense that the lines pass through each other without changing direction), while the labels from our complete labeling scheme might still be affected.

The finite web for the $(4, 2, 1, 1)$ partition has somewhat similar features and moreover contains superimposed vestigial line of mixed type $(++)_{-1}$, $(--)_{-1}$, see fig. 23.

As an example with endlings of three different ages, consider the $(5, 3, 1)$ partition,

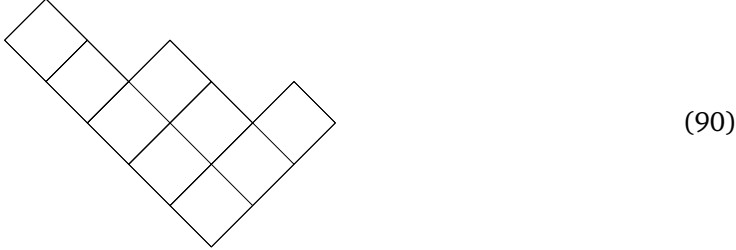

$$(90)$$

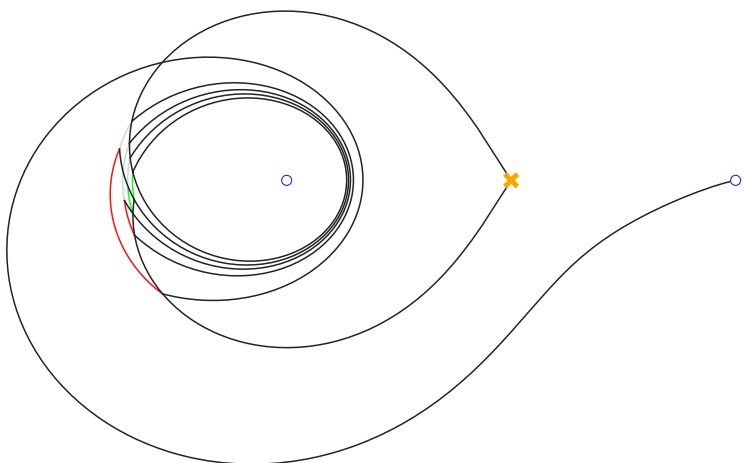

Figure 24: Finite web for partition $(5, 3, 1)$.

whose web has two red and two green straps labeled

$$
\begin{aligned}
d_1 | e_1\,, \\
c_1 c_2 | d_1 d_2\,, \\
b_1 b_2 | c_1 c_2 c_3\,, \\
a_1 | b_1 b_2\,,
\end{aligned}
\tag{91}
$$

Here, the endling $d_2$ interacts with the second green strap for a vestigial line of type $(\text{——})_{-2}$ labeled $c_1 d_2 | d_1 d_2$ emitting $b_2$. $c_3$ interacts with the first red one to give $b_1 b_2 c_3 | c_1 c_2 c_3$, which then interacts with $c_2$ to make $b_1 c_2 c_3 | c_1 c_2 c_3$ emitting $b_2$. The second red strap finally absorbs $b_2$ to make $a_1 b_2 | b_1 b_2$ see fig. 24.

Our final example is the finite web for the partition $(4, 2, 2)$, shown in fig. 25. It illustrates that while as emphasized above, core lines of boxes in the same team always lie on top of each other, teams of the same generation (age) might differ. The core of the box labeled $d_1$ is the innermost circle, of multiplicity 1, while the core of $d_3$ has come to lie on top of the cores of younger age 3.

# 5 Beyond fixed points, speculations and future directions

The aim and focus of this paper has been to describe a correspondence between Young diagrams, parameterizing torus fixed point on $\mathrm{Hilb}^n(\mathbb{C}^2)$, and a certain type of finite webs in critical exponential networks on the pair of pants. Among the outcomes has been a clarification of the role of holomorphic disks, and the development of a somewhat botanical language for linear partitions that is well-adapted to the finite webs. While we feel that this is evidence enough, we have not claimed that this correspondence is explicitly one-to-one, how it will extend beyond the fixed points to a full B-model description of the Hilbert scheme, or even how exactly wall-crossing witnesses holomorphic disks. The purpose of this section is to describe our best understanding of the remaining issues.

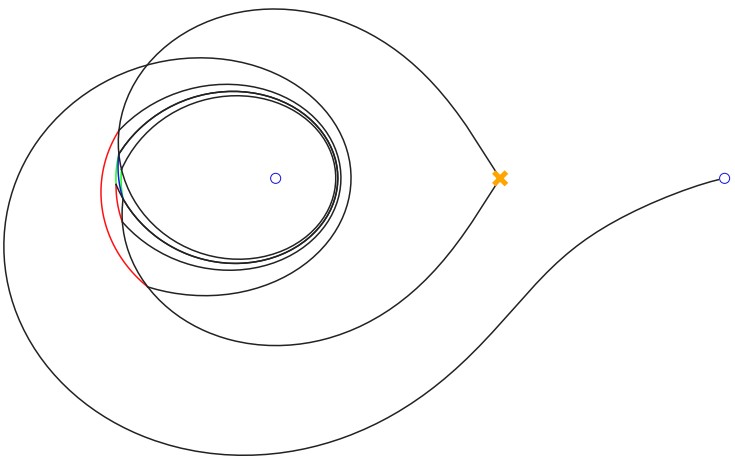

Figure 25: Finite web for partition $(4, 2, 2)$.

## 5.1 Navigating through A-brane moduli space

Our matching of partitions with finite webs verifies a rather non-trivial prediction of (toric) mirror symmetry at the level of the Euler characteristic of the A-brane moduli, each of our finite web being viewed as the projection of a special Lagrangian in the mirror geometry (38) to the $\mathbb{C}_x^\times$-plane. In the following, we give some partial evidence for projections of special Lagrangians deformed away from the torus fixed points, but leave it for a future investigation whether one can actually construct the full family of special Lagrangians corresponding to D4-$n$D0, and show that the corresponding family of A-branes is indeed parameterized by $\mathrm{Hilb}^n(\mathbb{C}^2)$, as it should be [25]. More precisely, as is explored (for specific examples) in [11, 26] it is expected that the family of deformations of special Lagrangians $L$ of a fixed homology class $[L] \in H_3(Y, \mathbb{Z})$ forms a foliation of (a region of) $\Sigma$ and a fibration of (a region of) $Y$. Denote this family by $\mathfrak{M}_L$. The critical leaves of such a foliation are conjectured to correspond to critical finite webs. Let us describe this family in more detail. At zero string coupling, each Lagrangian submanifold $L$ of $Y$, can be used to describe a supersymmetric A-brane, by considering the pair of $L$ together with a unitary line bundle over it. Then, the family of classical space of deformations of A-branes $\mathcal{M}_L$, specified by $[L]$ takes the form [11, 26]:

$$T^{b_1(L)} \to \mathcal{M}_L \to \mathfrak{M}_L\,, \tag{92}$$

where the fiber $T^{b_1(L)}$ parameterizes the flat connections over a generic point in $\mathfrak{M}_L$. However, the fibration (92) is not smooth, since there are points in $\mathfrak{M}_L$ where $L$ has some of its 1-cycles pinched, and, on those points, the fiber drops rank. According to the proposal of [11], there should exist points in $\mathfrak{M}_L$ where all one-cycles in $H_1(L, \mathbb{Z})$ pinch and these should be identified with the finite webs i.e. the critical leaves. Moreover, if $\mathfrak{D} \subset \mathfrak{M}_L$ is the collection of such points, the 5d BPS count associated with $[L]$ is given by $\Omega(L) = (-1)^{b_1(L)}|\mathfrak{D}|$. However, we should strongly emphasize that this analysis is based on (92) which is the classical space of deformations of an A-brane of charge $[L]$, in general this should be corrected by superpotential relations. In the examples considered in [11, 26] and for instance, for small number of D0-branes, the BPS superpotential vanishes identically, allowing us to rely on the classical space of deformations.

Another consequence, suggested by the examples analyzed in [11, 26], is that $\mathcal{M}_L$ has an stratification where each strata is labeled by the number of pinched cycles at the corresponding points in $\mathfrak{M}_L$. Starting from the zero-dimensional stratum of the critical leaves, one can

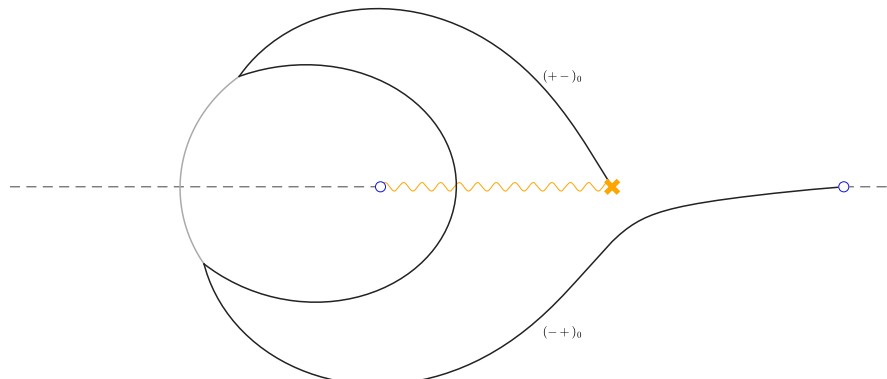

Figure 26: Deformation of the finite web for D0-D4 bound state.

parameterize each pinching cycle in terms of some non-contractible cycle in the corresponding foliation which belongs to a generic point in the moduli space $\mathfrak{M}_L$. Consequently, one can study $\mathfrak{M}_L$ near each fixed point through Lagrangian surgeries that restore one pinched cycle at a time, and study how it degenerates. Finally, we can recover certain strata of the moduli space of A-branes $\mathcal{M}_L$ via (92).

To apply this to the case at hand, we would need to ensure two crucial points: First of all, the finite webs that we have identified should indeed be the fixed points under the action of $T^{b_1(L)}$ introduced in (92) in the space of bound states of one D4 and $n$ D0-branes. Secondly, there should be a cell decomposition in the sense of [26] of the A-brane moduli space. This is also very reasonable to expect. Indeed, the Hilbert scheme of points in $\mathbb{C}^2$ does admit a cell decomposition where the cells retract to fixed points of the $(\mathbb{C}^\times)^2$-action on it [46]. Thus, given our identification of the finite webs precisely with these fixed points, and assuming that these finite webs indeed arise as $T^{b_1(L)}$-fixed points on the A-brane moduli space, the cell-decomposition of the A-brane moduli space seems very natural.

Leaving a more thorough investigation of these matters for the future, we will here be content with exploring some low-dimensional strata of the moduli space of calibrated finite webs on $\mathbb{C}_x^\times$ that arise (for fixed $\vartheta = \vartheta_c$) by deformation away from the critical exponential networks for small values of $n$, interpreted as (purported) moduli space of special Lagrangians in the full three-fold geometry. For finite webs, one option to do this is via oriented Lagrangian surgery, such as to free up one of the pinched cycles and then letting it parameterize a part of the moduli space. We will find substantial evidence that this indeed can be viewed as the beginning of the above-mentioned cell decomposition.

$\boldsymbol{n = 1}$   In this case, there is only one finite critical web of the shape in figure 8. Through surgery one obtains a bubble with a gray line as in figure 26 (see also fig. 19). In view of $\mathrm{Hilb}^1(\mathbb{C}^2) \cong \mathbb{C}^2$, the fixed point is the origin. Performing surgery on the $+$-sheet, the length of the gray line is the distance of the corresponding point lying on one of the axes of $\mathbb{R}^2$ in the moduli space of foliations, while surgery on the $-$-sheet corresponds to a point on the other axis. Thus, we conclude $\mathfrak{M}_L \cong (\mathbb{R}_{\geq 0})^2$, which reproduces $\mathcal{M}_L \cong \mathbb{C}^2$.

$\boldsymbol{n = 2}$   Factoring our translations and blowing up the symmetric product gives $\mathrm{Hilb}^2(\mathbb{C}^2) = \mathbb{C}^2 \times \mathcal{O}_{\mathbb{P}^1}(-2)$, which has an exceptional curve with two torus fixed points. Although we do not yet know how to describe the full moduli space using foliations, we can identify the stratum containing the two fixed points by deformation of the critical finite web

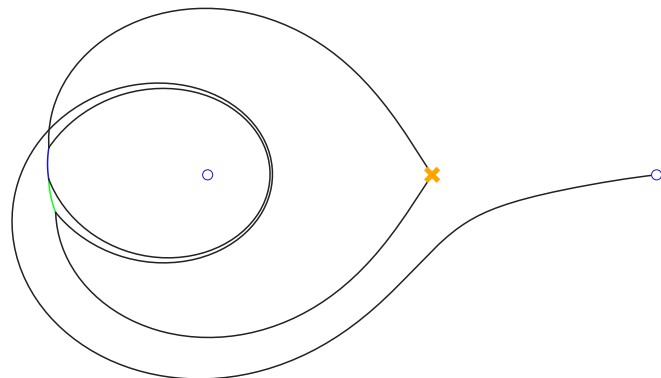

Figure 27: Interpolating the finite webs for the $(2)$ and $(1,1)$ partitions along an interval.

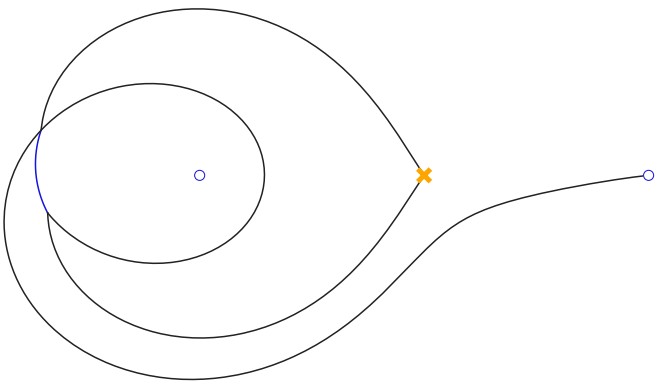

Figure 28: Midpoint of interpolation in fig. 27.

shown in fig. 13. For inspiration, we look at the ADHM representation matrices from section 2.1, which in this stratum take the form [22]

$$B_1 = \begin{pmatrix} \lambda & 0 \\ \alpha & \lambda \end{pmatrix}, \quad B_2 = \begin{pmatrix} \mu & 0 \\ \beta & \mu \end{pmatrix}, \quad I = \begin{pmatrix} \sqrt{2\zeta} \\ 0 \end{pmatrix}, \tag{93}$$

where $[\alpha : \beta] \in \mathbb{P}^1$. Inserted in (14) one obtains the equation $|\alpha|^2 + |\beta|^2 = \zeta$, and the two torus fixed points correspond to the endpoints of this interval. Indeed, as we have emphasized throughout our correspondence between partitions and finite webs, the matrix elements can be identified with the surgery parameters of the finite webs. In figure 13 in section 3.4, if the green line is of type $(++)_{-1}$, it corresponds to the case $\alpha = 0$ and if it is of type $(--)_{-1}$, $\beta = 0$.

To recover the rest of the exceptional $\mathbb{P}^1$ fiber of the Hilbert-Chow morphism over 0 parameterized by $[\alpha : \beta]$, we can follow the same logic with generic values of the surgery parameters, and indeed find that there is a suitable family of finite webs as shown in fig. 27. Starting from (say) the fixed point with green line of type $(++)_{-1}$, the blue line (of type $(-+)_{-2}$) starts growing until the green line has shrunk to zero size, at which point we can grow it back with opposite type $(--)_{-1}$ to collapse the blue line again and reach the second fixed point. The exceptional $\mathbb{P}^1$ is then the $U(1)$-fibration over the interval parameterizing this family, realized by $U(1)$ local systems on the corresponding special Lagrangians.

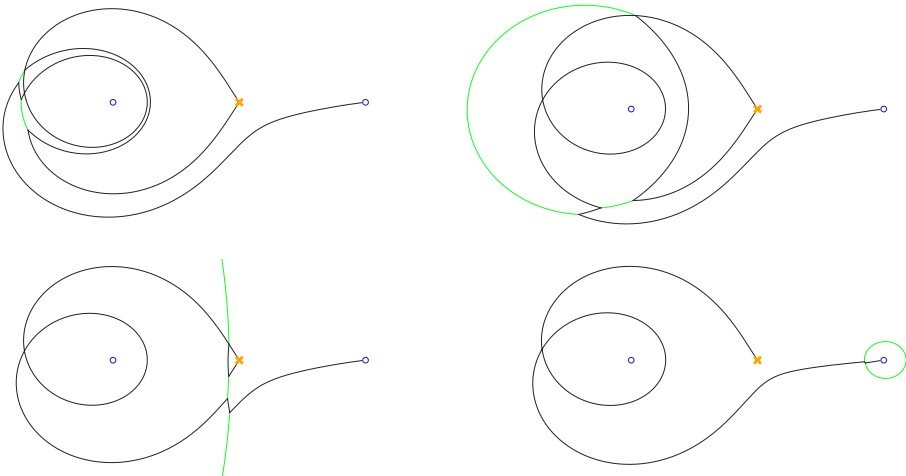

Figure 29: A sequence of deformations of 2D0-D4.

The finite web representing a bound state with $n = 2$, but without any green line, at the midpoint of the interval, shown in fig. 28, is intriguing in several ways. From the point of view of moduli, it seems to correspond to a special locus on the exceptional $\mathbb{P}^1 \subset \mathrm{Hilb}^2(\mathbb{C}^2)$ that is otherwise unknown, or at least we are not aware of any other stratum touching that point. From the point of view of wall-crossing in exponential networks, one should emphasize that although this finite web looks tantalizingly similar to those that we have identified as torus fixed points, one can check that it does in fact *not* contain any double $\mathcal{E}$-walls in the sense of section 3.2, and will hence not contribute to the BPS index of eq. 65. How exactly these observations fit into the broader picture will however, as remarked before, have to be reserved for a future investigation.

We can also move off the exceptional $\mathbb{P}^1$, corresponding to having one D0 at the origin and the other on one of the coordinate axis. We show a sequence of such deformations in fig. 29, the last of which can be interpreted as attaching a generic D0 in the moduli space presented in [7] to the D0-D4 fixed point of fig. 8.

**$n = 3$** Continuing in this vein, a generic point on the Hilbert scheme off the exceptional divisor looks as in fig. 30, in which each green circle corresponds to a D0-brane that has been moved away from the origin along one of the coordinate axes.[23]

An interpolation between the $(1, 1, 1)$ and $(3)$ partitions that generalizes fig. 27 is shown in figs. 31 and 32. The $\mathbb{P}^1$ that is parameterized by the concatenation of these intervals corresponds, at the level of quiver representations, to the family of matrices

$$B_1' = \alpha B_1 + \beta B_2, \qquad B_2' = \alpha B_2 + \beta B_1, \tag{94}$$

where $B_1$ and $B_2$ represent the torus fixed points and satisfy $[B_1, B_2^\dagger] = [B_2, B_1^\dagger]^\dagger = 0$ such that each member of the family satisfies the D-term. The web at the midpoint of the interval in this case contains a single blue line of type $(-+)_{-3}$ and with the appropriate changes looks just like fig. 28.

Resolving instead only one of the self-intersections results in the family of finite webs show in fig. 33, which corresponds to an interpolation between partitions $(3)/(1, 1, 1)$ and $(2, 1)$ along a rational curve in moduli space.

---

[23]A generic leaf of foliation has some real moduli which get fixed at the fixed point. A fully generic foliation has the maximal number of real moduli possible [26].

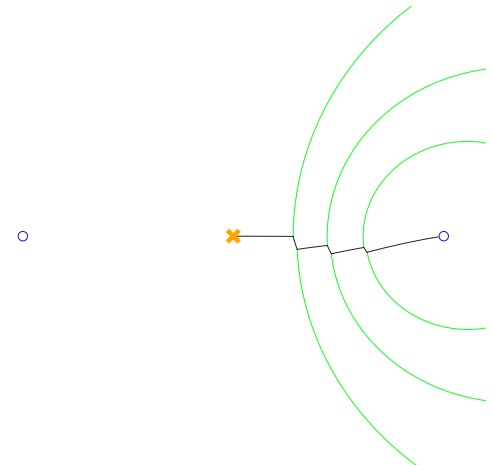

Figure 30: Fully generic deformation of 3D0-D4.

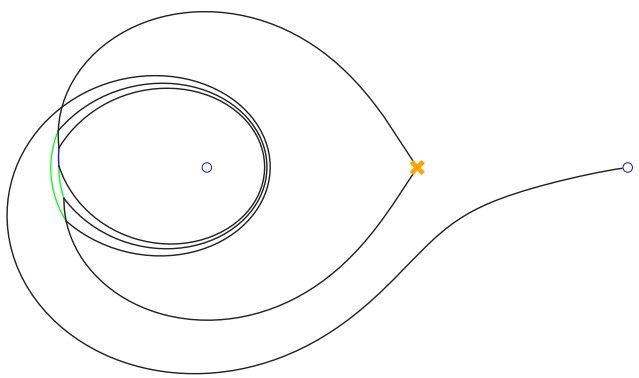

Figure 31: Interpolating from $(3)$ to $(1, 1, 1)$, part A.

**$n = 4$**    Adding another D0-brane by surgery to this last example produces the family of finite web shown in fig. 34. This corresponds to an interpolation between the partition $(4)$ and the initially somewhat mysterious $(3, 1)$, and sheds some light on the origin of the "vestigial line" in the latter that we observed in section 3.4. Indeed, by a happy numerical coincidence, the two outer green lines coalesce precisely at the same time as the two black "core" lines running around the puncture at infinity, at which point we can replace the 2 $(--)_{-1}$ lines with the indistinguishable $(--)_{-2}$ to obtain exactly fig. 18.

## 5.2    More on obstructions...

Following observations of section 3.4, finite webs in critical exponential networks that bound uncanceled holomorphic disks can be interpreted as "stable non-modules", by which roughly speaking we mean torus-fixed points in the quiver configuration space that solve the D-terms,[24] but violate the F-terms. In the example specifically, holomorphic disks passing through the self-intersection of the D0-brane at the branch point can be identified with particular entries in the ADHM relation $[B_1, B_2] - IJ$ from (5), viewed as the derivative of the superpotential (3) with respect to the field $B_3$ that originates at that point.

---

[24]Algebraically, it would mean that there is a stability vector generating $\mathbb{C}^n$ under the action of $B_1$ and $B_2$.

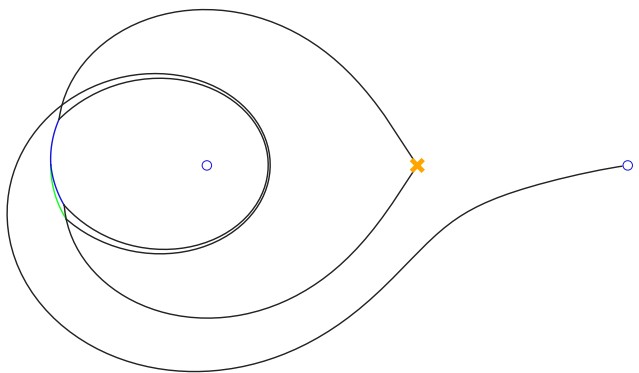

Figure 32: Interpolating from (3) to (1, 1, 1), part B.

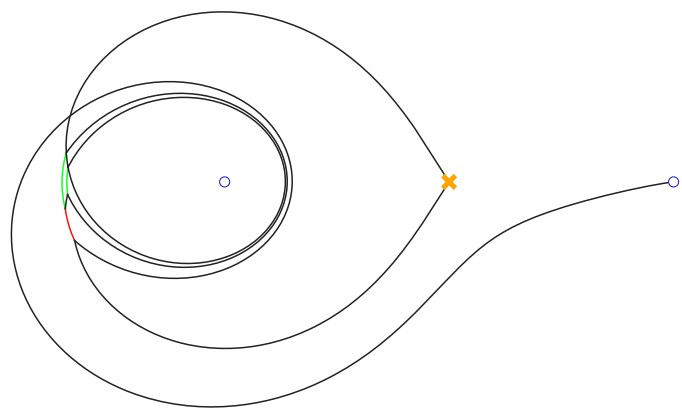

Figure 33: Finite web interpolating (3) and (2, 1).

As further evidence for this interpretation, we show in fig. 35 a critical finite web in the class of one D4 with $n = 3$ D0-branes, two of which have their "flaps" connected and detached from the branch point. According to the by now standard identification of surgery parameters with representation data, if the green line is of type $(--)_{-1}$, this finite web should correspond to the matrices

$$B_1 = \begin{pmatrix} 0 & 0 & 0 \\ 1 & 0 & 0 \\ 0 & 0 & 0 \end{pmatrix}, \quad B_2 = 0, \quad B_3 = \begin{pmatrix} 0 & 0 & 0 \\ 0 & 0 & 0 \\ 0 & 1 & 0 \end{pmatrix}, \quad I = \begin{pmatrix} 1 \\ 0 \\ 0 \end{pmatrix}, \tag{95}$$

viewed as a "torus-fixed stable non-module" of the extended ADHM quiver (2), and indeed it is not hard to locate a holomorphic disk in fig. 35 corresponding to the non-zero entries in $[B_1, B_3] \neq 0$. Exchanging $B_1$ and $B_2$ of course corresponds to the green line being of type $(++)_{-1}$, which then covers all stable obstructed fixed points at $n = 3$.[25] So again holomor-

---

[25]The moduli space of obstructed networks for general $n$ is conjecturally given by

$$\left\{ (I, B_1, B_2, B_3) \in \mathbb{C}^n \oplus \mathrm{End}(\mathbb{C}^n)^3 \ \middle| \ \begin{matrix} \mathbb{C}\{B_1, B_2, B_3\} \cdot I = \mathbb{C}^n \\ B_3 I = 0 \end{matrix} \right\} /\!\!/ GL(\mathbb{C}^n).$$

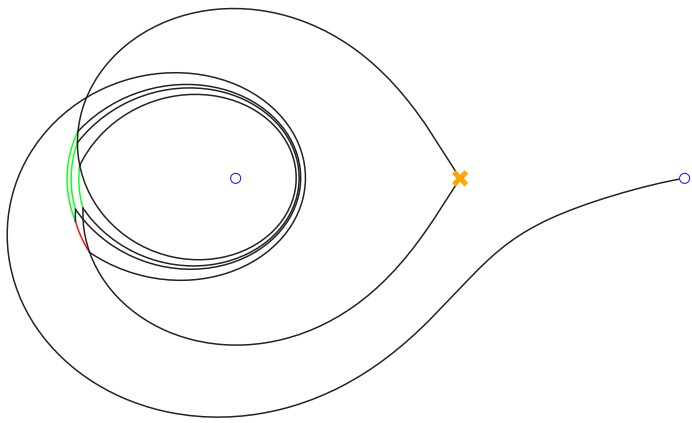

Figure 34: Finite web interpolating $(4)$ and $(3,1)$.

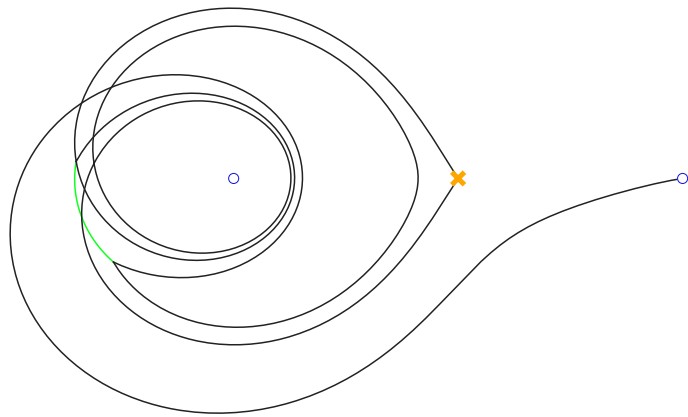

Figure 35: Finite web corresponding to "non-module" for $n = 3$ with $B_3 \neq 0$.

phic disks justify our ignoring fig. 35 for the purposes of BPS counting, if we did not a priori exclude any finite web involving a surgery between different D0-branes at the branch point. An interesting aspect of this identification is that the calibrated surgery is possible at all, while in general [47] one expects to be able to form bound states between D0-branes using $B_3$ only under framing by a D6-brane, which at the moment we do not know how to realize at the level of (exponential) networks. That the requisite entry in $B_3$ is in fact tachyonic can be seen for example by superposing the finite web in fig. 13 with the plain D0 from fig. 5.

## 5.3 ...and their possible resolution via ghost webs

The most pressing question in the story however might well be the relation between the obstruction of finite webs by holomorphic disks and the wall-crossing formalism for BPS state counting of [9, 10] that we reviewed in section 3.2. Indeed, while according to our identification obstructed webs are stable fixed points of the torus action, they do not represent any point in quiver moduli, which is the main reason for excluding them from the count. Yet they are

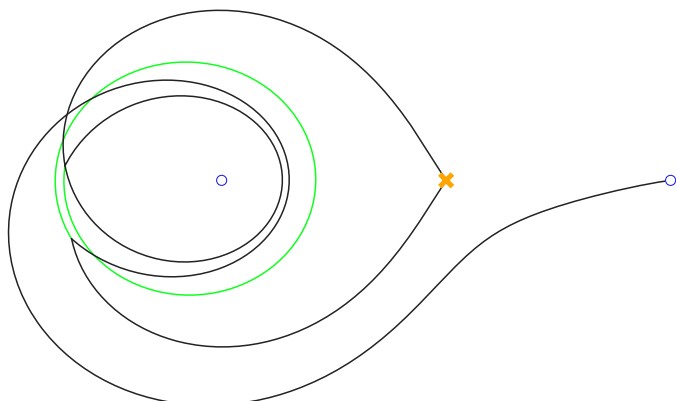

Figure 36: Ghost web for obstructed network from fig. 15.

included in the double walls $\mathcal{W}_c$ inside the critical exponential network $\mathcal{W}(\vartheta_c)$ at the appropriate angle $\vartheta_c = \vartheta_{(1,n)}$. They would hence appear to contribute in eq. (63), and must either be excluded explicitly by hand, or else cancel against some other "critical" contribution that we have so far ignored.

To entertain this latter possibility, we recall the observation, recorded around eq. (78), that a general holomorphic disk obstructing one of our finite webs will end on some closed subcycle of the cycle in $\widetilde{\Sigma}$ lifting the finite web with vanishing period integral (66). As a consequence, one expects it to be possible to contract the holomorphic disk by deforming the cycle without changing the central charge. In the first and simplest example, coming from the finite web in fig. 15 with two green lines of different type, this is indeed possible. The resulting finite web is shown in fig. 36. Here, the green line necessarily crosses the branch cut, where (++) and (−−) get exchanged. This would not be compatible with the two green lines initially being of the same type. While these finite webs cannot be obtained by small surgery from our basic constituents (in particular, they do not coalesce onto one D4 plus 3 D0-branes in the limit $\vartheta_{D4} \to 0$), they are in fact part of the double wall network $\mathcal{W}_c$, and it seems reasonable that they will contribute via (65) to canceling our obstructed webs.

The same mechanism also works for the finite web in fig. 35 corresponding to a non-zero vev for $B_3$, where contracting the holomorphic disk leads to the finite web shown in fig. 37.

Thus we see that at least for $n = 3$, each obstructed finite web appears to be accompanied by another "ghost web" that results from contracting the uncanceled holomorphic disk. The formal interpretation could be that the finite "critical" webs are in fact organized in some kind of graded complex whose degree 0 piece is generated by webs corresponding to stable modules for (2), though possibly violating the F-terms, and the differential comes from (the dual of) contracting holomorphic disks. In an ideal world this complex would have cohomology concentrated in degree 0 with $\dim H^0 = \Omega(\gamma)$. Although this would be a very satisfying result, at this stage we have little to offer beyond the case $n = 3$ that we just recorded (while the cases $n = 1$ and $n = 2$ are trivial).

As a small piece of evidence for $n = 4$, we give an account of the ghost webs for the 6 obstructed webs obtained from fig. 38 by mixing green lines of different type. By initially contracting the two (potential) disks visible in the figure, we find the 2 webs shown in fig. 39, each of which supports 4 choices of labels for a total of 8 ghost webs. This clearly is an overcounting as a result of the finite webs with label sequence (++)(−−)(++) and (−−)(++)(−−)

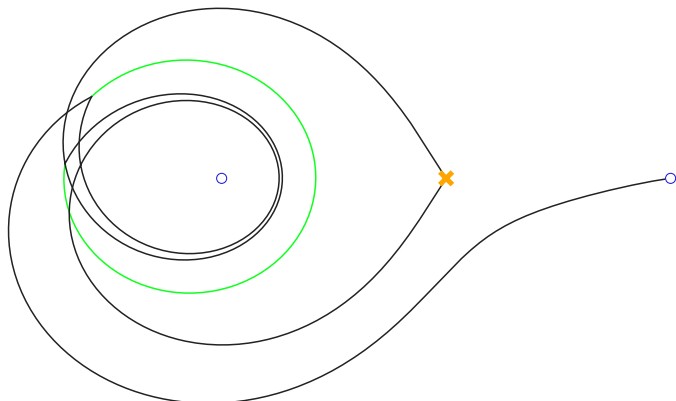

Figure 37: Ghost web for obstructed network from fig. 35.

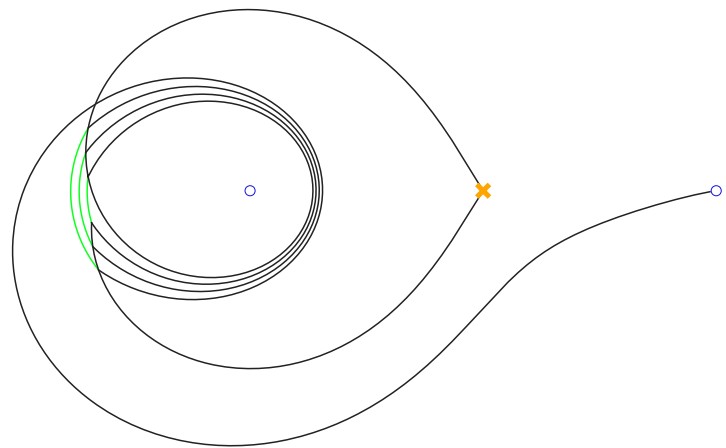

Figure 38: Finite web corresponding to $(4)$ and $(1,1,1,1)$ partition.

being canceled twice, and should have a familiar resolution from "ghosts for ghosts". Indeed we find two such, as shown in fig. 40, such that our final count could again be balanced.

In conclusion, the combinatorics of these purported ghost networks is already quite com-

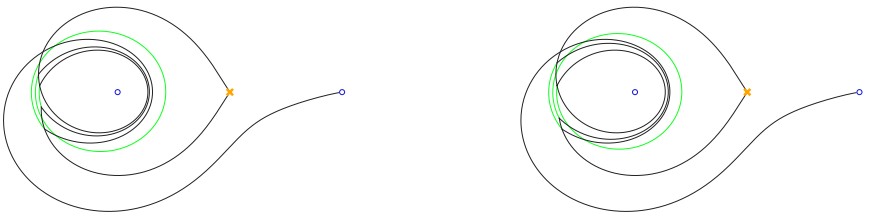

Figure 39: Ghosts for the obstructed finite webs in fig. 38.

plicated for $n = 4$ since the webs in fig. 38 can have 2 non-canceling holomorphic disks, and it is not immediately clear how this will generalize. In particular, we have observed that also canceling holomorphic disks can be contracted to give ghost webs and "ghosts for ghosts". For example, contracting both holomorphic disks in the finite web for the $(2, 2)$ partition in figure 17 gives rise to the "ghost for ghosts" web shown in figure 41. The elucidation of these ghost networks and their possible role in the relation between finite webs and non-abelianization as presented in section 3.2 is high on the list for future investigation.

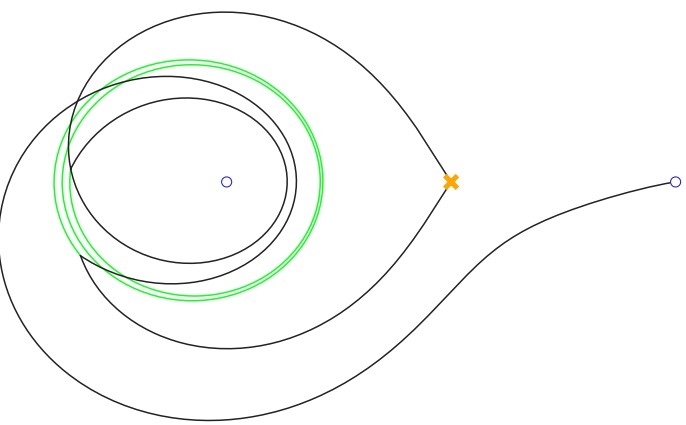

Figure 40: "Ghost for ghosts" web for fig. 39.

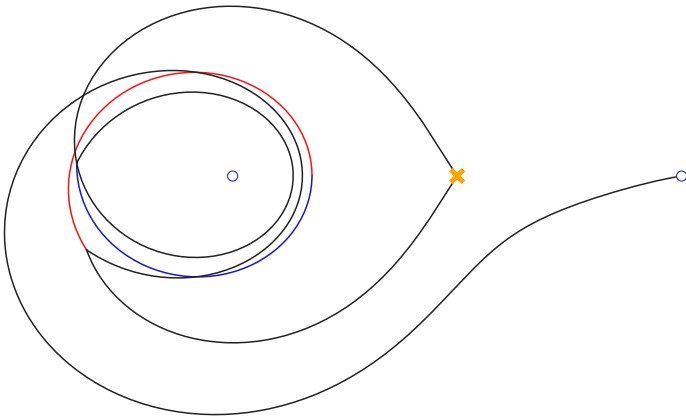

Figure 41: "Ghost for ghosts" for the finite web in fig. 17.

# Acknowledgments

We would like to thank P. Longhi, L. Merkens, B. Pozzetti, V. Tatitscheff for valuable discussion and collaboration on related projects, and all and any available ADHM enthusiasts, including in particular T. Dimofte, M. Kontsevich, N. Nekrasov, I. Saberi, for much-needed encouragement. S.B. thanks Max Planck Institute for Mathematics, Bonn where part of the work was done, and acknowledges Heidelberg University for providing excellent working conditions at the late stages of this work. R.S. expresses his sincere gratitude to the Institut des Hautes Études Scientifiques (IHES) in Bures-sur-Yvette, for their gracious hospitality. J.W. thanks the organizers of the TSIMF conference on "Homological Algebra of the Infrared", where this work was presented, and YMSC for kind hospitality during finalization of the paper.

**Funding information** M.R. acknowledges support from the National Key Research and Development Program of China, grant No. 2020YFA0713000, the Research Fund for International Young Scientists, NSFC grant No. 1195041050. MR also acknowledges Heidelberg University, IHES, SCGP, Steklov Institute, Moscow for hospitality at the final stages of this work. The work of S.B. has been supported in part by the ERC-SyG project "Recursive and Exact New Quantum Theory" (ReNewQuantum), which received funding from the European Research Council (ERC) under the European Union's Horizon 2020 research and innovation program, grant agreement No. 810573. J.W.'s partial funding statement reads: This work is funded by the Deutsche Forschungsgemeinschaft (DFG, German Research Foundation) under Germany's Excellence Strategy EXC 2181/1 — 390900948 (the Heidelberg STRUCTURES Excellence Cluster).

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
