# Peer review of "Exponential Networks for Linear Partitions"

_SciPost Physics, doi:SciPost Phys. 18, 128 (2025)_

## Round 2 · Referee Report · Anonymous (Referee 1) · 2024-5-21

Report

This paper is a continuation in a series of papers discussing exponential network techniques applied to counting BPS D-brnaes on Calabi-Yau threefolds. The heart of this technique is in consideration of a mirror dual curve, then D-branes are dual to sLags, whose forms and filling numbers may be identified explicitly from solving first order differential equations numerically. In this paper authors concentrate on the case of C3 when a non-compact D4-brane is present. From the point of view of an effective quiver QFT this situation is captured by specific quiver framing. In this particular case the quiver setup boils down to a Hilbert scheme of points on C2. Authors propose to filter out the BPS spectrum in this case by a specific limit for the C3 mirror curve. The proposal is confirmed by explicit comparison of the resulting BPS spectrum with equivariant fixed points on Hilb(C2). Speculations on the life outside the fixed point locus are also present in the text. On the quiver theory side framing imposes significant corrections to the BPS spectra and may be involved in wall-crossing via mutations. This is why it is important to study a proper mirror dualization of the framing elements, and this manuscript is the first step down this route. This research is new and vibrant. And I definitely recommend this paper for publication. However if the authors choose to do so, I would suggest substantial modifications in the text structure: 1) Review sections 2 and 3 seem too be rather excessive. This is a bulky textbook material the reader might easily acquire from other sources. And it is hard to grasp it from a concise summary of this paper. For example, parallel transport variables introduced in (3.12) seem to be never used outside this section. 2) What review material seems to be missing is from references [6] and [8] where the cut-off proposal to mimic D4-brane was born. The very proposal why a suitable cut-off would mimic quiver framing is not apparent: a) As a naive answer to the question why the BPS spectrum is equivalent to EQUIVARIANT fixed points I would suggest that the D4-divisor was chosen in an equivariant way -- as a (x,y)-plane in C3. However there are two more apparent choices (x,z) and (y,z) that would be distinguished by the last term in superpotential (1.3): -IJB_2 or -IJB_1. Should those be other punctures in Figure 9? b) Surely, it is natural to ask what to do if the D4-divisor (or even D2-divisor) is chosen in a more generic way, say two intersecting planes in C3? c) It seems to be not very clear if eq.(3.33) has a priori origin. Do we first calculate the BPS spectrum then compare it with fixed pints on Hilb(C2) and afterwards identify dual objects in two spectra, or is there a mechanism to dualize D4-brane to a sLag on the mirror curve directly?

Recommendation

Ask for minor revision

  • validity: -
  • significance: -
  • originality: -
  • clarity: -
  • formatting: -
  • grammar: -

Author:  Johannes Walcher  on 2024-10-08  [id 4846]

(in reply to Report 1 on 2024-05-21)
Category:
answer to question

  1. Question: Review sections 2 and 3 seem too be rather excessive. This is a bulky textbook material the reader might easily acquire from other sources. And it is hard to grasp it from a concise summary of this paper. For example, parallel transport variables introduced in (3.12) seem to be never used outside this section. Answer : The intention of the section 2 was to introduce the torus fixed points and Young diagrams (box stacking), while also providing the matrices that correspond to these fixed points. All of these ingredients are crucial for our later presentation. Subsection 2.1 does review some well-established tools (such as GIT quotient and symplectic quotient). However, different points of views are useful for understanding different aspects of our investigations of the A-brane moduli spaces later. So to keep the paper self-contained, we choose to keep section 2 as it stands. As for section 3, we are not aware of any textbook that has these materials. Some parts indeed build on what was known from before, such as geometric engineering of the BPS states that we propose to count. However, part of it delves into connecting with non-abelianization that was introduced in this context in [8] (as an anecdote, the justification based on physics is very important to define the notion of counting) and subsection 3.1 provides an alternative point of view of counting based on the geometry of the network, expanding on [6], [10]. In this paper, we exhibit clearly the parallel and complementarity of these two approaches. The rest of section 3 is quite new in our opinion. Some of this was indeed hinted at in [6], however from section 3.3 onwards, new notions were introduced. In section 3.3 we define finite webs and connect them to exponential networks. In section 3.4, we describe the bound states of a D4 with $n$ D0 branes. Most importantly, from the point of view of the A-branes, how to form these bound states via Lagrangian surgery, how to match them with the ADHM modules and the Young diagrams are described in details with examples. Finally, the concept of anomaly free networks is also introduced. We are not aware if it exists and is utilized to get the right count of finite webs, elsewhere. So we would like to keep section 3 untouched as well.
  2. Question : What review material seems to be missing is from references [6] and [8] where the cut-off proposal to mimic D4-brane was born. The very proposal why a suitable cut-off would mimic quiver framing is not apparent: (a) As a naive answer to the question why the BPS spectrum is equivalent to EQUIVARIANT fixed points I would suggest that the D4-divisor was chosen in an equivariant way -- as a $(x,y)$-plane in $\mathbb C^3$. However there are two more apparent choices $(x,z)$ and $(y,z)$ that would be distinguished by the last term in superpotential (1.3): $-IJB_2$ or -$IJB_1.$ Should those be other punctures in Figure 9? (b) Surely, it is natural to ask what to do if the D4-divisor (or even D2-divisor) is chosen in a more generic way, say two intersecting planes in $\mathbb C^3$? (c) It seems to be not very clear if eq.(3.33) has a priori origin. Do we first calculate the BPS spectrum then compare it with fixed pints on ${\rm Hilb}(\math C^2)$ and afterwards identify dual objects in two spectra, or is there a mechanism to dualize D4-brane to a sLag on the mirror curve directly? Answer : The cut-off is introduced from the point of view of the networks in section 3.3. At the end of section 3.1., we briefly recall the justifaction based on decompactification of local P2 geometry. It is correct that the D4 is distinguished by being equivariant (this is also discussed to some extent at the end of section 3.1). Among the equivariant divisors, the $(x,y)$-plane is further distinguished by being dual to the base curve C understood as the classical moduli space of the Harvey-Lawson Lagrangian brane. This is mentioned at the beginning of section 3.2. Topologically, $(x,z)$ and $(y,z)$ divisors would indeed correspond to cycles connecting other punctures of the mirror curve. However, these do not correspond to proper exponential networks under the projection to $x$. To achieve this, one would have to project to $1/x$ (or $y/x$, respectively). We have added figure 7 that should further clarify the state of affairs. Finally, we agree that it would be nice to give an a priori justification of the identification of D0 and D4, based on SYZ dualization. We believe that this is possible, but chose not to pursue this direction in the present paper.

---

## Round 2 · Referee Report · Anonymous (Referee 2) · 2024-9-1

Report

Review for arXiv:2403.14588 “Exponential Networks for Linear Partitions”

The aim of this paper is to find the mirror statement of the counting of D4-D0 brane configurations, where the D4-brane is wrapped on a divisor of a non-compact toric CY, which in this paper is taken to be $\mathbb{C}^3$. The paper argues that the torus-invariant D4-D0 brane configurations correspond to certain finite webs in the exponential network on the base of the mirror curve.

The paper is a continuation of the earlier collaboration [6], where a first attempt was made in the identification between the Hilbert scheme of points on $\mathbb{C}^2$ and special Lagrangians in the mirror manifold. This paper resolves a few important issues from [6], leading to an explicit correspondence between 2d partitions and finite webs, and presents some new ideas beyond the fixed point analysis.

In my opinion this paper is certainly strong enough to be accepted for publishing. Yet, I would recommend the authors to spend a little time on restructuring sections 3 and 4 and adding in some additional explanations. The paper tries to be self-contained by recapitulating a lot of the necessary background but does not quite achieve this goal because a few essential explanations are missing.

In particular:

P3: The introduction would benefit from including a few important arguments, that are presumably found in [6] but seem essential to repeat here. It would be good to summarize the dualities that relate the M5-brane picture to the IIB mirror picture, and specifically to review how the D0-D2-D4-D6 configurations on the toric CY correspond to BPS configurations in the mirror CY. For instance, there may be a known correspondence between D2 and D4’s with special Lagrangian manifolds in the mirror projecting down to A- and B-cycles on the mirror curve? And why would we expect D0’s and D6’s to correspond to interesting configurations on the mirror curve at all?

P15: “D4-branes completely transverse to R^{3,1}”. Is this correct? Shouldn’t they have a leg in R^{3,1} on which you find a SQM? Also, phrasing this in terms of coherent sheaves is a bit confusing, as you are not including D2’s here. Might it actually be interesting to consider bound states including non-compact D2’s?

P15: In the second paragraph of section 3 you are comparing the D4-D0 setup with studying Lagrangians in the mirror. Perhaps you can remind the reader about the dualities that are required to make this relation? (Or refer to an earlier explanation.)

P16: Might it be possible to explain the “charge conservation” in more detail near eqn (3.9)?

P20: In the paragraph above section 3.2 you again note the relation between the D4-D0 setup on the IIA side with studying Lagrangians on the mirror, without explanation.

P21: “Each of these vacua …, so each of them …”. Could you explain why there would be a correspondence between BPS configurations corresponding to some special Lagrangian L and points on the deformation space of L?

P21: “.. the kinky vortices .. can interpolate between .. gauge field at infinity.” Could you say which object the M2-branes correspond to on the IIB side? And are you referring at the end of this sentence to some gauge field at infinity on the IIB side?

P22: “.. can be viewed as defining, in dependence .. ” Could you make this more precise? It’s probably the fact that the path doesn’t cross any wall which implies the corresponding connection is flat?

P27: Section 3 is rather long. So far it has been a review of exponential networks, whereas section 3.3 starts the comparison with D4-D0-systems. Might it make sense to start a new section here?

P27: Around eqn (3.32), wouldn’t it be easier to see $\Sigma$ as a 1-sheeted covering?

P27: “This identification was justified rigorously … “ Would it be possible to turn that sentence into an intuitive explanation of why this network could represent a D0-brane? And what happens to the torus action on the IIB-side?

P28: “we need to take into account also … “ Please refer to Figure 6.

P28: “toric divisor dual to the base curve $C$ ..” Isn’t it better to phrase this in the IIA language?

P29: “Together with the three self-intersections of the D0-brane ..” Could you perhaps remind the reader why there are three self-intersections? Possibly on P27 already, as this justifies the identification as a D0-brane.

P29: Perhaps you could also remind the reader about how the holomorphic discs in the ADHM description appear in the exponential network picture?

P30: So far you have introduced how you are about to find the relevant exponential networks, which is mostly clear. Yet, the two paragraphs on P30 are abracadabra to me. Might it be possible to add more details to the first paragraph, perhaps at the end of section 3, when the reader has seen some examples. Similarly, it might be helpful for the reader to either illuminate all the quotes in the second paragraph with figures (possibly in the examples) or leave the technicalities for the end of section 3.

P41: Further to my previous question about holomorphic discs, could you perhaps add some explanations here?

P42: Might it be possible to include the exponential network corresponding to the (4) partition?

P47: “that web has to be deformed, both away from ϑ = 0, and out on its moduli space,” This is slightly confusing to me, the web just appears at the new $\theta_{1,n}$, no?

Section 4 in general: sections 3.3/3.4 and section 4 describe the networks from slightly different perspectives, but it feels like there could be a bit more cohesion between the sections.

Section 5: as an additional question, what stands in the way of characterising D6-D0 bound states (with the same techniques used in this paper), or even D6-D4-D2-D0 bound states?

Requested changes

And some typos:

P4 “space space”
P9 ‘hyerkahler”
P21: “quantum of gauge field”
P27: “the infinite tower of descendent wall(s) that originate(s) at the self-intersection”
P30: “multiplicies”
P59 “\mathbb{P}^1, fiber”

Recommendation

Ask for minor revision

  • validity: -
  • significance: -
  • originality: -
  • clarity: -
  • formatting: -
  • grammar: -

Author:  Johannes Walcher  on 2024-10-08  [id 4847]

(in reply to Report 2 on 2024-09-01)

  1. Question : P3: The introduction would benefit from including a few important arguments, that are presumably found in [6] but seem essential to repeat here. It would be good to summarize the dualities that relate the M5-brane picture to the IIB mirror picture, and specifically to review how the D0-D2-D4-D6 configurations on the toric CY correspond to BPS configurations in the mirror CY. For instance, there may be a known correspondence between D2 and D4's with special Lagrangian manifolds in the mirror projecting down to A- and B-cycles on the mirror curve? And why would we expect D0's and D6's to correspond to interesting configurations on the mirror curve at all? Answer : The chain of dualities starts with reducing the M5- to NS5-brane, followed by a transverse T-duality and a 9/11'' flip. We have added one fairly recent reference that emphasizes the utility for relating theWitten'' and ``Vafa'' dscriptions of geometric engineering. That the exact mapping of the cycles is however cumbersome, and that in particular the fate of the D6-brane remains mysterious, is indeed one of the main motivations for our paper.

  2. Question : P15: “D4-branes completely transverse to ${\mathbb R}^{3,1}$. Is this correct? Shouldn't they have a leg in ${\mathbb R}^{3,1}$ on which you find a SQM? Also, phrasing this in terms of coherent sheaves is a bit confusing, as you are not including D2's here. Might it actually be interesting to consider bound states including non-compact D2's? Answer : We have corrected the statement about wrapping of D4-branes. Indeed D4/D0-brane bound states are only a subset of all possible configurations.

  3. Question : P15: In the second paragraph of section 3 you are comparing the D4-D0 setup with studying Lagrangians in the mirror. Perhaps you can remind the reader about the dualities that are required to make this relation? (Or refer to an earlier explanation.) Answer : The identification is expected on general grounds of mirror symmetry, as initiated and championed by Vafa and collaborators in the early 2000's. Some of the most relevant works are cited in this \S.

  4. Question :} P16: Might it be possible to explain the “charge conservation” in more detail near eqn (3.9)? Answer : We are just referring to the fact that the individual strands, labelled by the sheet $i$ of the covering and the integer winding number $n$, have to match up on $\Sigma$. We find the analogy with Kirchhoff's first rule (the ``current law'') quite suggestive.

  5. Question : P20: In the paragraph above section 3.2 you again note the relation between the D4-D0 setup on the IIA side with studying Lagrangians on the mirror, without explanation. Answer : Indeed we are assuming some familiarity with the Hori-Vafa formulation of mirror symmetry, and in particular the Aganagic-Vafa point of view on the Harvey-Lawson fibration.

  6. Question : P21: “Each of these vacua …, so each of them …”. Could you explain why there would be a correspondence between BPS configurations corresponding to some special Lagrangian L and points on the deformation space of L? Answer : The classical moduli sace of special Lagrangian submanifolds is unobstructed and parameterized locally by the complexification of $H^1(L)$. Quantum mechanically, this is lifted, but any vacuum can still be thought of as living on the geometric deformation space.

  7. Question : P22: “.. can be viewed as defining, in dependence .. ” Could you make this more precise? It's probably the fact that the path doesn't cross any wall which implies the corresponding connection is flat Answer : The precise statements are contained in the ensuing sentences: Parallel transport is trivial (and hence trivially flat) on open charts before getting modified according to the kinky-vortex data. Preservation of flatness is equivalent to the wall-crossing formulas for the soliton degeneracies.

  8. Question : P27: Section 3 is rather long. So far it has been a review of exponential networks, whereas section 3.3 starts the comparison with D4-D0-systems. Might it make sense to start a new section here? Answer : We considered this option while writing of the paper, but in the end decided against it. The rationale was to gather all the minutia of wall-crossing and the ``experimental'' data for small values of $n$ in Section 3, before launching into the complete explanation of the correspondence in Section 4. We understand that one might feel differently about the distribution. However, at this stage there seems to be no clear benefit from dividing Section 3 that would justify the extra reorganizational work.

  9. Question : P27: Around eqn (3.32), wouldn't it be easier to see $\Sigma$ as a 1-sheeted covering? Answer : The upstairs curve having 3 punctures, while the ``downstairs'' $2$, implies that one-sheeted coverings cannot exist.

  10. Question : P27: “This identification was justified rigorously … “ Would it be possible to turn that sentence into an intuitive explanation of why this network could represent a D0-brane? And what happens to the torus action on the IIB-side? Answer : We have added one sentence about the intuitive ingredients of the identification. Preliminary remarks about the torus action on the IIB side appear in the previous work and here in Section 5, especially subsection 5.1.

  11. Question : P28: “we need to take into account also … “ Please refer to Figure 6. Answer : We have added another reference.

  12. Question : P28: “toric divisor dual to the base curve C ..” Isn't it better to phrase this in the IIA language Answer : The identification of $C={\mathbb C}^\times$ as moduli space of the Harvey-Lawson Lagrangian indeed is type IIA language.

  13. Question : P29: “Together with the three self-intersections of the D0-brane ..” Could you perhaps remind the reader why there are three self-intersections? Possibly on P27 already, as this justifies the identification as a D0-brane. Answer : We have added figure 7 to further clarify the state of affairs with the three self-intersections, the holomorphic disks, and the non-compact D4-branes.

  14. Question : P29: Perhaps you could also remind the reader about how the holomorphic discs in the ADHM description appear in the exponential network picture? Answer : This explanation is the content of page 41, the new fig.\ 7 guiding the intuition.

  15. Question : P30: So far you have introduced how you are about to find the relevant exponential networks, which is mostly clear. Yet, the two paragraphs on P30 are abracadabra to me. Might it be possible to add more details to the first paragraph, perhaps at the end of section 3, when the reader has seen some examples. Similarly, it might be helpful for the reader to either illuminate all the quotes in the second paragraph with figures (possibly in the examples) or leave the technicalities for the end of section 3. Answer : We understand that the content of these two paragraphs, which are crucial for the rigorous justification of our identification of linear partitions with finite webs, takes some time getting used to. For this reason, we have spent quite some time fine-tuning the wording. The first paragraph connects the counting of finite web to the proper definition of the enumerative invariants. The second paragraph appeals to the physical intuition relating Lagrangian surgery to open string tachyon condensation, which we hope readers of our paper will have been exposed to in their string theory course. It is hard for us to see how to rewrite these paragraphs in any better way.

  16. Question : P41: Further to my previous question about holomorphic discs, could you perhaps add some explanations here? Answer : Echoing some of the previous replies, we have found it most helpful to identify holomorphic disks by visualizing the combinatorics on the pair of pants depicted in the figures, especially the new fig.\ 7. We wish we had a better explanation at this point that we could share with the referee.

  17. Question : P42: Might it be possible to include the exponential network corresponding to the (4) partition Answer : The finite web for the (4) partition looks essentially like that for the (3) partition in fig.\ 15, with one extra loop and green line of multiplicity $1$. We have added a reference, but see no added value in reproducing it explicitly.

  18. Question : P47: ``that web has to be deformed, both away from $\vartheta =0$, and out on its moduli space,'' This is slightly confusing to me, the web just appears at the new $\vartheta_{1 , n}$, no? Answer : Indeed. Since $\vartheta$ is constant on the finite web representing the actualy bound state, the pieces corresponding to the individual D0-brane constituents have to be deformed away from their initial position and orientation.

  19. Question : Section 4 in general: sections 3.3/3.4 and section 4 describe the networks from slightly different perspectives, but it feels like there could be a bit more cohesion between the sections. Answer : Indeed, Section 4 is meant as a ``reboot'' of sorts for readers who did not have the patience to track all the strands in section 3.

  20. Question : Section 5: as an additional question, what stands in the way of characterising D6-D0 bound states (with the same techniques used in this paper), or even D6-D4-D2-D0 bound states? Answer : We do not know how to model the D6-brane using exponential networks.

---

## Round 3 · Referee Report · Anonymous (Referee 1) · 2024-11-5

Report

I recommend to publish the updated version.

Recommendation

Publish (meets expectations and criteria for this Journal)

---

## Round 3 · Referee Report · Anonymous (Referee 2) · 2024-12-24

Report

We thank the authors for taking the time to respond to our comments and questions, and even though we are still not keen on the ordering of section 3 and 4, we think most other comments have been addressed satisfactorily . We do have one remaining question regarding the top paragraph on p4. Could the authors perhaps explain the "local finiteness" in spectral networks vs exponential spectral networks and why this makes the GMN formalism more complicated? (Or perhaps refer to the relevant section in [13].)

Recommendation

Publish (easily meets expectations and criteria for this Journal; among top 50%)

---

## Round 3 · List of Changes

see replies to referee reports

---

## Round 4 · Author Response

We have clarified the meaning of the phrase "locally finite" in the comparison between exponential and spectral networks

---

## Round 4 · List of Changes

a new half-sentence at the top of page 4

---

## Editorial Decision

published